# A lateral hypothalamic region supporting diverse visual processing and modulation of visually-guided behaviour

J. W. Mouland [1], E. Tamayo[1], A. S. Ebrahimi [2], C. Williams[1], W. Fleming[1], A. Watson[1], M. P. Hogan [2], R. J. Lucas [2,3], R. Storchi [2,3] & T. M. Brown [1,3] ✉

Hypothalamic retinal input is traditionally considered distinct from the subcortical pathways supporting vision, specialised to adjust physiology and behaviour alongside variations in ambient illumination. Investigations of retinohypothalamic function have overwhelming focussed on the suprachiasmatic nucleus circadian clock, however. Here we employ multielectrode recording, viral tracing and chemogenetic manipulation in mice to show that another retinohypothalamic target, the anterior lateral hypothalamic area (LHA), displays diverse visual processing capabilities, supporting regulation of more complex visually-guided behaviours. Hence, while some visually responsive LHA cells track irradiance, a majority are highly selective for spatiotemporal contrast or motion signals. We further provide evidence for a retinotopic order to LHA visual responses, show that retinorecipient LHA neurons provide excitatory projections to behavioural control centres including the septal complex and habenula, and that LHA retinal inputs modulate behavioural responses to light flashes and 'looming' stimuli. Collectively, these data establish the LHA as a locus for regulation of visually-guided behaviours and environmental threat responses.

In addition to supporting vision, via projections to the visual thalamus and other accessory visual structures, a subset of retinal ganglion cells (RGCs) target the hypothalamus[1–4]. The established role for such projections is to regulate physiology and behaviour according to daily changes in ambient illumination, most notably via projections to the master circadian clock in the suprachiasmatic nucleus (SCN)[5,6]. Consistent with this view, retinal projections to the SCN overwhelmingly derive from a particular RGC class that are specialised to encode irradiance and are intrinsically photosensitive (ipRGCs), due to expression of the photopigment melanopsin[1,7–9]. Of note, however, the SCN is not the only hypothalamic target of RGCs, and the function of retinal input to these other regions and the sensory properties of the cells therein remain poorly understood. Hence, in addition to retinal projections to the ventrolateral preoptic area, which mediate light-driven sleep in nocturnal rodents[10–14], a lateral branch of the retinohypothalamic tract targets anterior portions of the lateral hypothalamic area (LHA; including a region sometimes also referred to as the peri-supraoptic nucleus, pSON)[2–4,15,16].

The presence of a retinal projection to the LHA is widely conserved across mammals (including humans)[17–19], but the properties and function of this pathway remain unknown. Hence, while calcium imaging studies have identified lateral hypothalamic cells responding to diverse sensory stimuli, including visual, the temporal resolution of such approaches have made it challenging to determine whether such responses might be directly driven by retinal inputs to the LHA (as opposed to more complex pathways or indirectly as a result of stimulus-driven behavioural responses)[20–22]. Nonetheless, given that the portion of the LHA region receiving retinal input projects to a range of brain regions implicated in control of goal-directed behaviours (including defensive, social, feeding/drinking)[15,23], it could in

[1]School of Medical Sciences, University of Manchester, Manchester, UK. [2]School of Biological Sciences, University of Manchester, Manchester, UK. [3]Centre for Biological Timing, University of Manchester, Manchester, UK. ✉e-mail: timothy.brown@manchester.ac.uk

principle play a variety of roles ranging from adjusting behavioural strategy according to environmental conditions to more complex regulation of visually-guided behaviours. Accordingly, while it is clear that, as for the SCN, retinal inputs to the LHA partly derive from ipRGCs[1–3,16], there is also evidence that other RGC types may project to this region[1,24]. Given also recent data indicating that even SCN neurons show more diverse and sophisticated visual response properties than previously expected[25,26] we set out to map the sensory properties of LHA visual neurons and better understand the functions of this unexplored visual circuit.

Here, then, we first comprehensively assess the photoreceptive mechanisms underlying LHA visual responses, employing mice where native M-cone opsin is replaced by the human L-cone opsin (Opn1mw^R)[27], which allows the use of multispectral stimuli to dissociate rod, cone and melanopsin photoreceptor contributions by increasing the separation between cone and melanopsin spectral sensitivity[26,28–30]. We go on to employ spatially structured stimuli to map the receptive field (RF) properties and assess the spatiotemporal tuning of LHA visual neurons. We next employ viral tracing approaches to identify RGC projections to the LHA and the projection targets of retinorecipient LHA neurons and, finally, employ selective chemogenetic manipulation of RGC inputs to the LHA to provide insight into the functional roles of the identified visual circuits.

## Results

### Contralateral retina-driven sustained and transient responses across subsets of lateral hypothalamic neurons

We surveyed light-evoked activity across lateral portions of the hypothalamus, focusing on the anterior aspect, previously shown to receive retinal input[2–4,15,16]. We started using 32-site linear probes to record multiunit activity (MUA) responses to full-field bright light steps applied to the contralateral eye of anaesthetised Opn1mw^R mice (Fig. 1a and Supplementary Fig. S1a). Unsurprisingly, given the limited distribution of retinal projections to this region, many electrode sites lacked any significant light-evoked MUA changes (n = 1997/2176 sites from 68 recordings across 20 Opn1mw^R mice; Supplementary Table S1). Importantly, however, we found discrete clusters of channels displaying very robust light-evoked MUA, spanning ventral parts and extending latero-dorsally across the anterior LHA (Fig. 1a–c and Supplementary Fig. S1a–c).

A hallmark of SCN and other cells receiving ipRGC input is the ability to encode changes in ambient illumination through sustained increases in firing[26,28,29,31,32]. By contrast, LHA responses often decayed very rapidly (within ~1 s) and were accompanied by robust transient increases in firing on return to darkness (Fig. 1b and Supplementary Fig. S1b), suggesting LHA neurons may be less tuned to encoding irradiance. To assess this, in recordings where we detected significant light-evoked MUA, we isolated responses of individual cells and more extensively investigated their sensory properties. In total, we isolated n = 44 light-responsive LHA neurons (from 18 recordings in 18 mice). In the same recordings, we were also able to isolate many cells that lacked light responses, but these primarily came from electrode sites dorsal or ventral to those where we detected single or multiunit light responses (n = 86/104 cells; Supplementary Fig. S1c and Supplementary Table S1).

Consistent with our hypothesis, while a subset of visually responsive cells from these LHA recordings maintained elevated firing across the light step (n = 19 'sustained' cells with firing rate significantly above baseline even during last 500 ms), a greater proportion (n = 25/44) showed only transient increases in firing at light ON (and in most cases) light OFF (Fig. 1d). By contrast, in recordings from the SCN/peri-SCN using an identical stimulus, the great majority of cells exhibited the expected sustained responses (n = 39/49 neurons from 15 mice) with 'transient' cells encountered significantly less frequently (n = 10/49 neurons, Fig. 1e and Supplementary Fig. S1d).

In parallel experiments, we further confirmed that our findings held true also for LHA cells in wildtype (WT; C57) mice, via multi-electrode recordings targeting the visually responsive portions of the LHA (n = 18 recordings from 18 mice). Consistent with the above, across regions of the LHA where we detected multiunit light responses, we could readily isolate light-responsive neurons (n = 34/51 cells), almost half of which (n = 15/34) displayed transient rather than sustained responses to high-intensity contralateral light steps, equivalent to the proportion found in Opn1mw^R mice and significantly more than in recordings from the WT SCN using an identical stimulus[33] (Supplementary Fig. S1d).

In the WT LHA recordings, we further took the opportunity to evaluate whether our assessment of LHA visual responses might have been skewed by focusing on responses to stimuli applied solely to the contralateral eye. In fact, however, we found little response to ipsilateral stimulation (Supplementary Fig. S1e). Indeed, only two cells (1 sustained and 1 transient) showed any significant response to ipsilateral stimuli (in both cases substantially smaller than the response to contralateral visual stimuli). Moreover, across all LHA cells, responses to light steps applied to contralateral and bilateral stimuli were essentially indistinguishable.

In sum, our data reveal two substantial differences in the properties of visually responsive neurons of the LHA compared to the SCN; (1) a significantly greater propensity to show transient rather than the sustained responses (Supplementary Fig. S1d) and (2) an overwhelming bias towards contralateral driven responses (in stark contrast ~2/3 of SCN cells show ipsilateral sensitivity using similar stimuli[33]; Supplementary Fig. S1e). Analysis of response latencies further revealed that these were equivalent for sustained and transient LHA neurons and, on average, significantly faster than for either SCN or visual thalamic cells tested with equivalent stimuli (Supplementary Fig. S1f), consistent with a direct retinally-driven origin.

### Rod and cone inputs to lateral hypothalamic neurons

Examination of irradiance response relationships for short-wavelength light-evoked activity in both Opn1mw^R (Fig. 1f–i) and WT mouse LHA neurons (Supplementary Fig. S1g–j) revealed a further overt difference relative to responses of SCN cells[28,33,34]. LHA neurons (particularly transient cells) frequently responded at intensities several orders of magnitude lower than the threshold for SCN responses (~$10^{11}$ photons/cm²/s).

To gain more insight into the underlying photoreceptive mechanisms, we focussed on our Opn1mw^R recordings, where we compared responses of LHA neurons to L-cone isoluminant, 460 nm and 655 nm light steps presented over 7-log spaced irradiances (Fig. 1f, h). We first examined the activity of sustained LHA cells, whose sensitivity more closely aligned to that previously reported for SCN neurons. Whereas we previously found the sensitivity of initial acute ON excitation among Opn1mw^R SCN cells to 460 nm vs. 655 nm light fully matched that expected for L-cone opsin[28], here the acute ON responses of LHA sustained cells could not be entirely explained by L-cone opsin sensitivity (Fig. 1g). Rather, irradiance response curves diverged at lower intensities, with larger responses to 460 nm, consistent with a rod influence on at least some of these cells. Accordingly, examination of threshold for detectable ON responses (Supplementary Fig. S2a, b) revealed a subset of LHA sustained cells (n = 5) whose relative sensitivity to 460 nm vs. 655 nm light aligned with that expected for rods (≥ 2 log units more sensitive to 460 nm) or which showed robust responses even to the dimmest 460 nm and 655 nm stimuli (below range where cone responses would be expected; 8.4 log cone effective photons/cm²/s).

Evidence for rod inputs was even more pronounced in the acute ON and OFF responses of LHA transient cells (Fig. 1h, i). Hence, at the population level, irradiance response curves for acute ON responses to 460 nm vs 655 nm light diverged substantially at lower intensities

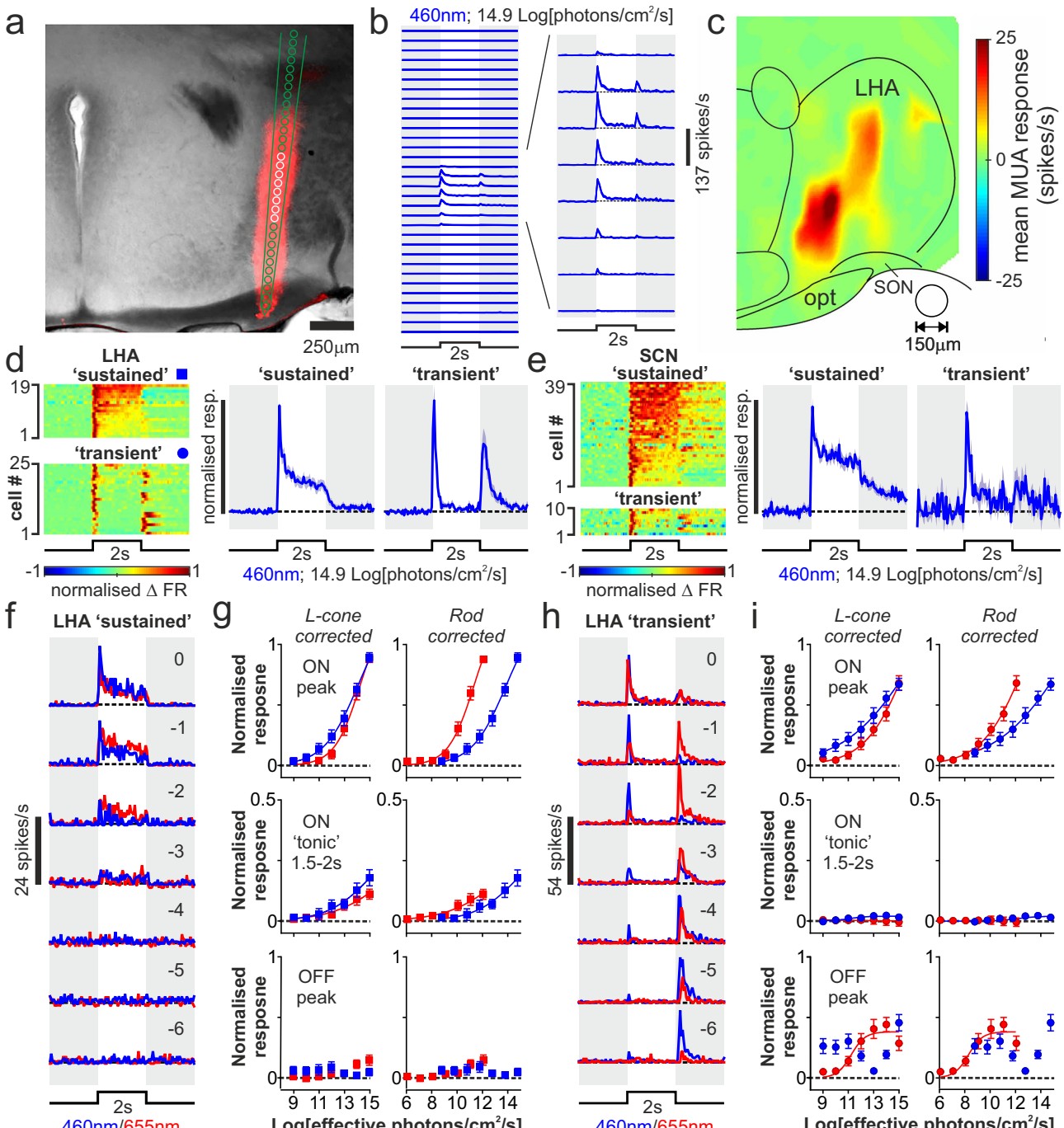

**Fig. 1 | Rapid and sensitive light-evoked activity across the mouse lateral hypothalamic area. a** Histological image from *Opn1mw^R* mouse showing DiI labelled probe track spanning the LHA, with superimposed schematic of the 32-channel electrode geometry (channels with significant light-evoked activity shown in white). **b** Mean multiunit activity (MUA, average of 20 trials) across the probe placement in (**a**), following 2 s light steps (460 nm; 14.9 log photons/cm²/s) applied to the contralateral eye. **c** Heat map showing mean MUA response within 250 ms following the appearance the contralateral 460 nm light step (*n* = 2176 electrode sites from 68 probe placements in 20 *Opn1mw^R* mice; 150 μm diameter circular window binning). **d, e** Left: normalised change in firing for isolated LHA (**d**) and SCN/peri-SCN (**e**) neurons classified as sustained or transient (cells with or without significantly elevated firing during last 500 ms of 2 s light step), Right: mean ± SEM response profiles across the populations of sustained and transient cells (*n* = 19 and 35 and *n* = 25 and 10 cells, from 18 and 15 *Opn1mw^R* mice respectively). **f, h** Mean

firing activity for representative sustained (**f**) and transient (**h**) LHA neurons following 2 s, L-cone isoluminant, 460 nm and 655 nm light steps with intensities spanning 7 log units (numbers indicate log intensity relative to unattenuated values of 14.9 and 15.4 log photons/cm²/s, respectively). **g, i** Mean ± SEM normalised responses of sustained (**g**, *n* = 19) and transient (**i**, *n* = 25) LHA neurons to 460 nm and 655 nm light steps as a function of irradiance for L-cone opsin (left) or rhodopsin (right). Panels top to bottom respectively show peak change in firing (50 ms bin) and mean across the last 500 ms of light ON ('tonic') and peak following light OFF (50 ms bin as above). Data fit by 4-parameter sigmoid curves, where appropriate fit parameters compared between 460 nm and 655 nm by two-tailed F-test (**g** left, ON peak: $F_{2,261}$ = 5.2, *P* = 0.006, 'tonic': $F_{2,261}$ = 4.0, *P* = 0.02; **g** right, ON peak: $F_{2,261}$ = 110.1, *P* < 0.0001, 'tonic': $F_{2,261}$ = 5.6, *P* = 0.004; **i** left, ON peak: $F_{2,345}$ = 8.0, *P* < 0.0001, OFF peak: $F_{2,345}$ = 11.4, *P* < 0.0001; **i** right, ON peak: $F_{2,345}$ = 21.7, *P* < 0.0001, OFF peak: $F_{2,345}$ = 6.6, *P* = 0.002).

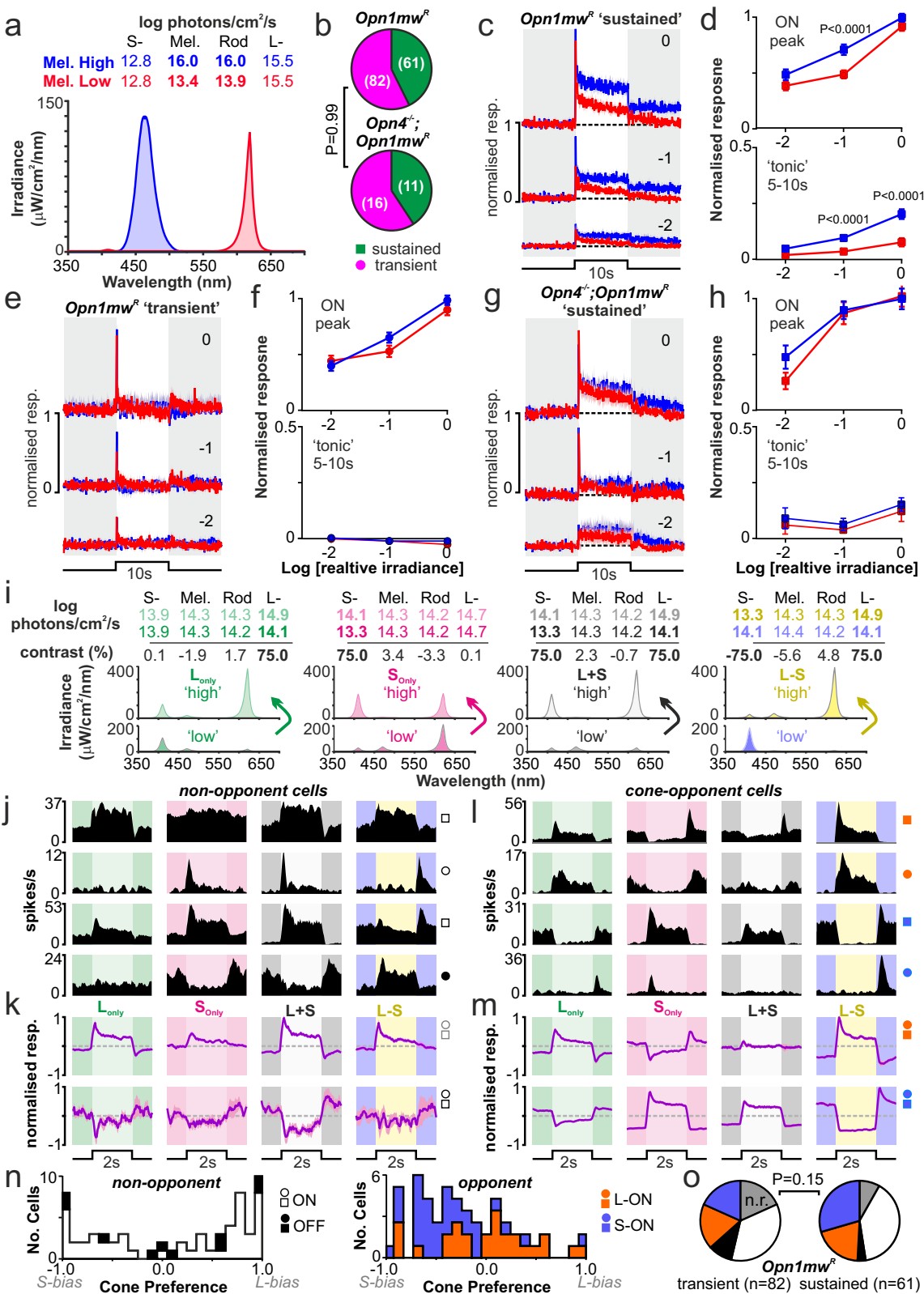

when irradiance was expressed in L-cone effective photons and diverged at higher intensities when irradiance was expressed in rod-effective photons (Fig. 1i). Thus, population level responses appeared to involve a combination of rod and L-cone driven components. Subsequent individual cell analyses confirmed that the threshold for acute ON responses was best explained by a rod influence for most neurons ($n = 18/25$; Supplementary Fig. S2a, b). Acute OFF responses of LHA

transient cells were more complex, with a biphasic response to 460 nm light (robust response at low and high intensities but weaker responses at intermediate intensities) and monophasic responses to increasing irradiances of 655 nm (Fig. 1h, i). This presumably reflects a separation of rod and cone-driven OFF responses at low vs. high irradiances of 460 nm, since, when irradiances were corrected in terms of rod effective photons, dimmer 460 nm light responses broadly aligned

**Fig. 2 | Melanopsin-driven sustained activity and cone-opponent responses across subsets of mouse lateral hypothalamic neurons. a** Spectra of S and L-cone isoluminant stimuli that differentially activate melanopsin (Mel. High vs. Mel. Low). **b** Proportions of LHA neurons showing sustained or transient responses to the Mel. High stimulus ($n = 143$ and $n = 27$ cells from 9 $Opn1mw^R$ and 5 $Opn4^{-/-};Opn1mw^R$ mice respectively; analysis by Fisher's exact test, two-sided, $P = 0.99$). **c, e, g** Mean ± SEM normalised responses of $Opn1mw^R$ sustained (**c**, $n = 61$), $Opn1mw^R$ transient (**e**, $n = 82$) and $Opn4^{-/-};Opn1mw^R$ sustained (**g**, $n = 11$) to 10 s Mel. High and Mel. Low light steps across 3 irradiances (numbers indicate log intensity relative that shown in **a**). **d, f, h** Mean ± SEM normalised firing response for cells in (**c, e** and **g**)across the first 250 ms and last 5 s of the 10 s Mel. High and Low light steps (upper and lower panels, respectively). Data analysed by RM 2-way ANOVA with Sidak's post-tests (**d** upper: Irrad.-$F_{2, 120} = 64.2$, $P < 0.0001$, Stim.-$F_{1, 60} = 15.5$, $P = 0.05$, Irrad.XStim.-$F_{2, 120} = 2.8$, $P = 0.06$; d lower: Irrad.-$F_{2, 120} = 37.7$, $P < 0.0001$, Stim.-$F_{1, 60} = 32.4$, $P < 0.0001$, Irrad.XStim.-$F_{2, 120} = 14.1$, $P < 0.0001$; f upper: Irrad.-$F_{2, 162} = 57.4$, $P < 0.0001$, Stim.-$F_{1, 81} = 2.3$, $P = 0.13$, Irrad.XStim.-$F_{2, 162} = 2.8$, $P = 0.07$; f lower: Irrad.-$F_{2, 162} = 0.8$, $P = 0.43$, Stim.-$F_{1, 81} = 0.8$, $P = 0.36$, Irrad.XStim.-$F_{2, 162} = 1.0$, $P = 0.38$; h upper: Irrad.-$F_{2, 20} = 15.1$, $P < 0.0001$, Stim.-$F_{1, 10} = 3.5$, $P = 0.09$, Irrad.X-Stim.-$F_{2, 20} = 2.0$, $P = 0.17$; h lower: Irrad.-$F_{2, 20} = 1.9$, $P = 0.17$, Stim.-$F_{1, 10} = 5.0$, $P = 0.05$, Irrad.XStim.-$F_{2, 20} = 0.04$, $P = 0.97$). **i** Spectra and quantification of stimuli providing selective contrast for L- and S-cone opsin in isolation, unison or antiphase (left to right, respectively). **j, l** Peristimulus histograms for four representative non-opponent (**j**) and opponent (**l**) $Opn1mw^R$ LHA neurons to full-field cycles of cone-isolating stimuli in (**i**) (means of 25 trials). **k, m** Mean ± SEM normalised responses of ON and OFF non-opponent neurons (**k**, $n = 53$ and $n = 10$, upper and lower respectively) and L-ON and S-ON opponent neurons (**m**, $n = 27$ and $n = 33$ upper and lower respectively) to stimuli in (**a**). **n** Distribution of cone opsin preference (see "Methods") for non-opponent (left) and opponent (right) cells in (**j–m**). **o** Cone responses of $Opn1mw^R$ LHA neurons classified as transient (left) or sustained (right) from analysis in (**j–n**), compared by two-tailed $\chi^2$-test ($\chi^2 = 6.7$, df = 4, $P = 0.15$); n.r. indicates no response.

---

with those produced by 655 nm of equivalent intensity for rods. Here again, individual neuron analyses confirmed that the response threshold for acute OFF responses, where present, was best explained by a rod influence for most neurons ($n = 15/23$; Supplementary Fig. S2c).

### Melanopsin-driven responses in lateral hypothalamic neurons

Based on neuroanatomical tracing studies[1–4] and our previous evaluations of visual responses in other brain regions[26,28,29,31], we reasoned that the populations of LHA cells with sustained vs. transient responses likely reflected cells that received input from ipRGCs vs. other RGC types. Accordingly, we found 'tonic' components of the $Opn1mw^R$ LHA sustained cells responses to cone-isoluminant 460 vs. 655 nm light (last 500 ms of light step) diverged at higher irradiances, with larger responses to the 460 nm stimuli that are more able to engage melanopsin phototransduction (Fig. 1g). Focusing of the highest irradiances tested, where wavelengths both should be sufficiently bright to evoke a transiently saturating rod response, while only the 460 nm stimulus should be sufficiently bright to activate melanopsin[28,35], we found significantly increased firing for 460 nm vs 655 nm among sustained (but nor transient) cells of both the LHA and SCN (Supplementary Fig. S2d, e). By contrast, the initial ON peak excitations, which (given the comparatively sluggish kinetics of melanopsin[36]) are expected to be rod/cone dominated, were equivalent between wavelengths for both sustained and transient cells in both the LHA and SCN (Supplementary Fig. S2d, e).

While these data are consistent with the view that sustained but not transient cells receive ipRGC input, it is formally possible that the relatively brief stimulus duration used in those experiments (2 s) was insufficient, or the light intensity too low, to reveal a melanopsin component in all cells. Conversely, since the intensity of 460 nm light was, in principle, bright enough to engage S-cones (11.7 log S-cone effective photons/cm²/s), wavelength-dependent differences observed in responses of sustained cells might not originate with melanopsin. We next, then, aimed to more definitely confirm melanopsin influences on LHA responses. Accordingly, in subsequent experiments, we targeted the visually responsive regions of the LHA with high density 32-site polytrodes and compared responses to longer (10 s), bright, 460 nm light steps ('Mel. High') with a polychromatic (620 nm + dim 410 nm) stimulus matched to be isoluminant for both S- and L-cone opsin but providing much weaker activation of melanopsin ('Mel. Low; Fig. 2a). We then tested responses to these stimuli at 3 log-spaced irradiances where even the Mel. Low stimulus would be expected to maximally activate rods[29,37–39].

From polytrode recordings from the $Opn1mw^R$ LHA ($n = 9$), we identified $n = 143$ neurons with significant light-evoked changes in firing. Consistent with the data above, while a subset ($n = 61/143$) displayed robustly sustained responses to the Mel. Highlight steps, a greater proportion lacked this feature and instead displayed transient responses to light onset and/or offset ($n = 82/143$; Fig. 2b). Moreover, consistent with our prediction that the sustained population received input from ipRGCs, this group of cells reliably displayed significantly greater tonic responses to the Mel. High vs. Mel Low stimulus across the irradiance range tested (Fig. 2c, d). This observation did not reflect some unintended difference in rod/cone activation between the Mel. High and Low stimuli since the peak ON excitation of these sustained cells was not significantly different between these stimuli at maximal irradiance, nor were either the ON peak or tonic responses in transient cells (Fig. 2e, f).

To further confirm that our observations reflected a genuine contribution of melanopsin, we recorded from the LHA of melanopsin knockout animals ($Opn4^{-/-}; Opn1mw^R$, $n = 5$). From $n = 27$ light-responsive LHA neurons detected here, we again found we could readily separate cells into sustained and (a more numerous) transient population ($n = 11$ and $n = 16$, respectively, Fig. 2b). Hence, melanopsin is not absolutely required for sustained responses in LHA neurons. Importantly, however, there were no detectable differences in either the ON peak or tonic components of the cells' responses to the Mel. High vs. Low stimuli for both transient (Supplementary Fig. S2f) or sustained cells in the LHA of $Opn4^{-/-}; Opn1mw^R$ mice (Fig. 2g, h).

### Chromatic preference and colour opponency among lateral hypothalamic neurons

Having estbalished rod, L-cone and melanopsin inputs to subsets of LHA neurons, we next examined S-cone inputs and the possibility of colour opponency, as detected in other subcortical visual nuclei in the mouse[26,29,30,40,41]. Accordingly, we employed validated polychromatic stimuli designed to selectively manipulate S- and L cone opsin excitation singly, in unison, or in antiphase for $Opn1mw^R$ mice[29,30]. Stimuli were presented as square wave (0.25 Hz) full-field modulations around a common background with increasing contrasts up to 75% Michelson for the targeted opsins (Fig. 2i).

Across the light-responsive LHA neurons detected in our polytrode recordings, the vast majority ($n = 123/143$) exhibited detectable responses to our cone-isolating stimuli. Consistent with our findings in other brain regions[26,29,30,40,41], when tested with single cone-opsin modulations (S$_{Only}$ and L$_{Only}$) many of those responsive cells ($n = 63/123$) were strongly biased towards one of the two cone-opsin classes and, where responses driven by both cone types were detectable, these were of the same sign (usually 'ON', $n = 53/63$; Fig. 2j, k, n, o and Supplementary Fig. S2i). As a result of this bias, for these achromatic ON and OFF cells, responses to stimuli that modulated both cone types in unison (L + S) or antiphase (L-S) typically produced near-identical responses (albeit phase inverted for S-biased neurons; Fig. 2j, k and Supplementary Fig. S2g).

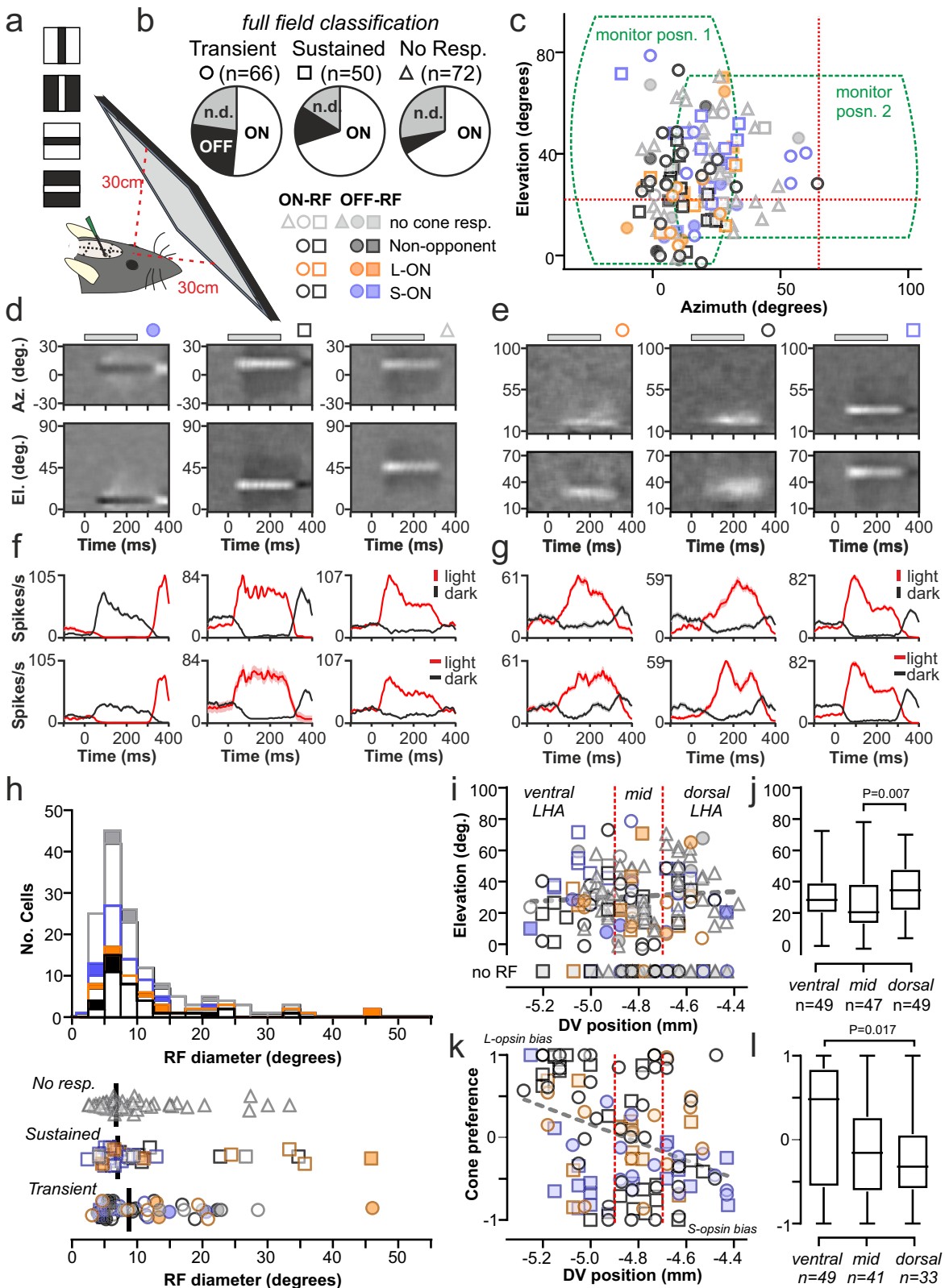

By contrast, an almost equal proportion of LHA cells ($n = 60/123$) exhibited robust colour opponent responses (Fig. 2l–o and Supplementary Fig. S2i). Such cells either displayed excitatory/ON responses to $L_{Only}$ and inhibitory/OFF responses to $S_{Only}$ stimuli ($n = 27$ L-ON opponent neurons) or the converse ($n = 33$ S-ON opponent neurons). Opponent cells typically showed some bias towards the opsin responsible for the ON component of their response, such that they

remained capable of responding to achromatic ($L + S$) modulating stimuli. Nonetheless, these opponent cells reliably displayed much larger changes in firing when presented with chromatic (L-S) contrast, as expected for colour opponent cells (Fig. 2l–n and Supplementary Fig. S2h). The occurrence of these opponent vs. non-opponent cone-driven LHA responses was equivalent across cells showing sustained or transient responses to light steps from darkness (Fig. 2o). Hence, a

**Fig. 3 | Lateral hypothalamic neurons receive visual input from discrete regions of visual space. a** Setup for mapping LHA neuron receptive field (RF) positions, using flashing bar stimuli. **b** RF classification across cells with sustained, transient or non-responsive to full field light steps ($n = 188$ neurons from 5 $Opn1mw^R$ mice; n.d. = no detectable RF). **c** RF centres for ON ($n = 177$) and OFF ($n = 28$) LHA neurons; symbols and colour coding respectively indicate response to full field light steps and cone-directed stimuli. Dotted lines show the spatial extent of the visual stimulus (tested sequentially at the two different locations) and the projected dorsal-ventral and nasal-temporal midpoints of the retina. **d, e** RFs for three representative neurons in frontal (**d**) or lateral visual space (**e**). Plots show normalised firing around the appearance of light minus dark bars as a function of bar position on the azimuth (upper) or elevation plane (lower panels). **f, g** Mean ± SEM responses to light and dark vertical (upper) or horizontal (lower) bars within the RF centre for representative neurons in (**e, f**). **h** RF diameters for LHA neurons with ON ($n = 117$) or OFF ($n = 28$) RFs as a function of cell classification based on full light steps and cone-directed stimuli. **i** Relationship between RF centre elevation and LHA anatomical location on the dorsal-ventral plane. **j** Distribution of RF centre elevation located in ventral, mid or dorsal portions of the LHA, as indicated by dotted lines in (**l**) (box = quartiles 2-3; centre = median; whiskers = min to max). **k** Relationship between cone preference and dorsal-ventral anatomical position within the LHA for $n = 124$ cells with responses to full field cone isolating from 9 Opn1mwR mice. **l** Cone preferences for cells located in ventral, mid or dorsal LHA, as indicated by dotted lines in (**k**) (box = quartiles 2-3; centre = median; whiskers = min to max). Data analysed by one-way ANOVA (**j**: $F_{2,142} = 4.75$, $P = 0.01$; **l**: $F_{2,120} = 4.62$, $P = 0.01$) with Tukey's post-tests.

high proportion of cells that did or did not show evidence of ipRGC input exhibited colour opponency, under the conditions studied here.

## Spatial contrast tuning across lateral hypothalamic neurons

We next sought to better understand how such light-responsive neurons sampled across visual space. Accordingly, in a subset of the $Opn1mw^R$ animals used above ($n = 5$) we went on to assess spatial RF properties by presenting flashing light or dark, horizontal or vertical, bars at varying locations via a conventional visual monitor either positioned directly in front of the animal of moved laterally to cover substantial portions of the central and upper visual field (Fig. 3a–c).

A high proportion of light-responsive LHA neurons ($n = 93/116$ cells responding to full-field light steps) displayed robust and repeatable responses to the appearance of bars in discrete regions of visual space (Fig. 3b–g and Supplementary Fig. S3a–d). Strikingly, many of the cells that lacked detectable full-field light responses also displayed readily detectable RFs ($n = 52/72$; Fig. 3b–f and Supplementary Fig. S3a–d), indicative of strong selectivity for spatial contrast. Accordingly, clear antagonistic surrounds were apparent in the RFs of many cells (both those that displayed or lacked responses to full-field light steps; Fig. 3d, e and Supplementary Fig. S3a, b), consistent with the view that many cells preferentially detect spatial contrast.

A majority of LHA cells displayed ON- rather than OFF-centre (or no) RFs, regardless of the cells' response to full-field light steps or cone-directed stimuli (Fig. 3b, c). This included cell populations that lacked full field light responses and those with colour-opponent responses for which the (achromatic) stimuli used here might be suboptimal. In fact, consistent with our experience using similar approaches in the visual thalamus[30], RFs were readily identifiable for many colour-opponent cells ($n = 44/55$), in some cases presenting as ON and other cases OFF responses ($n = 32$ and $n = 13$ cells respectively), presumably because the flashing bar stimuli preferentially revealed one component of the cells cone-driven RF. Excluding these colour opponent neurons, OFF-centre RFs were most common among transient cells ($n = 9/30$ cells with mappable RFs) rather than sustained cells or cells that lacked responses to full field light steps ($n = 2/18$ and $4/52$ cells respectively).

Response latencies for LHA cells with mappable RFs did not systematically differ between those with sustained, transient or no response to full field light steps (Supplementary Fig. S3e) while, at the population level, LHA responses were modestly but significantly faster than observed for visual thalamic cells tested with the same stimuli[30] (Supplementary Fig. S3f). Consistent with our analysis of sustained and transient cell responses to full-field stimuli, above, these observations support a direct retinally driven origin for the LHA responses observed here. Further, estimates of RF sizes provided by our flashing bar stimuli revealed that LHA neurons typically displayed small RF-centres (Fig. 3h; median diameter ± SD = 7.6 ± 8.3°), at the lower limit of that typically observed in recordings from the visual thalamus[30,42–46] and strikingly different from RFs previously observed for mouse SCN neurons[25], where many cells respond to visual signals across wide

regions of visual space. Here we only saw occasional cells with relatively large RFs ($n = 18/145$ cells with RF diameters > 20°) and no evidence that RF size varied systematically as a function cell response type as revealed by full field stimuli (Fig. 3h).

## Retinotopic order across lateral hypothalamic visual responses

Our data above reveal a high proportion of LHA cells are strongly selective for visual inputs coming from a very discrete region of visual space. Significantly, we also found that RF-centre locations were clustered to a specific region of the visual field, corresponding to temporal portions of the retina (~directly in fount of the animal) and between ± 30° relative to the projected dorsal-ventral midpoint (Fig. 3a). Given the near 180° field of view of the mouse eye and the extent of the visual field covered by our monitor, it is possible that cells that did not respond to our stimuli had RFs in other visual locations. Nonetheless, it is notable that, overall, we could map an RF for ~ 77% of the cells we recorded ($n = 145/188$), despite the fact our visual stimuli only covered ~ 33% of the full visual field. Hence, our data indicate a non-random distribution of visual input to the LHA, with a sizable portion coming from central and ventral portions of the temporal retina.

We further, then, asked whether the precise region a cell received input from might vary anatomically across the retinorecipient LHA, as it does for conventional visual regions like the lateral geniculate nucleus (LGN)[42,45]. The nature of our recordings (which specifically targeted the anterior, retinorecipient[2–4,15,16], portion of the LHA) precluded a detailed analysis of how RF position changes as a function of location on the rostral-caudal axis. Nonetheless, we did not find any clear evidence that RF azimuth or elevation varied between cells recorded at more rostral or caudal locations (Fig. S3g). Further, RF azimuth position did not vary systematically across dorsal-ventral and medial-lateral positions in the LHA (Supplementary Fig. S3h–k), nor did RF elevation across medial-lateral LHA (Supplementary Fig. S3j, k). We did, however, find a tendency where higher elevations were preferentially represented by more dorsally located LHA cells (Fig. 3i, j). Whilst this anatomical variation was less pronounced than for more conventional visual nuclei, these data reveal some degree of spatial segregation of inputs from dorsal vs. ventral retina within the LHA.

Given those data, we next asked whether the incomplete coverage of the full mouse visual field afforded by our spatial stimuli may have skewed our assessment of LHA retinotopy (by under-representing cells at more extreme eccentricities). To assess this, we utilised the known gradient in retinal cone opsin co-expression[47–50] as an additional method of localising the retinal origins of LHA responses. We 1st confirmed that we could find the expected bias towards S-opsin driven responses in higher and more lateral RFs (Supplementary Fig. S3l, m) among the subset of cells ($n = 83$) for which we had both full-field cone responses and mapped RFs. Having validated the approach, we interrogated the relationship between S- vs. L-cone opsin bias and anatomical location for the wider population of LHA neurons, where we determined cone preference ($n = 124$ cells from 9 $Opn1mw^R$ mice,

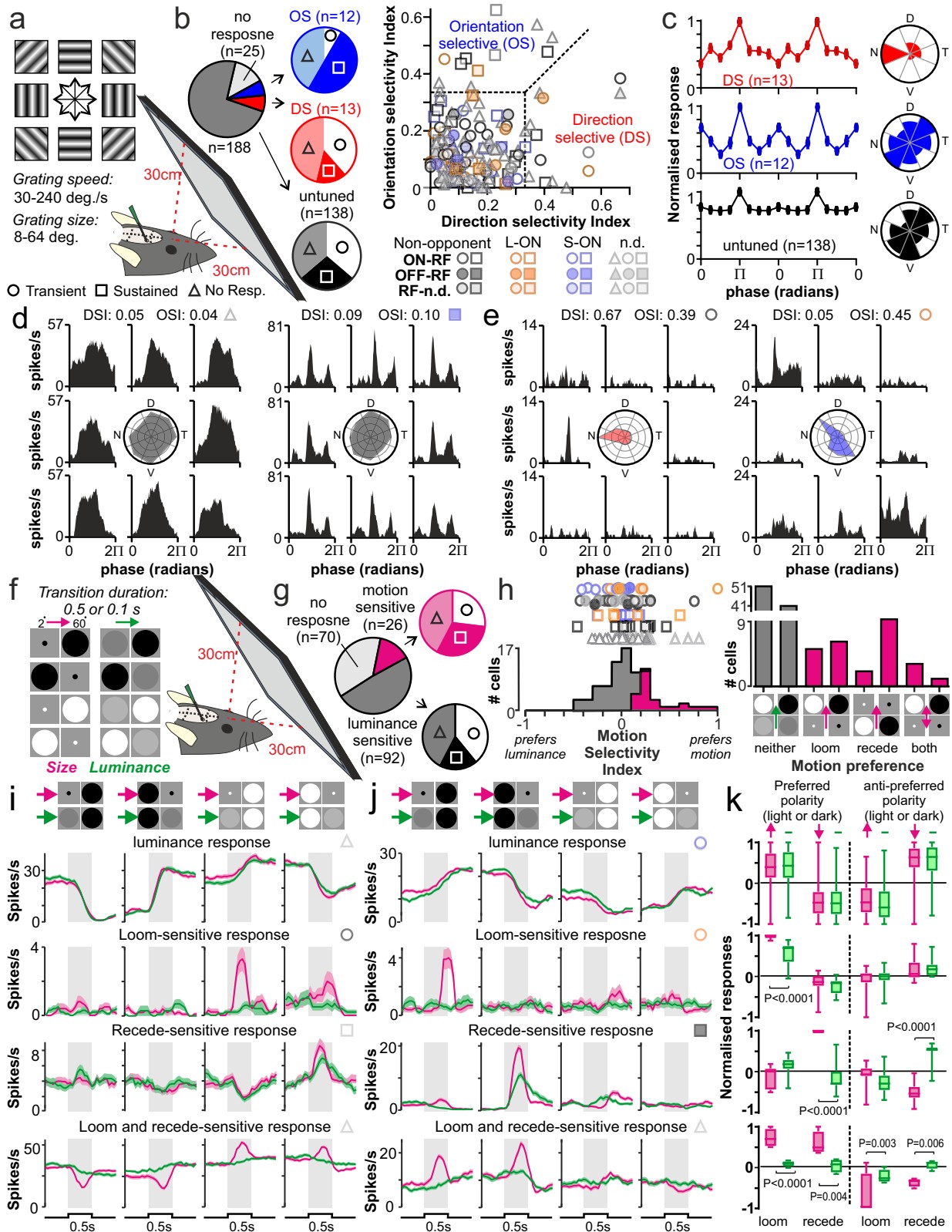

including those where we could not detect or did not test for an RF). We found that cells in the ventral LHA were indeed more likely to show strong L-cone opsin bias and cells in the dorsal LHA stronger S-cone opsin bias (Fig. 3k, l), but no significant variations in preference as a function of medial-lateral position (Supplementary Fig. S3m, o). Collectively, then, these data are consistent with a non-random arrangement of visual inputs to the LHA, with ventral regions preferentially receiving input from (L-opsin biased) dorsal retina and dorsal LHA regions from (S-opsin biased) ventral retina.

## Motion sensitivity and spatiotemporal contrast tuning in lateral hypothalamic neurons

Prompted by the strong spatial contrast sensitivity apparent in LHA visual responses and a previous report that retinal neurons expressing

**Fig. 4 | Motion selective responses in lateral hypothalamic neurons. a** Setup for assaying LHA neuron spatiotemporal tuning using drifting gratings. **b** Left: proportions of *Opn1mw*[R] LHA cells (from *n* = 188 cells tested) with direction or orientation selective (DS/OS) responses or responses untuned according to grating angle, Right: distribution of direction vs. orientation selectivity indices (DSI/OSI; see methods) for all cells with significant responses to drifting grating stimuli (*n* = 163/188 cells form 5 mice, tested at two monitor positions as per Fig. 3c). **c** Mean ± SEM normalised response of DS (top), OS (mid), and untuned cells (bottom) at optimal spatiotemporal frequency as a function of grating angle relative preferred angle (π radians). Inset plots show the distribution of preferred directions/orientations. **d, e** Responses of representative untuned (**d**) and DS/OS (**e**) cells to gratings of optimal spatiotemporal frequency across the 8 tested directions. Central plots show normalised response as a function of grating direction. **f** Setup for assaying LHA neuronal responses to approaching and receding motion. **g** Proportions of *Opn1mw*[R] LHA neurons (from *n* = 188 cells tested) responding to stimuli in (**f**) with or without a significant preference to motion stimuli. **h** Left:

distribution of motion selectivity index (see "Methods"), Right: preferred stimulus type for all responding *Opn1mw*[R] LHA neurons (*n* = 115/188 cells tested). **i, j** mean + SEM firing rate (over 20 trials) of representative cells with (top to bottom, respectively) ON (**i**) or OFF (**j**) luminance responses or preferential responses to approaching, receding or both approaching or receding motion for light (**i**) or dark spots (**j**). **k** Mean + SEM normalised population responses (box = quartiles 2-3; centre=median; whiskers=min to max) for cells exhibiting each of the four response classes in **i** and **j**, aggregated by plotting as a function of responses to stimuli of preferred vs. non-preferred polarity (light or dark) and preferred or opposite direction of motion (alongside associated control stimuli). Data analysed by 2-way RM ANOVA with Sidak's post-tests (**k** luminance: Stim.-$F_{3,273}$ = 125.9, $P < 0.0001$, Motion-$F_{1,91}$ = 1.13, $P = 0.29$, Stim.XMotion-$F_{3,273}$ = 4.35, P = 0.005; **k** loom: Stim.-$F_{3,30}$ = 32.1, $P < 0.0001$, Motion-$F_{1,10}$ = 8.93, $P = 0.01$, Stim.XMotion-$F_{3,30}$ = 14.47, $P < 0.0001$; **k** recede: Stim.-$F_{3,30}$ = 18.82, $P < 0.0001$, Motion-$F_{1,10}$ = 0.08, $P = 0.80$, Stim.XMotion-$F_{3,30}$ = 41.66, $P < 0.0001$; **k** loom and recede: Stim.-$F_{3,9}$ = 18.79, $P < 0.0001$, Motion-$F_{1,3}$ = 0.29, $P = 0.63$, Stim.XMotion-$F_{3,9}$ = 38.69, $P < 0.0001$).

markers of direction-selective (ds)RGCs might project to the LHA[24], we next assessed spatiotemporal tuning properties in more detail and asked whether any LHA neurons might selectively respond to visual motion. To this end, we first assessed responses to drifting sinusoidal gratings of varying speed, size and direction of motion (Fig. 4a) across the two spatial locations used for RF mapping.

Unsurprisingly, the overwhelming majority of cells where we could map an RF also displayed significant modulations in firing in responses to drifting gratings (*n* = 143/145 cells; Fig. 4b–e and Supplementary Fig. S4a–f). We also, however, found many cells where we could not map an RF but nonetheless detected robust and reproducible responses to drifting gratings (*n* = 20/43 cells). Remarkably, then, almost all of the cells we recorded from visually responsive regions of the LHA could track drifting gratings, including most of the cells that lacked responses to full-field light stimuli (*n* = 61/72; 30% of the 188 total LHA cells tested in these experiments).

For each responding cell, we next asked whether the response varied as a function of grating direction or orientation, using established indices[45]. While the majority of cells responded similarly to drifting gratings across all tested directions of motion (*n* = 138, Fig. 4b–d and Supplementary Fig. S4a), a subset of cells showed pronounced direction or orientation preference (*n* = 13 and *n* = 12, respectively, Fig. 4b, c, e and Supplementary Fig. S4b). Cells in the orientation-selective (OS) group varied in their preferred orientations, while cells in the direction-selective (DS) group most commonly displayed a preference for nasal motion (Fig. 4c). We also noted that OS and DS neurons tended to differ in their responses to full field light steps; with transient responses under-represented among OS cells and sustained responses under-represented among the DS cells (Fig. 4b).

We next evaluated whether preference for grating size or speed varied across the various subgroups of neurons identified in these experiments. For the great majority of LHA cells (regardless of response type), responses were most robust for the coarsest and slowest moving gratings (64°, 30°/s; Supplementary Fig. S4c–j). There was more substantial variation, however, in the minimal sizes and fastest grating speeds that individual neurons could follow. While these parameters did not vary systematically across neurons categorised based on their responses to full-field stimuli (Fig. S4g–j), both of those parameters differed significantly between DS/OS cells and cells that were indifferent to grating angle (Supplementary Fig. S4k–n). Hence, whereas cells in the latter group generally exhibited relatively high spatial and temporal acuity (following gratings of 16° or less and speeds ≥120°/s), most DS/OS cells displayed lower acuity in both dimensions (Supplementary Fig. S4k–n).

Given our finding above and previous data suggesting the LHA could modulate responses to threatening stimuli (including visual cues[21,51]), we finally assessed responses expanding and contracting light and dark disks stimulating approaching ('looming') and receding

motion (Fig. 4f). Spots changed size between 2 and 60° diameter over 0.5 or 0.1 s, centrally within the visual display at the two different locations used for RF mapping (see Fig. 3a) and were paired with control stimuli (static 60° diameter disks) that provided matched changes in luminance to allow us to identify neurons that were preferentially responsive to visual motion.

Since the regions of visual space encompassed by spots in these experiments was less than the full spatial extent of our display, somewhat fewer LHA neurons responded to this stimulus set than others we tested (Fig. 4g). Nonetheless, we identified robust and reproducible responses among many of the cells where we could map an RF (*n* = 109/145) as well as small number of cells where we could not (*n* = 9). Whilst we therefore found a sizeable number of cells that responded to expanding and/or contracting spots, in most cases such cells displayed equivalent responses to matched luminance control stimuli (either excited by increases or decreases in luminance) as expected for cells with simple, motion-insensitive, RFs intersecting the locations of stimulus delivery (*n* = 92/118; Fig. 4h–k). We did, however, find a subset of neurons that displayed a significant preference for either light or dark spots and one or both types of motion signals; most commonly expansion/looming or contraction/receding motion, but also occasionally both directions of composite motion (Fig. 4h–k).

Taking an integrated view of these data, in our recordings targeting visually responsive regions of the LHA, almost every cell detected responded to at least some aspects of the full stimulus set (*n* = 178/188 cells tested with stimuli used in Figs. 2–4). More significantly, these diverse stimuli reveal multiple subpopulations of cells with different sensory tuning properties (Fig. 5), indicating that the proposed role for retinohypothalamic projections in tracking variations in ambient light intensity can only constitute a small subset of the functions that visual portions of the LHA subserve.

Indeed, while the functional groupings of LHA cells identified here includes subsets of cells that simply report local 'luminance' or contrast (including most sustained cells and subpopulations of transient cells with ON or OFF biased responses; Fig. 5, Groups I-III), only ~ 22% of visually responsive LHA cells fall into the 'luminance-coding' group that one might expect to dominate in the hypothalamus (*n* = 39/178 visually responsive cells). By contrast, we identify a nominally greater number of cells that are, in fact, highly selective for spatial contrast and do not respond at all to full-field stimuli (*n* = 45/178, ~ 25% of visually responsive cells; Fig. 5, Group IV) as well two groups of cells tuned to aspects of visual motion (collectively *n* = 39/178, > 20% of visually responsive cells; Fig. 5, Groups V-VI). The latter, motion-sensitive, subpopulations include cells that preferentially respond to specific directions of motion (*n* = 16 DS/OS cells with DSI > 0.33) as well as cells (*n* = 23) that lack conventional direction selectivity but are sensitive to complex motion as revealed by preferential responses to approaching

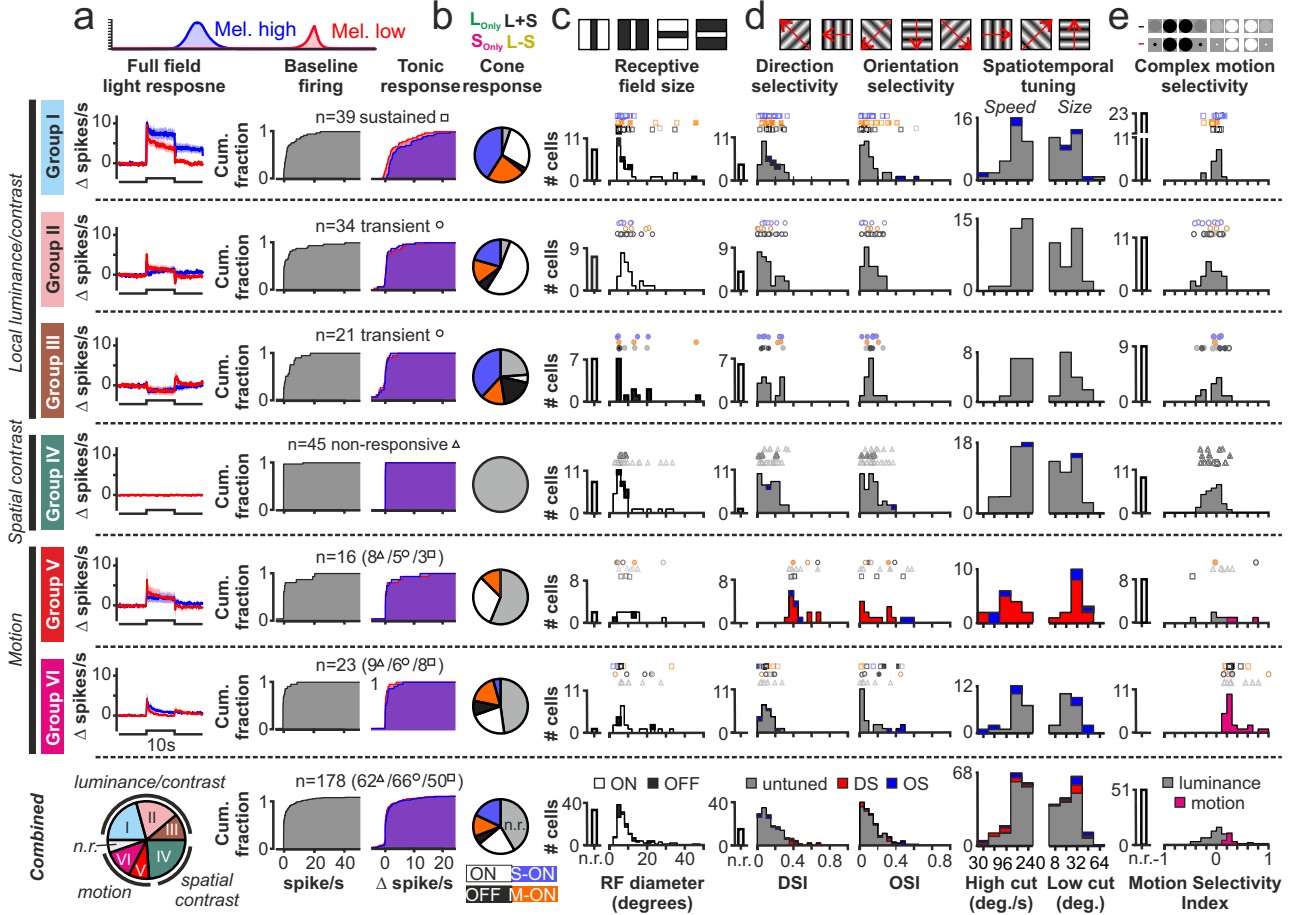

**Fig. 5 | Diversity of sensory tuning properties across LHA neurons.**
**a**–**e** Identification of multiple subgroups of LHA neurons based on sensory tuning properties revealed by full-field and spatially structured stimuli reported in Figs. 2–4 (from n = 188 LHA neurons recorded in 5 *Opn1mw^R* mice). Rows I-VI represent different cell classes (see "Methods" for details of cell classification), bottom row shows pooled population data. **a** Left panels show the mean ± SEM response of each population to Mel High. and Mel. Low, full-field, light steps (as per Fig. 2a), the middle panel shows the cumulative distribution of baseline (dark-adapted) firing rates, right panels show cumulative distributions of change in firing rates 5–10 s after the light steps. **b** Distribution of non-opponent (ON/OFF) and opponent (M-ON/S-ON) responses to full-field cone-directed stimuli (as per Fig. 2i). **c** Distribution of ON and OFF RF sizes across each population as determined by flashing bar stimuli (as per Fig. 3a). **d** Panels, left-right, show distributions of direction- and orientation-selectivity index (DSI/OSI), temporal high-cutoff frequency and spatial low-cutoff frequency determined using drifting grating stimuli (as per Fig. 4a). **e** Distribution of motion-selectivity index for each cell population as revealed by looming/receding disk stimuli (as per Fig. 4f).

and/or receding spots (Fig. 5d, e). Responses to full-field stimuli varied within these motion sensitive cells groups, although transient or no responses were substantially more common than sustained light responses, especially among the direction tuned group (Fig. 5a). We further noted that, unlike local luminance/contrast sensitive cell groups, the motion sensitive cells often failed to respond to our cone-directed stimuli, consistent with a low sensitivity to full field contrast, and almost never displayed evidence of S-ON colour opponency (Fig. 5b).

## Retinal inputs to the lateral hypothalamic area
To understand the origins of the diversity in LHA sensory properties, we performed retro-labelling from the LHA Ai32 mice[52], to drive Cre-dependent expression of their ChR2-EYFP reporter across the cell bodies and processes of cells projecting to the LHA. Since our electrophysiological data indicated the LHA was most unlikely to receive retinal inputs uniformly from across each retina, we started by uni-laterally injecting AAV2retro-hSyn-Cre into the LHA (n = 5 Ai32 mice) and assessing the distribution of EYFP+ neurons in contralateral retina wholemounts, alongside immunohistochemical detection of the gradient of S-cone opsin expression as a marker of retinal orientation (Fig. 6a, b and Supplementary Fig. S5a). In line with our RF mapping

studies, these experiments revealed a subpopulation of LHA-projecting neurons (n = 118 ± 50 EYFP + cells, mean ± SEM) that were non-uniformly distributed across the contralateral retina, with the highest density in the ventrotemporal and the lowest in the dorsonasal retinal quadrants (Fig. 6c). As expected, labelled neurons were much less numerous in the ipsilateral retinas (n = 10 ± 5 EYFP + cells) and entirely restricted to the ventrotemporal quadrant (Supplementary Fig. S5b–d).

Since our electrophysiological data indicated many visually responsive LHA neurons lacked evidence of ipRGC input, we next assessed the extent to which LHA-projecting RGCs displayed detectable melanopsin expression. Accordingly, in a subset of retinas from animals that received bi- or unilateral injections of AAVretro-hSyn-Cre into the LHA (n = 7 retinas from 5 mice), we co-labelled for melanopsin and LHA-projecting (EYFP +) RGCs (Fig. 6d, e and Supplementary Fig. S5e). Consistent with our electrophysiological assessments, while we could readily identify melanopsin expression among a sizable proportion of retro-labelled RGCs, almost half (42 ± 8%, mean ± SEM) of such cells lacked detectable melanopsin immunoreactivity. By contrast, when we injected the same AAV into the SCN, almost all retro-labelled RGCs were melanopsin-positive (93 ± 3%, n = 6 retinas from 3 Ai32 mice; Fig. 6f and Supplementary Fig. S5ij).

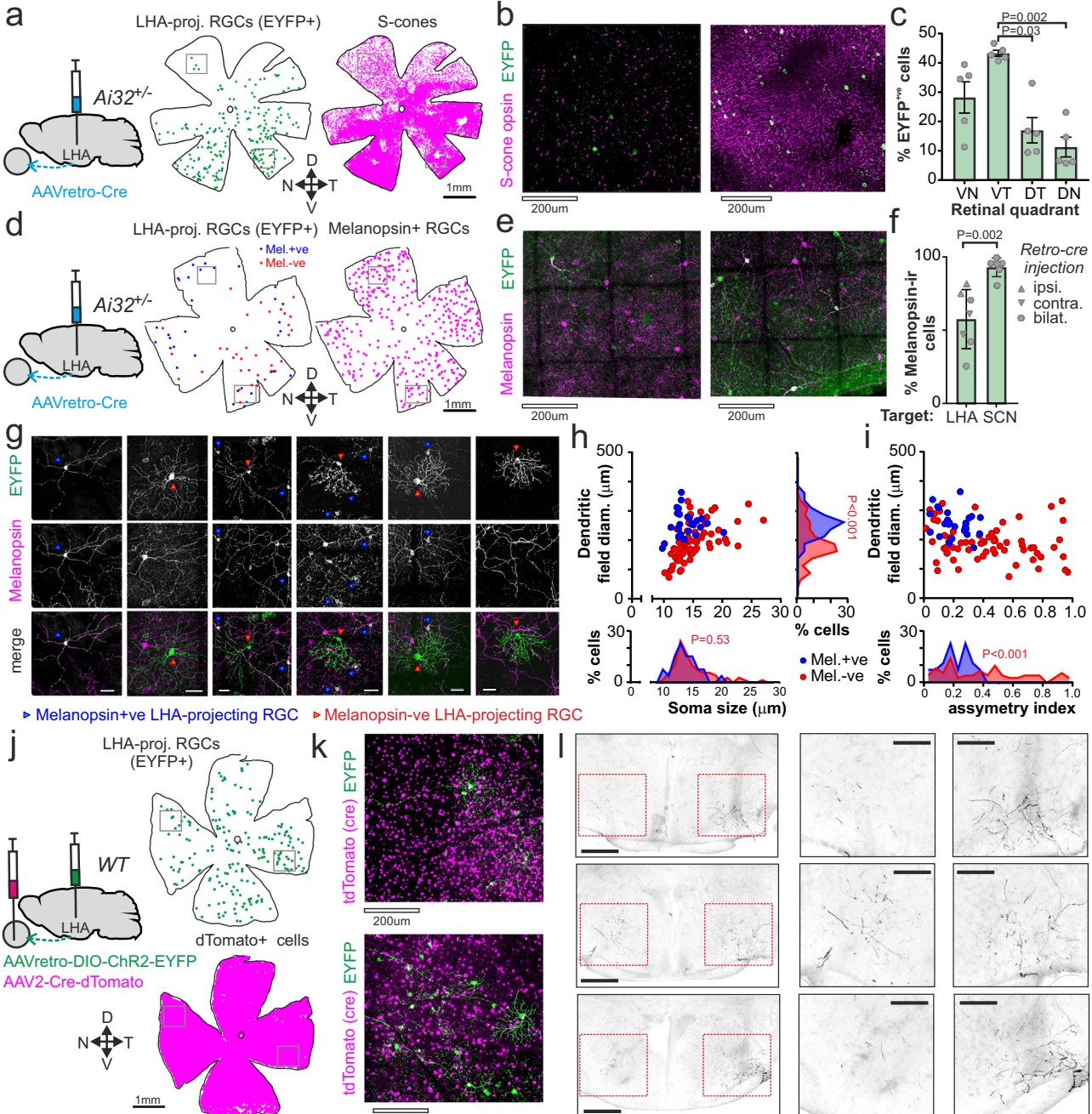

**Fig. 6 | Diversity of retinal ganglion cell projections to the lateral hypothalamic area. a** Contralateral retina from an Ai32 mouse following unilateral AAV2retro-Cre injection into the LHA, showing locations of LHA-projecting RGCs and S-opsin immunoreactive cones (representative of 5 similarly treated mice). **b** Confocal max-projections showing EYFP+ RGCs alongside the dorsal (left) and ventral (right) S-cone distribution, corresponding to boxed regions in (**a**). **c** Mean ± SEM percentage of LHA-projecting RGCs as a function of contralateral retinal quadrant (from 5 Ai32 mice receiving unilateral AAV2-retro-Cre injections). Data analysed by one-way RM ANOVA (F$_{1.63,6.5}$ = 9.84, $P$ = 0.01) with Tukey's post-tests. **d** Contralateral retina from a unilaterally injected Ai32 reporter mouse (as in **a**) showing the locations of Melanopsin + and − ve LHA-RGCs and overall distribution of melanopsin-immunoreactive RGCs (representative of 7 retinas from 4 mice). **e** Confocal max-projections showing EYFP and melanopsin-labelled RGCs from dorsal (left) and ventral (right) regions corresponding to boxes in (**d**). **f** Mean ± SEM percentage of melanopsin + retrograde labelled RGCs following unilateral or bilateral injection of AAV2-retro-Cre into the LHA vs. SCN ($n$ = 7 retinas

from 4 mice and $n$ = 6 retinas from 3 mice, respectively). Data analysed by a two-tailed unpaired $t$ test. **g** Example LHA-projecting RGCs from retinas co-stained for EYFP and melanopsin ($n$ = 7 retinas examined from 4 mice in total). Scale bars = 50 μm. **h**, **i** Dendritic field diameter vs. soma size (**h**) or dendritic field asymmetry (**i**) for RGCs retro-labelled from the LHA ($n$ = 96 cells from 6 retinas). Distributions of soma size, dendritic field diameter and asymmetry (inset histograms from **h** and **i**) compared across melanopsin+ and melanopsin-ve cells by a two-tailed Kolmogorov-Smirnov test. **j** Retina from a WT mouse receiving bilateral LHA micro-injections of AAVretro-DIO-ChR2-EYFP and intravitreal injections of AAV2-Cre-P2A-tdtomato showing the locations of EYFP + (LHA-projecting) RGCs and dTomato + (Cre-transduced) cells. **k** Confocal max-projections showing EYFP + RGCs and Cre-transduced cells corresponding to boxed regions in (**j**) (representative of 4 retinas from 2 mice). **l** Rostral to caudal (top to bottom) images showing labelled RGC process in the LHA following intersectional labelling in (**j** and **k**), Scale bars = 500 μm for left panels and 250 μm for insets (rightmost panel reflects LHA contralateral to retina in **j**).

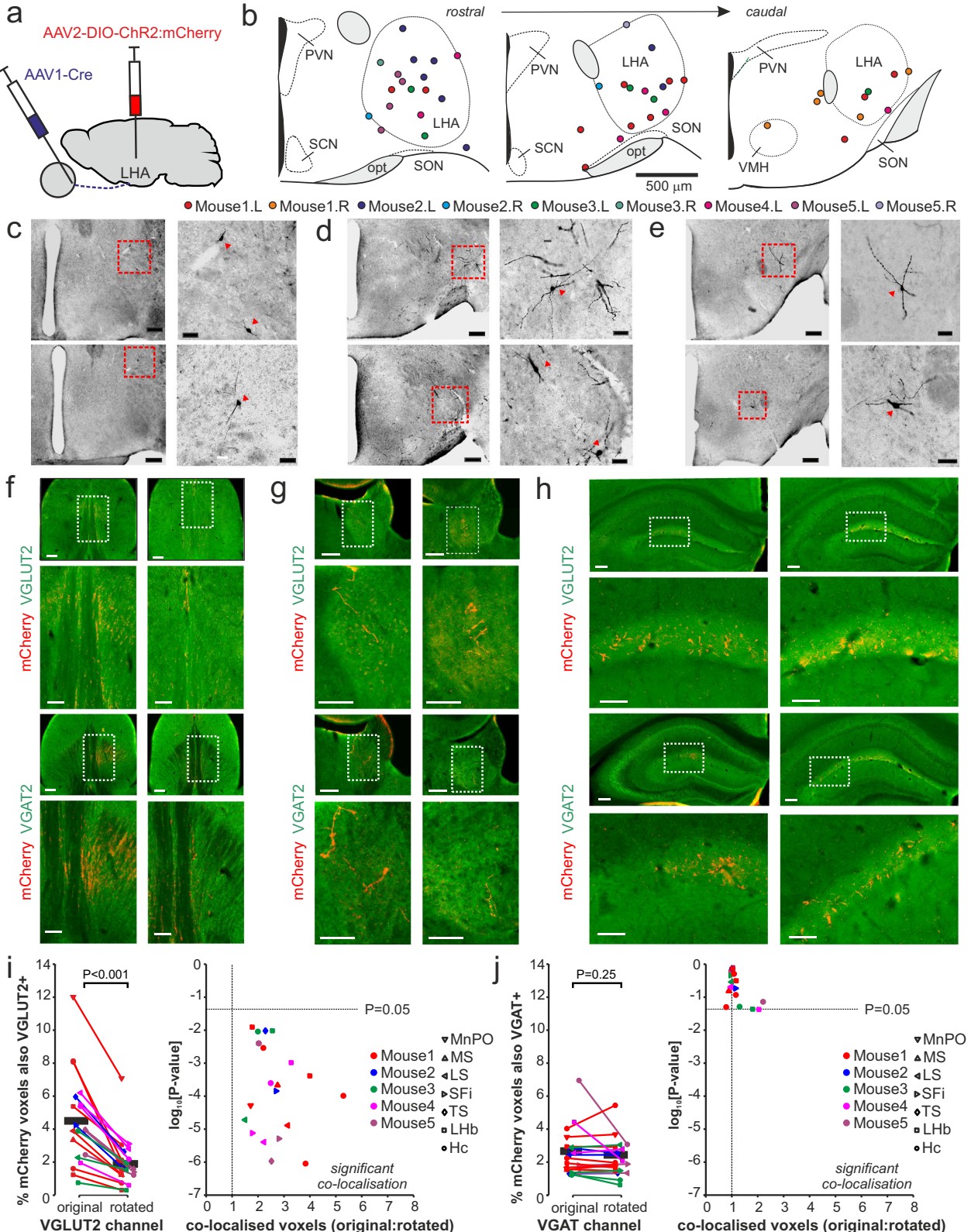

Where LHA- or SCN-projecting RGCs where sufficiently separated from neighbouring EYFP + cells for reliable analysis, we went on to assess the morphological properties of labelled cells. Among the LHA-projecting RGCs that displayed detectable melanopsin expression, we typically observed the sparse, relatively broad dendritic arbours that typify M1 ipRGCs (Fig. 6g–i). Indeed, we did not detect any significant difference in dendritic field diameter or soma size between such cells

and ipRGCs retro-labelled from the SCN (Supplementary Fig. S5k–m). We did, however, observe more variability among the morphological properties of LHA-projecting RGCs that lacked melanopsin expression. On average, these melanopsin-negative LHA-projecting RGCs had similar soma size but smaller dendritic fields than their melanopsin immunoreactive counterparts, although we also observed occasional cells with very large cell bodies and broader dendritic fields (Fig. 6g–i).

**Fig. 7 | Retinorecipient lateral hypothalamic neurons provide glutamatergic input to the septal region, lateral habenula and hippocampus. a** Schematic of a strategy to label retinorecipient LHA cells via bilateral intravitreal injections of AAV1-Cre and intra-LHA microinjection of a Cre-dependent reporter construct (AAV2-DIO-ChR2-mCherry). **b** Locations of retinorecipient hypothalamic neurons labelled on left and right hemispheres (from 5 mice), mapped onto anatomical templates corresponding to rostral – mid/caudal levels of the LHA. **c–e** Representative mCherry-labelled (retinorecipient) neurons across rostral – mid/caudal LHA (from 5 injected mice in total). Scale bars = 200 μm for macro images (Left panels) and 50 μm insets (right panels). **f–h** Epifluorescent images of mCherry labelled fibres in the septal region (**f**), lateral habenula (**g**) and hippo- campus (**h**) co-stained for VGLUT2 (upper panels) or VGAT (lower panels); repre- sentative images from two separate animals in each case. Scale bars = 200 μm for

macro images and 100 μm for insets. **i, j** Analysis of mCherry co-localisation with VGLUT (**i**) and VGAT (**j**) via confocal imaging of ROIs across identified LHA targets in the septal region, lateral habenula and hippocampus ($n$ = 20 ROIs analysed in total across 5 mice, in each of **i** and **j**). Left panels show the mean percentage of mCherry + voxels that were VLUGT2 + or VGAT + vs. the proportion observed when the green channel was rotated 90° (each data point derives from 10, 70 × 70 μm, subfields/ ROI; see Supplementary Fig. S7 for examples); data analysed by paired $t$ test. Right panels show the mean proportion of co-localised voxels, relative to chance, and the respective probability (via two-tailed paired $t$ test) of significant co-localisation across the 10 subfields analysed for each ROI. MnPO = median preoptic nucleus, MS = medial septum, LS = lateral septum, SFi = septofrimbral nucleus, TS = triangular septum, LHb = lateral habenula, Hc = hippocampus.

While we didn't undertake a detailed analysis, we also noted that dendritic fields tended to be substantially more densely branched among the melanopsin-negative cells and, in many cases, highly asymmetrical (potentially indicative of direction-tuned cells[53], Fig. 6i).

Given a previous report suggesting dsRGCs might project to the LHA[24], in another subset of retinas, we labelled LHA-projecting RGCs for the dsRGC marker CART[54]. Consistent with that earlier study, we could indeed identify occasional CART + LHA-projecting RGCs, although these were rare (4 ± 2% of labelled cells from 5 retinas; Sup- plementary Fig. S5f–h). Hence, the majority of LHA-projecting cells that lacked detectable melanopsin expression also did not express CART (as confirmed in four retinas triple-stained for EFYP, CART and melanopsin; Fig. S5f–h). While the identities of RGCs in this latter group remain unknown, collectively these data align well with our electro- physiological findings, suggesting LHA visual responses derive from input via multiple classes of RGC.

The greater diversity of LHA-projecting vs. SCN-projecting RGC types identified above strongly aligns with our electrophysiological studies and argues against any major bias in our labelling approach as the origin of the high proportion of melanopsin-ve RGCs labelled from the LHA. Likewise, our findings that LHA-projecting RGCs are enriched in ventrotemporal portions of the retina align with our electro- physiological data (Fig. 3). To provide additional confidence that this asymmetric labelling did not reflect some intrinsic bias in the approach used, we performed additional control experiments and analysis. We first examined the distribution of SCN-projecting (Supplementary Fig. S5i–k) and also LGN-projecting RGCs (Supplementary Fig. S5n–p), labelled in the same way. Unlike our studies of LHA-projecting RGCS (Fig. 6a–f), these did not show a consistent enrichment of labelled RGCs in ventral portions of the retina, with LGN injections in particular labelling large numbers of RGCs across the retina (Fig. S5t). By con- trast, cholera toxin β-subunit injections into the LHA of wild-type mice did consistently produce a pattern of RGC equivalent to that observed using AAVretro-hSyn-Cre in Ai32 mice, albeit with somewhat lower efficiency (Supplementary Fig. S5r–t).

Since our AAV-based tracing approach also labels central neurons projecting to the injected region, we further took the opportunity to assess the presence of projections from other visual regions that might explain or contribute to LHA visual responses. As expected[55], while we observed occasional retro-labelled neurons in the shell region of the SCN, retro-labelled cells were notably absent from the visual thalamus, superior colliculus and visual cortex (Supplementary Fig. S6). While the presence of labelled cells with this approach would not itself imply direct input to the retinorecipient LHA population, their absence clearly rules out any contributions from the primary visual relays.

Having confirmed a neuroanatomical arrangement of LHA- projecting RGCs consistent with the observed functional properties of LHA visual responses, we next turned our attention to their pro- jection targets and the possibility that they send collateral projections to other brain regions. To this end, we used wildtype mice and per- formed bilateral intra-LHA microinjections of a retrograde, Cre-

dependent, reporter (AAVretro-DIO-ChR2-EYFP), coupled with bilat- eral intravitreal injection of AAV encoding Cre recombinase (AAV2- Cre-dTomato; Fig. 6j–l). As expected, while dTomato + (i.e., Cre- expressing) cells were found at high density across the retina, this approach resulted in a modest number of EYFP + (i.e., LHA-projecting) RGCs ($n$ = 76 ± 40 cells/retina $n$ = 4 retinas from 2 mice), enriched in ventral-temporal portions of the retina (Fig. 6j, k). Accordingly, examining the brains from these animals, labelled RGC projections were clearly observable across the anterior LHA (and peri-SON) region, consistent with the locations of light-responsive neurons from our electrophysiological studies and previous neuroanatomical tracing studies[2–4,15,16] (Fig. 6l and Supplementary Fig. S7d). By contrast, we did not see any overt labelled fibres across any other visual region exam- ined (Supplementary Fig. S7a–c, e–g). Hence, we did not find evidence that these cells send major collateral projections to other brain regions.

## Identity of retinorecipient lateral hypothalamic neurons

To better understand potential functions of LHA visual neurons, we next sought to identify their projection targets, using a dual virus approach to sparsely but selectively label retinorecipient LHA cells. Hence, we performed bilateral intravitreal injections of AAV1-hSyn-Cre (capable of anterograde transsynaptic transfection[56]) and combined with intra-LHA microinjections of a Cre-dependent ChR2-mCherry reporter AAV (Fig. 7a). As expected, this approach consistently labelled a small subset of neurons, within or close to the LHA/pSON region ($n$ = 44 cells identified from 5 mice; Fig. 7b–e) but not other retinor- ecipient targets (Supplementary Figs. S8, S9a–c).

Since the ChR2-mCherry reporter used in these experiments strongly labels axons and terminals, we were able to trace the pro- jection targets of the population of retinorecipient LHA neurons. For three animals that showed the most robust neuronal labelling, we performed detailed fibre tracing across one hemisphere (Supplemen- tary Fig. S8), revealing a pattern of projections that aligned with pre- vious reports of LHA-projections using untargeted tracing[15,23]. Hence, fibres of retinorecipient LHA neurons were distributed widely across the hypothalamus and encompassed a range of external targets including the septal complex, midline thalamus and epithalamus, hippocampus and midbrain. Of particular note, outside of the hypo- thalamus, the most consistent projections observed were to the septal complex, habenula and hippocampus (Fig. 7f–h).

To provide additional insight into the nature of retinorecipient LHA-neuronal projections, we co-labelled alternate brain sections from these tracing experiments with markers of glutamateric or GABAergic terminals (VGLUT2 and VGAT, respectively; Fig. 7f–h). We then per- formed confocal imaging across identified terminal fields of mCherry- labelled retinorecipient LHA neurons across the septal complex, habenula and hippocampus and quantified the proportion of voxels that co-localised with VGLUT2 or VGAT (Supplementary Fig. S9d–g). Since both these markers are widely expressed, we controlled for the possibility of spurious co-localisation by also performing the same

analysis after locally rotating the VGLUT2/VGAT channel 90° across individual subfields (see "Methods"). Across every region of interest examined from every animal (20 ROIs from 5 mice), we found significantly greater co-localisation of mCherry and VGLUT than expected by chance (Fig. 7i and Supplementary Fig. S9d, e). By contrast, we never saw evidence of significant co-localisation of mCherry and VGAT (Fig. 7j and Supplementary Fig. S9f, g; 20 ROIs from 5 mice, as above). These data therefore indicate that retinorecipient LHA neurons provide overwhelming, if not exclusively, glutamatergic input to key brain sites implicated in goal-oriented behaviours, including the septal complex, habenula and hippocampus.

To rule out the possibility that the apparent absence of VGAT-expression by retinorecipient LHA neurons reflected some inherent bias in the viral labelling approach employed above, we further employed optogenetic approaches to identify GABAergic neurons during electrophysiological recordings. To this end, we performed optrode recordings in mice in which (GAD2-expressing) GABAergic neurons expressed the optogenetic actuator, ChR2 ($n = 15$ GAD-ChR2 mice; Supplementary Fig. S10). In line with our neuroanatomical data, none of the visually responsive LHA neurons identified in these experiments displayed the hallmark rapid excitatory responses to optogenetic stimulation that would identify them as GABAergic neurons. Importantly, this dataset included samples of neurons spanning the full range of properties previously identified in *Opn1mw$^R$* mice. Hence, we readily identified populations of LHA neurons with transient and sustained responses to full-field light steps (Supplementary Fig. S10a) and, using a modified cone-directed stimuli validated for use in mice with native M-cone expression[40], found subsets of both groups displaying cone-opponent responses (Supplementary Fig. S10b, c). Moreover, in a subset of recordings ($n = 7$ mice) where we employed spatially structured stimuli (as per Figs. 3, 4), we routinely identified the full spectrum of response properties reported earlier, including cells that lacked responses to full field stimuli but responded to spatially patterned and/or moving stimuli (Supplementary Fig. S10d–h). Critically, while none of the visually responsive cells identified in these experiments ($n = 179$) were excited by optogenetic stimulation, we did find many that reduced their firing rates and/or displayed a strong rebound excitation on termination of the light flash ($n = 63/179$, Supplementary Fig. S10j). That latter observation is indicative of strong GABAergic input to the recorded cells and confirms that our stimuli were sufficient to activate GABAergic neurons. Accordingly, cells recorded from nearby, non-visually-responsive, LHA regions often displayed the rapid and robust increases in firing during optogenetic stimulation expected for GABAergic cells ($n = 67/229$ non-visually responsive neurons; Supplementary Fig. S10k). In sum, then, these data strongly support the view that retinorecipient LHA neurons are excitatory rather than inhibitory cells.

### Functional roles of lateral hypothalamic visual inputs

To provide insight into the functional significance of the LHA visual circuitry, we finally evaluated the impact of selectively manipulating retinal input to this region. To achieve this, we delivered Cre recombinase to LHA-projecting RGCs via bilateral microinjections of retrograde-optimised AAV[57] into the LHA of WT mice, combined with bilateral intravitreal injections of Cre-dependent DREADDs or control fluorescent vectors (Fig. 8a). This strategy therefore provided a means to selectively activate (AAV2-hSyn-FLEX-hM3D(Gq)-mCherry) or inhibit (AAV2-hSyn-FLEX-hM3D(Gi)-mCherry) LHA-projecting RGCs.

To confirm the effectiveness of our approach, animals were transferred to constant darkness and received a DREADD selective dose of clozapine[58,59] (CLZ, 0.1 mg/kg i.p) (Fig. 8a). Animals were culled 90 min later (around mid projected day; pZT 4–8) and retinas and brains collected for confirmation of viral expression and c-Fos expression as a marker of neuronal activity. As expected, across experimental groups, retro-labelled LHA-projecting RGCs were readily

identifiable by mCherry expression, and the proportion of such cells that were co-labelled for c-Fos immunoreactivity was significantly greater in the Gq-DREADD group ($n = 15$) than in either control or Gi-DREADD groups (both $n = 12$, Fig. 8b, c and Supplementary Fig. S11a). Perhaps unsurprisingly, given the low basal c-Fos expression in the retina of these extensively dark-adapted animals, there was no significant difference in c-Fos expression between LHA-projecting RGCs of Gi-DREADD vs. control vector transduced animals (Fig. 8c).

We next assessed c-Fos expression in the brains of these DREADD- and control vector-expressing animals. Importantly, we did not observe any significant differences in c-Fos expression between experimental groups in the SCN or other major visual targets - the LGN and superior colliculus (Fig. S11b-f). By contrast, there was a clear effect of chemogenetic manipulation in the LHA, with significantly higher density of c-Fos expressing cells in the Gq-DREADD vs. Gi-DREADD groups (Fig. 8d, e). In sum, these data are consistent with the expected selective activation of (excitatory) retinal input to the LHA.

Notably, we also observed a similar influence of chemogenetic manipulation on c-Fos expression across two of the identified targets of retinorecipient LHA neurons. Hence, c-Fos was significantly elevated in the Gq- vs. Gi-DREADD animals in both the septal region (Fig. 8f, g) and habenula (Fig. 8h, i), consistent with our earlier results indicating that retinorecipient LHA neurons provide glutamatergic projections to these regions. We also examined c-Fos expression in another identified target, the hippocampal complex (Supplementary Fig. S11h, i). Here, overall c-Fos expression was very low under our experimental conditions, and we did not observe any significant effect of chemogenetic manipulation.

Given the diversity of sensory properties of LHA-visual neurons and their neural connections, retinal inputs to the LHA could influence a diverse array of goal-oriented behaviours. To provide insight into such functions, we assessed the impact of chemogenetically manipulating LHA-projecting RGCs on behaviour in an open field area and responses to visual and non-visual sensory cues (light flashes, looming disks and auditory stimuli). To this end, prior to the neuroanatomical studies described above, experimental animals were treated with CLZ (0.1 mg/kg, i.p.) and, after 45 min, introduced to the experimental arena where they experienced intermittent flash, loom and auditory stimuli presented in block-randomised order (as described previously[60]; Fig. 9a). Via the aid of a calibrated 4-camera setup and established algorithms for markerless 3D-pose reconstruction, we then extracted a set of three metrics describing mouse posture (Body rear, elongation and bend) and six motion-related metrics that variously described the rate of change in mouse posture (Δ rear, Δ elongation, Δ bend), position in the behavioural area (Locomotion, rotation) or immobility (Freezing)[60,61].

Consistent with previous work[60], light flashes, overhead looming discs and auditory stimulation each produced a distinct multidimensional behaviour response, with flashes driving increased locomotion and body elongation, looming disks (generally considered to simulate ariel predators[62]) driving increased freezing and rearing and auditory stimuli driving increased body bending and rotation (Fig. 9b−e and Supplementary Fig. S12a−d). Most importantly, however, while we found no evidence that spontaneous mouse behaviour was influenced by chemogenetic manipulation under our experimental conditions (neither among the 9 behavioural features noted above nor in distance relative to the centre of the behavioural arena; Supplementary Fig. S12e, f), a differential modulation of behavioural responses to these various sensory stimuli was readily detectable.

In the case of light flashes, we observed a significant reduction in stimulus-evoked responses in the Gq-DREADD group relative to control, with significantly smaller increases in locomotion during the first 1 s after stimulus onset, relative to 2 s prior to stimulus onset (Fig. 9c, d). Responses of the Gi-DREADD group were not significantly different from control animals, although this group also displayed a

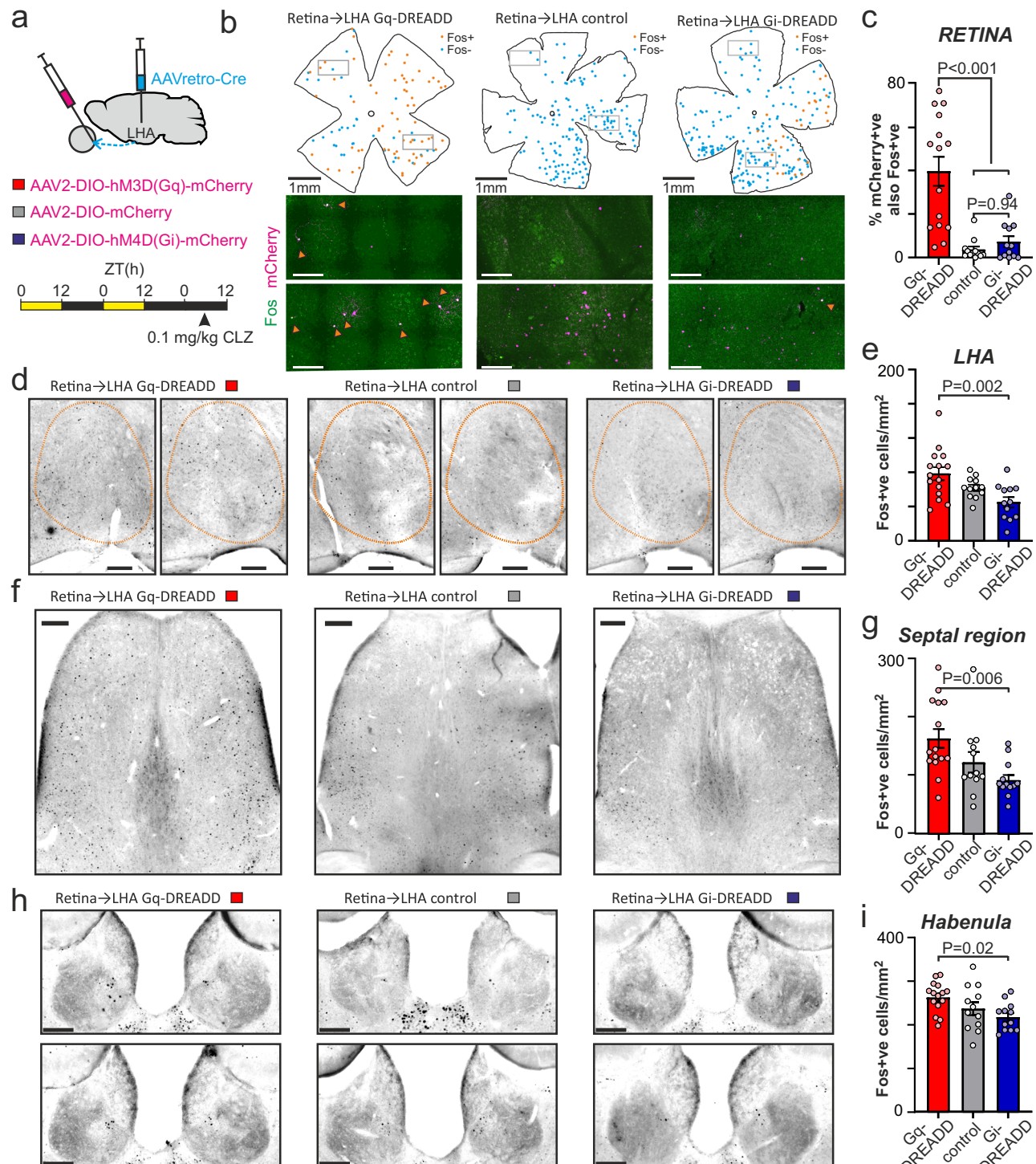

**Fig. 8 | Selective chemogenetic manipulation modulates activity of the retinorecipient lateral hypothalamic area and downstream targets. a** Approach for selective manipulation of LHA-projecting RGCs by bilateral microinjection of AAVretro-Cre into the LHA and Cre- dependent Gq- or Gi-coupled DREADDs (or control vector) into the retinas. Injected animals, in constant darkness, received clozapine (CLZ; 0.1 mg/kg i.p.), 90 min before culling (during mid projected day) for immunohistochemical examination. **b** Representative whole mount retinas co-labelled for Fos and mCherry to identify LHA-projecting RGCs (left to right, Gq-DREADD expressing, mCherry control, and Gi-DREADD expressing). Top panels show distribution of Fos + ve and Fos-ve LHA-projecting RGCs, lower panels show confocal max-projection corresponding to boxed regions in

upper panels, arrowheads indicate double labelled cells, scale bars = 200 μm. **c** Mean ± SEM percentage of retro-labelled retinal neurons co-labelled for c-Fos ($n$ = 15,12 and 12 animals respectively for Gq DREADD, control and Gi-DREADD). **d**, **f**, **h** Representative images of c-Fos expression across the LHA (**d**), septal region (**f**) and habenula (**h**) from Gq-DREAAD, control and Gi-DREADD (left- right, respectively) transduced animals. **e**, **g**, **i** Mean ± SEM density of c-Fos immunoreactive nuclei from LHA (**e**), septal region (**g**) and habenula (**i**) across experimental groups ($n$ = 15,12 and 12, respectively for Gq DREADD, control and Gi-DREADD). Data analysed by one-way ANOVA with Sidak's post-tests (**c:** $F_{2, 36}$ = 18.79, $P < 0.0001$, **e:** $F_{2, 36}$ = 7.25, $P$ = 0.002, **g:** $F_{2, 36}$ = 5.7, $P$ = 0.007, **i:** $F_{2, 36}$ = 4.33, $P$ = 0.021). Scale bars in **d**, **f**, **h** = 200 μm.

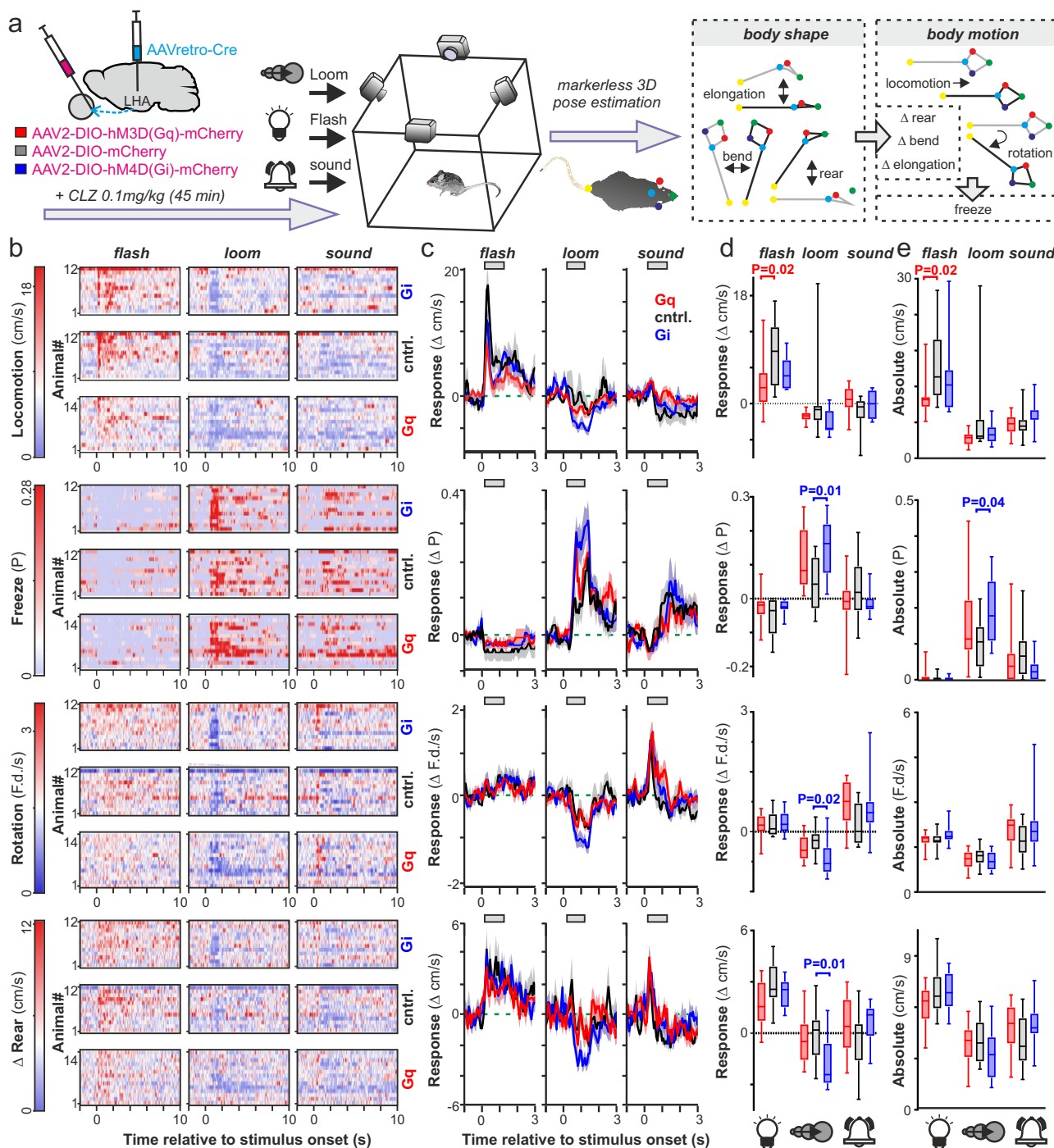

similar trend towards reduced light-flash-induced locomotion. Conversely, behavioural responses to the appearance of looming disks were selectively modulated in the Gi-DREADD group, with significantly greater freezing probability compared to controls, accompanied by significantly greater reductions in two movement related variables (Δ rear and rotation; Fig. 9c, d). Here, while responses of the Gq-DREADD group were not significantly different from control, they again trended in a similar direction to the Gi-DREADD group, with a nominally greater increase in freezing probability than control mice.

Consistent with our analysis showing spontaneous behaviour did not vary between groups in any of the tested dimensions (Fig. S12e, f), the effects of chemogenetic manipulation on flash and loom-evoked behavioural responses reported above were also apparent when we analysed post-stimulus behaviour without

subtracting pre-stimulus activity (Fig. 9e). Hence, Gq-DREADD animals showed significantly reduced locomotion after light flashes and Gi-DREADD animals showed an increased probability of freezing after looming stimuli. Similarly, we further found that the duration of looming-evoked freezing events (see "Methods") was significantly increased in the Gi-DREADD vs. control animals (Supplementary Fig. S12g). In additional analysis, we also assessed the degree to which the observed effects of chemogenetic manipulation on loom-induced freezing and flash-induced locomotion were influenced by averaging responses of each animal across multiple trials (Supplementary Fig. S13). In both cases, regardless of whether we quantified change from baseline (Supplementary Fig. S13a, c) or absolute post-stimulus behaviour (Supplementary Fig. S13b, d), the observed effects of chemogenetic manipulation did not vary significantly as we

**Fig. 9 | Retinal inputs to the lateral hypothalamic area modulate visually-guided mouse behaviour. a** Following Gq-DREADD, Gi-DREADD or control vector targeting to LHA-projecting RGCs, clozapine-treated mice (CLZ; 0.1 mg/kg, i.p.) were introduced to an arena where they experienced light flashes, looming disks and auditory stimuli. Multi-dimensional behavioural responses were quantified following 3D reconstruction of mouse body parts to generate 3 posture-related (rear, elongation and bend) and 6 movement-related indices ($\Delta$Rear, $\Delta$Elongation, $\Delta$Bend, Locomotion, Rotation (Frobenius distance; F.d) and Freeze probability; see "Methods"). **b** Heatmaps for each animal/experimental group (Gq-DREADD, $n = 14$; Gi-DREADD, $n = 12$; control, $n = 12$) and stimulus type (left to right: light flash, looming, auditory) showing motion-related behavioural components sensitive to chemogenetic manipulation (see Supplementary Fig. S12a–d for additional motion and posture-related components). Heatmaps show trail-averaged, absolute values of each component (colour limits set to $\pm 3.5$ S.D. around the grand mean of 2 s pre-stimulus values, pooled across stimulus types, animals, and groups). **c** Mean $\pm$ SEM change in each behavioural component (relative to 2 s pre-stimulus mean) for Gq-DREADD, Gi-DREADD and control mice (same animals as **b**) in responses to light flash (left), looming (middle) or auditory (right) stimuli. Bars above each plot represent analysis epoch in (**d** and **e**). **d**, **e** Box and whisker plots (box=quartiles 2-3, centre = median, whiskers = min-max) showing change from baseline (**d**; relative to 2 s preceding stimulus) and absolute values (**e**) for motion-related components from (**b**, **c**) across Gq-DREADD, Gi-DREADD and each stimulus type. Data analysed by 2-way mixed effects ANOVA, with stimulus type as within-subjects variable (**d**: *Locomotion:* DREADDXStimulus-$F_{4,70} = 3.80$, $P = 0.008$; DREADD-$F_{2,35} = 3.46$, $P = 0.04$, Stimulus-$F_{2.0,69} = 43.7$, $P < 0.0001$; *Freeze probability:* DREADDXStimulus-$F_{4,70} = 3.77$, $P = 0.014$; DREADD-$F_{2,35} = 1.78$, $P = 0.18$, Stimulus-$F_{1.9,65} = 39.4$, $P < 0.0001$; *Rotation:* DREADDXStimulus-$F_{4,70} = 3.66$, $P = 0.009$; DREADD-$F_{2,35} = 0.61$, $P = 0.55$, Stimulus-$F_{1.6,57} = 41.8$, $P < 0.0001$; $\Delta$ *Rear*: DREADDXStimulus-$F_{4,70} = 5.81$, $P < 0.0001$; DREADD-$F_{2,35} = 0.48$, $P = 0.62$, Stimulus-$F_{2.0,55} = 54.8$, $P < 0.0001$; **e**: *Locomotion:* DREADDXStimulus-$F_{4,70} = 3.44$, $P = 0.01$; DREADD-$F_{2,35} = 2.50$, $P = 0.10$, Stimulus-$F_{1.8,65} = 93.2$, $P < 0.0001$; *Freeze probability:* DREADDXStimulus-$F_{4,70} = 3.46$, $P = 0.01$; DREADD-$F_{2,35} = 0.31$, $P = 0.73$, Stimulus-$F_{1.7,61} = 58.5$, $P < 0.0001$; *Rotation:* DREADDXStimulus-$F_{4,70} = 2.16$, $P = 0.08$; DREADD-$F_{2,35} = 0.53$, $P = 0.59$, Stimulus-$F_{1.4,48} = 54.0$, $P < 0.0001$; $\Delta$ *Rear*: DREADDXStimulus-$F_{4,70} = 4.21$, $P = 0.004$; DREADD-$F_{2,35} = 0.02$, $P = 0.98$, Stimulus-$F_{1.8,65} = 84.8$, $P < 0.0001$). Dunnett-s post-tests between Gq-DREADD/control and Gi-DREADD/control were applied wherever ANOVA revealed a significant effect of DREADDXStimulus (Locomotion, Freeze probability, $\Delta$ rear, rotation); statistically significant differences indicated on the relevant plots, otherwise $P > 0.05$ for all relevant comparisons.

included progressively more trials, beyond the very first exposure to each stimulus.

By contrast to the effects reported above, we did not observe any significant impact of either excitatory (Gq-DREADD) or inhibitory (Gi-DREADD) modulation of LHA retinal inputs on mouse behavioural responses to auditory stimuli, either with respect to the acute stimulus-driven changes in the various parameters analysed, nor in the absolute post-stimulus values (Fig. 9c–e; Supplementary Fig. S12a–d). Collectively, then, we show that manipulating that activity of LHA-projecting RGCs alters mouse behavioural responses in a stimulus-specific manner, establishing a role for this pathway in shaping visually-evoked reflex behaviours, including defensive responses to potential environmental threats.

## Discussion

Here, we provide fundamental insight into the properties and roles of visual input to the LHA, which go far beyond the prototypical role for hypothalamic retinal input in regulating physiology and behaviour according to daily changes in ambient light levels. Indeed, while some visually responsive LHA neurons are clearly capable of providing information about ambient light levels, a greater proportion are highly selective for spatiotemporal contrast. This includes sizable subsets of LHA cells that preferentially respond to simple or complex motion or are entirely insensitive to spatially-uniform visual stimuli but respond robustly to spatial patterns. Alongside our findings of discrete, centre-surround, RFs and evidence for a retinotopic order to LHA visual input, the potential capabilities of this nucleus seem more closely aligned with conventional visual targets then expected for the retinorecipient hypothalamus. Accordingly, we show that retinorecipient LHA neurons provide excitatory input to a range of extra-hypothalamic sites involved in decision making and goal-oriented behavioural control and demonstrate overt contributions of LHA retinal inputs to visually-guided behaviour.

Despite a long history of neuroanatomical tracing, demonstrating retinal projections to the LHA across mammals[17–19], the visual response properties of neurons in the retinorecipient LHA region have been largely overlooked. The diversity of sensory properties identified herein (Fig. 5) imply a range of RGC types project to the LHA and, accordingly, our retrograde tracing studies provide direct evidence for this (Fig. 6). On one hand, it is well established that at least one subtype of ipRGC projects to the LHA[1–3] and both our findings that a subset of LHA visual neurons display detectable melanopsin-driven increases in firing and that a subset of LHA-projecting RGCs are melanopsin-immunoreactive are consistent with those data. Importantly, however,

we also find a sizable number of LHA-projecting RGCs which lack detectable melanopsin-immunoreactivity and seem to comprise more than one ganglion cell class, based on morphological properties. In principle, such cells may include subtypes of ipRGC with very low melanopsin expression (M4-6[63,64]), although it is abundantly clear from our electrophysiological data that many LHA visual neurons lack functionally detectable melanopsin-driven responses, even at very high light levels (Fig. 2a–h). Our conclusion that one or more subtypes of 'conventional' RGC projects to the LHA is further supported by two previous studies providing evidence that not all retinal input to the LHA derives from ipRGCs[1,24]. Indeed, our finding that a small subset of LHA-projecting RGCs express CART aligns with data from one of those studies[24], while the presence of CART and melanopsin-negative cells suggests additional, as yet unidentified, RGC types.

In sum, then, our retrograde tracing studies provide a neuroanatomical substrate for the diversity of LHA sensory properties revealed by our electrophysiological recordings. Our additional retrograde labelling experiments performed from the SCN (Fig. S5i–m) further rule out the potential concern that the proximity of our LHA injections to the optic tract resulted in non-specific labelling of RGCs projecting to other visual regions. Hence, with the equivalent retro-labelling strategy applied to the SCN (where injections are even closer to the optic tract), we overwhelmingly found melanopsin-immunoreactive (M1-like) RGCs, as expected.

Our neuroanatomical identification of LHA-projecting RGCs is also consistent with findings from our electrophysiological (RF-mapping) studies (Fig. 3). While such cells can be found across the retina, they are preferentially enriched in the ventrotemporal quadrant (corresponding to regions of visual space in front of the animal and overhead). This finding was consistent across a variety of labelling approaches and not observed for retro-labelling for other visual targets (Fig. 6 and Supplementary Fig. S5), ruling out the possibility it simply reflects some intrinsic bias in the approach used. Interestingly, another recent study reported that retinal projections to the SON arise preferentially from the dorsal retina[16]. That finding broadly aligns with our data, indicating a crude retinotopic order to LHA visual responses, with dorsal retinal locations more often represented by cells in the ventral LHA (overlapping the peri-SON region) and ventral retinal locations more often represented by cells in dorsal LHA. It is also noteworthy that the previous study focused specifically on a subset of M1 ipRGCs (labelled in GlyT2cre mice), which are seemingly entirely restricted to the dorsal retina, with more global ipRGC-targeting also revealing cells in the ventral retina[16]. Presumably, the still modest number of ventrally labelled cells reported in those previous studies

reflects specific targeting of ipRGC projections to the SON region rather than overall RGC projections to the wider LHA.

Beyond the diversity in LHA neurons sensory properties revealed here, another notable aspect of our findings is that overall visual responses are surprisingly extensive across the anterior LHA. Hence, our unbiased MUA mapping experiments (Fig. 1) revealed robust responses across the ventral tip of the LHA/peri-SON, where retinal fibres are most dense, but also extending to more dorsal LHA regions where retinal afferents are more sparsely labelled[1–4]. That arrangement, nonetheless, aligns with results from our intersectional anterograde tracing studies, which labelled retinal projections and retinorecipient LHA neurons across the regions where we detected LHA visual responses (Figs. 6l, 7). In interpreting these data, it is important to note that we cannot make definitive conclusions as to the exact proportions of LHA neurons that receive visual inputs. Clearly, such input is not homogeneous across even the anterior LHA, and for electrophysiological analysis of single neuron properties, we targeted recordings to visually responsive regions of the LHA (based on responses to full-field light steps). The most comprehensive of such recordings indicated virtually all detectable neurons in these regions responded robustly to at least some aspects of our stimulus set. It is possible, however, that other nearby neurons (perhaps many cells) remain silent under our experimental conditions and are thus undetectable. That point aside, it is clear from our data that neurons across the anterior portions of the LHA that receive visual input are certainly not rare.

It remains possible that aspects of the LHA visual responses we detect arise indirectly via other neurons. However, the robust and rapid LHA visual responses detected here (resembling those routinely detected in other visual regions in mouse[29,30,42–46]), alongside results of both our retrograde and anterograde tracing studies (Figs. 6, 7, Supplementary Figs. S5–7), argue against multisynaptic routes as a major source of LHA visual responses. Hence, firstly we see no evidence that the LHA receives input from primary visual regions (Fig. S6, consistent with existing Allen Mouse Brain connectivity data[55]). Secondly, find that on average, LHA visual responses are modestly, but significantly, faster than either those record from SCN or LGN under identical conditions (Supplementary Figs. S1f, S3e, f) and are similarly, on average, faster than reported latencies for visual thalamus and superior colliculus visual responses[45,46,65]. Hence, while we do find evidence that SCN neurons project to the LHA region and it is possible that these could contact visually responsive LHA neurons, they could not account for the responses we see; LHA visual responses are faster, more sensitive and more diverse than those of SCN neurons (Fig. 1 and Supplementary Fig. S1).

It is, however, possible that local communication with the LHA might indirectly confer visual responses to non-retinorecipient neurons or that some features of the LHA visual responses we see might arise by more complex multisynaptic pathways. Certainly, the rich array of single neuron sensory properties seen here seems to suggest it is unlikely that responses of every neuron are directly inherited via input from a single ganglion cell class. Thus, directly or indirectly, we suspect that convergence of signals from multiple RGCs underlies the properties of at least some LHA neurons. By the same token, for the many cells where we could map an RF, we overwhelmingly identified very spatially restricted excitatory centres and adjoining antagonistic surrounds. Thus, any convergent inputs must presumably arise from spatially congruent cells. Future studies using selective opto- or chemogenetic manipulation of retinal inputs to the LHA may provide more definitive insight into the respective roles of direct retinal vs. network-level mechanisms.

Our data also provide insight into the possible functions of LHA visual neurons. Cells within the anterior LHA region are known to project widely across hypothalamic and extra-hypothalamic sites implicated in regulating behavioural state, goal-directed behaviours

and viscera related control[15,23,66]. The pattern of fibre-labelling for retinorecipient LHA neurons aligns closely with those untargeted tracing studies (Figs. 7 and Supplementary Figs. S8, 9), although we typically observed the most pronounced extra-hypothalamic labelling in the septal complex, habenula and hippocampus. It is formally possible that the labelling approach used here does not reveal the full pattern of retinorecipient LHA neuronal projections. Nonetheless, in accordance with our neuroanatomical and electrophysiological data indicating these represent glutamatergic rather than GABAergic neurons (Supplementary Figs. S9, 10), we found selective activation of LHA visual projections drove increased c-Fos expression not only in the LHA but also across the septal region and habenula (Fig. 8). The lack of such changes in the hippocampus might reflect the comparatively sparse nature of LHA-projections to this region or might instead reflect the fact that impact of such projections is only evident when the animal is engaged in a salient task (i.e., by modulating responses to stronger driving inputs[67]).

In interpreting data from our chemogenetic manipulations, it is important to consider the degree of selectivity associated with the approach. Seemingly in common with other studies that have used related approaches[11,12,68–70], we did not detect substantive mCherry terminal labelling in the LHA (or elsewhere in the brain) of such animals. This may reflect the fact that the time course required for robust terminal expression is much longer than the 4 weeks used here and/or that the CLZ injections used to validate DREADD function drove some transient receptor internalisation[71,72]. Nonetheless, several observations provide confidence in the selectivity of our approach: (1) we observed a pattern of RGC labelling in these experiments consistent with all our other neuroanatomical investigations (Fig. 8 and Supplementary Fig. S11), (2) with a similar targeting approach, using ChR2-EYFP, we found no evidence for collaterals of LHA-projecting RGCs to other visual regions (Supplementary Fig. S7), under conditions where post-injection survival times (> 4 weeks) were longer than those previously found to produce robust labelling even in distant retinorecipient regions such as the superior colliculus[16,24,73–75], (3) we did not observe off-target labelling in visual regions with any other viral labelling approaches used here and (4) most notably, chemogenetic manipulations in the animals used for behavioural studies modulated c-Fos expression in the LHA and target regions but not SCN, visual thalamus or superior colliculus (Fig. 8 and Supplementary Fig. S11).

Given the known roles of the regions receiving input from retinorecipient LHA cells[76–78] and the sensory properties discussed above, the LHA could potentially contribute to a wide array of functions ranging from light-dependent effects on memory/cognition or mood to visually guided defensive or hunting behaviours[79,80]. In this regard, the relative enrichment of LHA visual input towards portions of visual space in front of the animal and overhead are certainly consistent with role in the latter types of goal-directed behaviours. Regardless, our electrophysiological data suggest two broadly different ways that LHA visual projections could, in principle, influence such functions. On one hand, they could provide information about ambient light levels that influence physiological and behavioural state and, in turn, indirectly modulate responses to environmental changes (e.g., through altering behavioural strategy according to differences in the perceived 'safety' of dimmer vs. brighter environments). Alternatively, LHA-visual projections could relay more specific information about salient environmental events to allow for more direct modulation of the resulting responses.

While our behavioural studies using chemogenetic manipulation likely only scratch the surface of the full range of roles that LHA visual input subserves, our present findings align with the latter possibility. Hence, if the primary role of LHA visual inputs was to adjust behavioural/physiological state according to ambient light levels one would expect chemogenetically activating or inhibiting LHA-projecting RGCs to exert opposite effects on spontaneous and/or stimulus-evoked

behaviour (and that responses to auditory as well as visual stimuli might be modulated). In fact, we found none of these things to occur. Rather, we found acute stimulus-specific changes in behaviour evoked by visual (but not auditory stimuli) that were, at least nominally, in the same direction following chemogenetic activation and inhibition (Fig. 9 and Supplementary Fig. S12). Since both such manipulations can, in principle, impair the reliable transmission of visual information, these data imply a more direct role for LHA visual inputs in the detection and response to potential environmental threats.

Of particular note, we found that chemogenetic inhibition of LHA-projecting RGCs increased freezing responses to overhead looming disks, simulating an aerial predator. Those data indicate that information about the appearance of this stimulus, relayed to the LHA, normally acts to reduce freezing responses, presumably because it favours alternative behaviours such as escape. In line with that view, a range of previous studies have implicated LHA glutamatergic neurons (projecting to the lateral habenula, periaqueductal grey and/or VTA) specifically in driving escape and associated aversive behaviours[21,51,81–83]. A role for the LHA is modulating responses to such stimuli is further consistent with our findings that a subset of LHA cells are selectively sensitive to motion (including cells to preferentially respond to 'looming'), as well as a previous study showing activation of LHA glutamatergic neurons by threatening stimuli[21]. Hence, our data now directly implicates retinal inputs to the LHA conveying information about potential threats by tuning the behavioural decision to freeze (a response driven by other brain circuits via the superior colliculus[84,85]) or escape.

Such a role for LHA visual inputs in driving escape is also consistent with our data showing that chemogenetic manipulation of LHA-projecting RGCs reduced light flash-induced increases in locomotion. In this case, we found significant effects for chemogenetic activation rather than inhibition, but for both flash- and loom-evoked responses, Gi-DREADD and Gq-DREADD manipulations changed the behavioural response profile in nominally the same direction. We interpret those data as reflecting the fact that both manipulations can impair the accurate transmission of visual information by RGCs (by respectively reducing or occluding responses), but that the degree of such impairment depends on the stimulus type. For light flashes, artificially increasing retinal input to the LHA presumably more strongly disrupts reliable assessments of changes in irradiance when penetrance of the manipulations is incomplete, whereas directly suppressing the transmission of information has a greater impact for looming responses than simply increasing baseline RGC activity.

To limit possible adaptation to the stimuli used here, for the present study, we applied the chemogenetic actuator (or control) 45 mins before animals experienced the test arena for the first time. As such, our behavioural experiments took place during a time window where we expect maximal DREADD effects should be maximal[58,86,87] in naïve animals. Morevoer, while animals experienced multiple repeats of each stimulus, we did not see any evidence that the observed effects of DREADD manipulations were significantly modulated according to how many trails (beyond the first) we included in our averages. Hence effects of chemogenetic manipulations on loom-induced freezing and increases in post-flash locomotion were evident from the first trial experienced by each animal. In future studies it will be interesting to more comprehensively explore the roles of LHA visual projections in modulating visually guided behaviour by varying the stimulus type (eg contrast, speed or position) and context (eg degree of prior familiarity, motivation to explore or availability of hiding places).

Certainly, the contributions of the retinorecipient LHA to visually-guided behaviour revealed here likely only constitute a subset of the full roles of this visual projection. A recent study further implicates GABAergic cells of the peri-SON region in light-dependent regulation of social recognition memory[88]. Since we here found no evidence that retinorecipient or visually responsive LHA cells expressed markers of GABAergic neurons, we suspect those later findings reflect a multi-synaptic pathway, potentially driven by the glutamatergic retinor-ecipient LHA neurons identified here. Such a view would explain the comparatively sluggish light-driven calcium reporter activity reported in those GABAergic peri-SON cells previously[88]. In either case, fully defining the roles of the LHA visual projection remains an exciting prospect for the future. Indeed, based on the sensory properties and projection targets identified here, this visual projection may provide potential origins for currently poorly understood aspects of how visual signals influence diverse facets of behaviour from consummatory behaviours to mood, memory and cognition.

## Methods

### Animals

All animals were used in accordance with the Animals, Scientific Procedures, Act of 1986 (UK), and all experiments received both University of Manchester ethics committee and the UK Home Office approval. Experiments utilised the following mice, bred in house: (1) Mice with the human L-cone opsin knocked-in to replace native mouse M-cone opsin ($Opn1mw^R$; $X^R Y$ males)[27]; (2) human L-cone opsin knockin mice crossed with melanopsin knockout mice ($Opn1mw^R$; $Opn4^{-/-}$)[31]; (3) mice heterozygous for a Cre-dependent ChR2-EYFP reporter construct knocked into the ROSA locus (Ai32; JAX #012569)[52]; (4) Ai32 mice crossed with GAD2-IRES-Cre (JAX #010802)[89] mice (Ai32$^{+/-}$; GAD2-Cre$^{+/-}$) to target the ChR2-EYFP reporter to GABAergic neurons. Other experiments utilised wildtype, C57BL/6NCrl, mice purchased from Envigo, UK. Unless otherwise stated, mice were group housed in 12:12 h light/dark cycles in a temperature-controlled environment (22 °C; humidity: 54%). Zeitgeber time (ZT) 0 was designated as the time of lights-on and ZT12 as lights-off. Food and water were provided *ad libitum*.

### In vivo electrophysiology

**Surgery and data acquisition.** Adult male mice (2–5 months old) were taken from the colony room 2-3 hr after lights on and anesthetised using urethane (1.55 g/kg i.p.; Sigma-Aldrich, Dorset, UK). Surgical procedures were similar to those previously described for SCN recordings[28,33]. For LHA recordings, mice were then placed in a stereotaxic frame; the skull was exposed, and a small craniotomy was performed 1.2 mm lateral and 0.7 mm posterior to bregma. Pupils were dilated by application of 1% atropine (Sigma Aldrich) and mineral oil applied to retain corneal moisture. Initial recordings utilised linear 32-site silicon probes with 50 µm site spacing (A32-1 x 32-10 mm-50-413; Neuronexus, MI, USA). Subsequent recordings used higher-density 32-site polytrode (2 parallel rows of 10–12 electrode sites, spaced 50 µm; A32-Poly3, Neuronexus) or Buszaki-style electrodes (4 shanks spaced 200 µm each with 8 closely clustered sites; A32-BuszakiL, Neuronexus). In all cases, probes were coated in coated in CM-DiI (V22888; Fisher Scientific, Loughborough, UK) for post-hoc histological visualisation, prior to insertion into the brain to a depth of ~ 5.3 mm from the pial surface using a micromanipulator (MO-10, Narishige International Ltd., London, UK). Mice were then left to dark adapt for 30 minutes before the start of recordings, such that recordings spanned the middle part of the light phase in the mouse's home cage LD cycle (Zeitgeber time 4.5-9).

During experiments, wideband neural signals were acquired using a Recorder64 system (Plexon, TX, USA), amplified to a gain of 3500×, digitised at 40 kHz and stored continuously in a 16-bit format. MUA was simultaneously acquired online from all channels by collecting timestamped 1 ms waveform segments triggered when the highpass (300 Hz) filtered data streams crossed a threshold set at -40µV. For initial experiments with linear probes, where light-evoked activity was not detected at any electrode site, we repositioned across a search grid ±200 µm medial and/or lateral to the initial starting position until light responses were detected (typically 3-4 positions tested per animal).

Single-unit activity was subsequently discriminated offline using commercial principle component-based sorting (Offline Sorter V3.3.5, Plexon) as described previously[28]. For experiments using high-density probes, single unit activity was isolated from the wideband data stream using an automated template-matching-based algorithm (Kilosort V2)[90]. Data for identified clusters were then exported to Offline Sorter as 'virtual tetrodes' (1 ms waveform segments detected across 4 adjacent channels) for manual refinement and validation as distinct single units on the basis of spike cross- and auto-correlograms as in refs. 30,40. A summary of the number of recordings, probe placements and isolated units for all electrophysiological recordings (and the relevant associated data figures) is provided in Supplementary Table S1.

**Visual stimuli and analysis.** All light measurements were performed using a calibrated spectroradiometer (Bentham Instruments, Reading, UK) and quantified according to the known opsin sensitivities after correction for prereceptoral filtering[91] as described previously[28–30,33,40].

Full field visual stimuli were generated via LED based systems with dichroic mirrors to combine two or more wavelengths and focus the resulting light path onto flexible 7 mm light guides that were positioned ~ 5 mm from contralateral (or in some cases both) eyes (systems from Carin Research, UK, or constructed in house from components supplied by Thorlabs, NJ, USA). For initial experiments (Fig. 1) the light system comprised 460 nm and 655 nm LEDS (identical to the system used in ref. 28) for all other experiments contralateral illumination was provided by a 3 LED system LEDs (λmax: 410, 470 and 617 nm), with illumination of the ipsilateral eye (where relevant) provided by a single 410 nm LED (system identical to that used in refs. 33,40,92,93). In all cases, LED intensity and timing was controlled via a PC running LabVIEW 2019 software and a USB-6343 DAQ board (National Instruments, TX, USA), with a neutral density (ND) filter wheels providing additional control over stimulus irradiance (up to a 1000000-fold; ND0-ND6).

For initial assessment of LHA irradiance response relationships (Fig. 1), mice were exposed to interleaved 460 nm and 655 nm light steps (2 s steps from darkness) spanning a 7 log-spaced irradiances (10 repeats of each stimulus starting at the lowest irradiance). Comparator SCN data at maximal irradiance was collected as part of an earlier study using the same stimuli[28]. Assessment of irradiance response relationships and occularity in wildtype mice (Supplementary Fig. S1g–l) used 410 nm light steps (2 s, 10 repeats/stimulus) and equivalent paradigms and analysis procedures to our previous studies in SCN and LGN[33,42]. Light-responsive cells (those that displayed a significant increase in firing during the first 250 ms following light ON or OFF) were classified as sustained when firing during the last 500 ms of the 2 s light step remained significantly higher than the 2 s before light onset (paired t-tests across trails). Latency measures for the new data presented here and the latter, previously published, studies (Supplementary Fig. S1f) employed a moving Gaussian kernel window (σ = 5 ms, 1 ms step size), with the first post-stimulus bin exceeding the 95% confidence limits of the (1 s) pre-stimulus firing activity taken as onset latency. Identification of melanopsin responses and cone inputs in Opn1mw$^R$ mice (Fig. 2) was performed using identical multispectral stimulus paradigms and analysis procedures to those used in our previous reports in LGN and pretectum (respectively 10 s and 025 Hz modulations, 10 and 24 repeats/stimulus)[29,30]. Identification of cone responses and optogenetic identification of GABAergic neurons in GAD2-ChR2 mice (Supplementary Fig. S10) was performed using identical procedures to those used for LGN and pretectum[40].

For assessment of spatial RFs and tuning properties, stimuli were delivered via an LCD display (width: 26.8 cm height: 47.4 cm; Hanns-G HE225DPB) angled at 45° from vertical and placed at a distance of ~ 21 cm from the mouse eyes. In all cases we evaluated responses at two positions sequentially, either with the monitor in 'portrait' orientation and directly in front of the mouse (subtending ~ 96° of elevation and

~ 63° azimuth across the 'binocular' visual zone) or rotated and moved laterally to subtend ~ 96° of azimuth and ~ 63° elevation of the lateral visual field (Fig. 3c). Throughout stimuli were generated and controlled via MATLAB (R2017a; Mathworks, MA, USA) using the Psychophysics toolbox v3[94,95].

For RF mapping, we used flashing light and dark horizontal or vertical bars (0.25 s duration, bars occupied ~7° visual angle and appeared across a total of 125 locations/monitor positions in pseudorandom order for 8 repeats each; Fig. 3), stimulus presentation paradigms and analysis were identical to refs. 42,43. For assessing spatiotemporal tuning, we presented sets of drifting gratings spanning 5 spatial frequencies (8–64°/cycle), 5 speeds (30–240°/s) and 8 equally spaced directions of motion. The resulting 200 different stimulus combinations were presented in pseudorandom order (13 repeats/stimulus at each of the two monitor positions). For analysis, we employed a modified $\chi^2$-periodogram approach[29,96] to test for the presence of a significant modulation in firing and quantify the percent variance (%Var) of firing rate accounted for by a periodic process matching the stimulus frequency for each stimulus. The maximal %Var detected across all tested stimuli was used to establish to optimal stimulus for each cell and the point at %Var fell below half its maximal value used as an estimate of spatial and temporal cutoff frequencies. For analysis of direction and orientation selectivity, we first quantified the peak-trough variation in firing rate (100 bins/cycle with 5-point smoothing) for each of the 8 tested directions of motion at the optimal spatial and temporal frequency. We then established the preferred direction and orientation (average of each opposing direction) as that which produced the biggest response and therefore the 'null' position (opposite direction or orthogonal orientation) and calculated DSI/OSI as: $[D_{Pref}\text{-}D_{Null}]/[D_{Pref}\text{+}D_{Null}]$. By convention[45] cells with DSI/OSI values > 0.33 were classified as direction or orientation selective (DS/OS). To assess responses to approaching/receding motion, we presented 8 different stimuli at each monitor location (light or dark spots on a grey background that either changed in size between 2° and 60° or provided matched changes in overall luminance with a fixed size of 60°). Stimulus were presented in interleaved fashion at two different speeds (transition duration of 0.5 or 0.1 s; 40 and 60 repeats/stimulus, respectively). Responses were quantified as the peak change in firing rate relative to baseline (500 ms pre-stimulus) within a 50 ms window up to 650 ms following stimulus delivery. Cells were considered motion selective when the greatest response quantified as above occurred for one of the motion-containing stimuli and the resulting change in firing rate evoked was significantly greater, and the relevant luminance matched control stimulus (unpaired t test, $P < 0.0065$ to incorporate a conservative correction for multiple comparisons). We further summarised preference via a motion selectivity index using the stimulus that produced the greatest response and the corresponding luminance control (or motion stimulus as appropriate) as: $[S_{Motion}\text{-}S_{Control}]/[S_{Motion}\text{+}S_{Control}]$ such that the resulting measure ranged between 1 (only responds to motion stimulus) to -1 (only responds to luminance control).

For functional classification of LHA neurons tested with the full stimulus set (Fig. 5), cells with DSI values > 0.33 (i.e., DS cells and occasional OS cells which also has high direction selectivity) were assigned to the direction-tuned (Group V). Cells that failed to meet the latter criteria but which displayed selective responses to approaching/receding spots, as described above, were assigned to the complex-motion selective group (Group VI). Visually responsive cells which did not fall into the motion-selective cell groups detailed above were then assigned based on responses to full-field (Mel High./Mel Low) stimuli; cells with sustained, ON- or OFF-biased transient responses were designated Groups I-III, respectively. Cells which lacked responses to full-field stimuli but which responded to one or more elements of the flashing bar, drifting grating or approaching/receding spot stimuli were assigned to the spatial-contrast selective group (Group IV).

**Histological processing.** Immediately following termination of in vivo electrophysiology experiments, mice were culled by cervical dislocation, brains removed and placed into 4% paraformaldehyde for 48 h, with subsequent cryoprotection in 30% sucrose for at least 24 h. Brains were then frozen in dry ice, and coronal slices (100 μm thickness) were taken with a freezing sledge microtome (800 Sledge; Bright Instruments, UK). Slices were then mounted to glass slides and coverslipped. Once dried, sections were observed and imaged using an upright light microscope to localise the Di-I labelled electrode tract in the hypothalamus (BX51, Olympus, UK). The resulting images were then aligned with corresponding sections form the mouse brain atlas[97] to define projected stereotaxic locations of recording sites and isolated neurons. Since there was minimal variation in projected recording locations on the rostral caudal plane (typical spanning 3 adjacent atlas sections centred on the target at -0.7 mm caudal to bregma; estimated mean deviation $n = \pm 133 \, \mu m$, SD = 81 μm, $n = 110$ recordings), where relevant, we mapped these onto a single anatomical template for display.

## Viral manipulation

**Stereotaxic injection procedures.** Mice were anaesthetised using isofluorane (Covetrus, Dumfries, UK) and placed in ear and bite bars. Iodine was smeared at the incision site and the mice were given buprenorphine (0.1 mg/kg, s.c) as an analgesic. Lubricant (lubrathal; Dechra, Shrewsbury, UK) was administered to the eyes to prevent drying out during surgery. A small incision was made in the skin to expose the skull. Using a stereotaxic frame and injection robot (Neurostar, Tubingen, Germany), we used skull landmarks (bregma and lambda) to correct for tilt and scaling. A hole was drilled at the appropriate entry site for LHA, SCN or LGN, and a glass capillary micropipette filled with virus was inserted into the brain region at the desired stereotaxic coordinates. Entry sites were chosen to ensure the injection track did not pass close to any other visual areas; for LHA injections: 0.55 mm posterior to bregma and 1.1 mm lateral (at its closest points, the injection track was > 1 mm rostral and 0.4 mm medial to tip of the visual thalamus and ~ 0.9 mm lateral to the SCN), for SCN: 0.45 mm posterior to bregma and 0.1 mm lateral, for LGN: 2.45 mm posterior to bregma and 2.45 mm lateral. Once at the appropriate depth, small volumes of virus (32–138 nl) were injected using a Nanoject II (Drummond Scientific, PA, USA). Injections were administered at a rate of 23 nl/sec and the injection needle left in place for > 5 mins before being slowing withdrawn. Subsequently, scalp incision sites were sutured and mice given topical application of analgesic (EMLA cream). Mice were supplied mashed food pellets and closely observed for 2-3 days following surgery and were allowed to recover for at least 4 weeks prior to data collection.

**Intravitreal injection procedures.** Mice were anaesthetised using Ketamine (75 mg/kg; Narketan-10: Covetrus) with Medetomidine (3 mg/kg, Domitor®; Covetrus) injected i.p. to act as an analgesic. Anaesthetised animals were placed on a heat mat to maintain body temperature during surgery. Pupils were dilated using tropicamide (1% w/v, Bausch & Lomb, UK) and then the eyes were lubricated with lubrathal (Dechra) and coverslips positioned to allow direct visualisation of the needle tips as they entered the eyes, via an operating microscope (M620 F20, Leica, Wetzlar, Germany). Using a Hamilton syringe (10 μl Hamilton glass syringe, World Precision Instruments, Worcester, MA, USA) attached to an ultrafine needle (Hamilton RN needle 35-gauge) 2.2 μl of virus was taken up for each injection. The needle was inserted at 45° through the pars plana and behind the lens into the vitreous cavity, where the virus was injected. Post surgery, the mouse was injected with Atipamezole (1 mg/kg, Antisedan®; Covetrus) to reverse the effects of medetomidine. Local analgesia was applied to the eye (a drop of 0.25% bupivacaine, Marcain Polyamp®), and a topical antibiotic (1% w/v chloramphenicol) applied just above the injection.

**Experimental paradigms and viral constructs.** For retrograde tracing of retinal projections, male and female Ai32$^{+/-}$mice (4-5 months old) received microinjections of retrograde-optimised[57] AAV (AAVrg-hSyn-Cre-WPRE-hGH, Addgene:105553) either unilaterally ($n = 5$ mice) or bilaterally ($n = 6$ mice) into the LHA or bilaterally into the LGN or SCN (both $n = 3$ mice) to drive expression of Cre recombinase in afferent neurons (and therefore the Cre-dependent ChR2-EYFP reporter in the Ai32 mice). In addition, 4 wildtype mice (female, 3 months old) received bilateral or unilateral (both $n = 2$) injections of Cholera Toxin Subunit B Alexa Fluor™ 555 Conjugate (ThermoFisher Scientific) into the LHA. A further 2 wildtype mice (female, 3 months old) received bilateral intra-LHA microinjections of AAVrg- EF1a-double floxed-hChR2(H134R)-EYFP-WPRE-HGHpA (Addgene: 20298). After 3 weeks these animals received bilateral intravitreal injections of hSyn-Cre-P2A-dTomato to drive ChR2-EYFP reporter in LHA-projecting cells. Transcardial perfusions and immunohistochemical investigations were performed 4–8 weeks following the last viral injection and 2-3 weeks following Cholera toxin injections.

For tracing of retinorecipient LHA neurons, female wildtype mice ($n = 5$; 4 months old) received bilateral intravitreal injections of AAV capable of anterograde transsynaptic transfection[56] (AAV1.h-Syn.Cre.WPRE.hGH; Addgene:1055530). After 2 weeks, the same mice then received bilateral intra-LHA microinjections of a Cre-dependent reporter ChR2-mCherry reporter construct (AAV5-EF1a-double floxed-hChR2(H134R)-mCherry-WPRE-HGHpA, Addgene:20297). Transcardial perfusion and immunohistochemical investigations were performed 5 weeks after the last viral injection.

For chemogenetic manipulation of LHA retinal input, wildtype female mice ($n = 48$, 2-3months old) received bilateral intra-LHA microinjections of AAVrg-hSyn-Cre-WPRE.hGH. After 3 weeks, the animals received bilateral intravitreal injections of AAVs encoding either a Cre-dependent excitatory DREADD ($n = 18$ mice; AAV2-hSyn-DIO-hM3D(Gq)-mCherry, Addgene: 44361) inhibitory DREADD ($n = 14$ mice; AAV2-hSyn-DIO-hM4D(Gi)-mCherry, addgene:44362) or a control fluorescent reporter ($n = 16$ mice; AA2-hSyn-DIO-mCherry, Addgene: 50459). Experimental procedures (described below) occurred at least 3 weeks after intravitreal injection, with transcardial perfusion and immunohistochemical investigations occurring >1week later. For the analysis presented in the manuscript, nine mice ($n = 3$ receiving Gq-DREADD, $n = 2$ receiving Gi-DREADD and $n = 4$ receiving control vector were excluded from analysis due to failed/mistargeted injections).

## Immunohistochemical investigations

**Tissue preparation and Immunohistochemistry.** Mice were overdosed with urethane (~5 g/kg) and transcardially perfused with 4% paraformaldehyde (methanol-free), brains and whole eyes removed and then stored in 4% PFA overnight. Brains were transferred to a 30% sucrose solution for cryoprotection and then were sliced at 40 μm using a freezing sledge microtome (800 Sledge; Bright Instruments) for subsequent immunohistochemistry. Whole eyes were transferred to phosphate-buffered saline (PBS) until retinal dissection under dim light with the aid of a dissecting microscope and Vanna scissors (World Precision Instruments, FL, USA). Retinas and brain sections were then permeabilised in 1% TritonX-100 solution in PBS (1% PBT; 2 x 20 mins) and then blocked in 10% appropriate animal blocking serum (either 10% Normal Donkey Serum, 10% Normal Goat Serum, or a combination 5% Normal Donkey Serum with 5% Normal Goat Serum, in 1% PBT) for 2 h at room temperature. Tissue was then incubated at 4 °C overnight in primary antibody solution (2.5% appropriate animal blocking serum in 0.2% PBT). The primary antibody was washed off using 0.2% PBT (3x20mins for retina or 3x10mins for brain sections) followed by addition and incubation of the secondary antibody solution (5% appropriate animal blocking serum in 0.2% PBT) for 3 h at room temperature. The secondary antibody was then washed off in PBS

(3 x 20 mins or 3 x 10 mins as above) before mounting retinas and brain sections on slides with Prolong Gold (ThermoFisher Scientific, UK) and coverslipping.

For retinal immunohistochemistry in Ai32 (and intersectionally labelled WT) mice, retro-labelled RGCs were detected with chicken anti-GFP (A10262, 1:1000; Abcam, MA, USA) and Alexa 488 goat anti-chicken (A32931, 1:500; ThermoFisher Scientific, MA, USA). For co-staining we used either rabbit anti-S opsin (AB5407, 1:500; Sigma Aldritch), rabbit anti-melanopsin (AB-N39, 1:500, Stratech, Ely, UK) or rabbit anti-CART(H-003-62, 1:1000; Pheonic Pharmaceuticals Inc, CA, USA) with Alexa-555 donkey anti-rabbit (A21428, 1:500; ThermoFisher Scientific). Where retinas were triple-stained, we used chicken anti-GFP and rabbit anti-melanopsin as above, with the addition of goat anti-CART (AF163, 1:1000; Bio-techne, MN, USA) and Alexa-647 donkey anti-goat (A21447, 1:500; ThermoFisher Scientific).

For anterograde tracing in brain tissue, retinorecipient LHA neurons were detected with chicken anti-mCherry (ab205402, 1:1000; Abcam) and Alexa-555 goat anti-chicken (ab150174, 1:500; Abcam). Alternate brain sections were co-stained for rabbit anti-VGAT (131-002, 1:1000; Synaptic Systems, Goettingen) or rabbit anti-VGLUT2 (AGC-036, 1:1000; Jerusalem, Israel) with Alexa-488 donkey anti-rabbit (A21206, 1:500; ThermoFisher Scientific). For studies involving chemogenetic manipulation, retro-labelled RGCs were detected with chicken anti-mCherry (ab205402, 1:1000; Abcam) and Alexa-555 goat anti-chicken (ab150174, 1:500; Abcam). FOS expression was detected with Rabbit anti-cFos (ab190289, 1:1000; Abcam) and Alexa-488 Donkey anti-Rabbit (A21206, as above).

**Imaging & analysis.** Imaging of retinal wholemounts used either Andor Dragonfly200 spinning disk inverted confocal (Andor Technology, Belfast, UK), or Leica SP8 confocal (Leica, Wetzlar, Germany) microscopes. In both cases, the entire retina was captured using tiling features along with a motorised stage (20X objective or 40x objective with 0.75X confocal zoom, respectively for Andor and Leica microscopes). At each position, a Z-stack (0.8 μm width) was generated, capturing the entire ganglion cell layer and inner plexiform layer and (as appropriate) photoreceptor layer. To prevent bleed-through between fluorescence channels, images were collected sequentially with each laser/filter. Laser power was set for each retina to maximise intensity range whilst minimising saturation. Brains were primarily imaged using a 3DHISTECH Panoramic 250 flash slide scanner with a 20x/0.8 Plan Apochromat objective and either FITC and/or TRITC filters (3DHISTECH, Budapest, Germany). For analysis of mCherry and VGAT/VGLUT2 colocalization, imaging was performed using an Andor Dragonfly200 spinning disk inverted confocal as above, targeting regions of the septal complex, habenula and hippocampus containing mCherry-labelled projections.

Subsequent analysis of retinal wholemounts was primarily performed on confocal Z-stacks in ImageJ (v1.52a; NIH, MD, USA), to identify retro-labelled cells and, as appropriate, quantify soma size, dendritic field diameter and/or co-expression with other relevant immunohistochemical markers. Estimates of dendritic field asymmetry for labelled neurons used a 90° rotating window centred on the cell body and with a radius corresponding to the distance to the most distal visible dendrite. Asymmetry was then defined by comparing labelling intensity in the densest quadrant with the quadrant 180 degrees opposite ($[I_{Max}-I_{Opposite}]/[I_{Max}+I_{Opposite}]$). Similarly, estimates of overall retinal labelling asymmetry used a rotating window centred on the optic disk that encompassed 25% of the retinal area to identify the quadrant with the greatest number of labelled RGCs. For analysis presented in Supplementary Fig. S5t, one of 6 retinas from SCN-AAVretro-Cre injected Ai32+/- mice was excluded since damage to a portion of the retina prevented reliable assessment and 1 of 8 retinas from LHA-ChTxB-injected WT mice was excluded since only one labelled RGC was observed. For analysis of colocalization with c-Fos in

chemogenetic studies, experimenters were blind to experimental group until completion of analysis. In all cases, for subsequent display, Z-stacks were converted to max-projections.

For brain anterograde tracing, we utilised CaseViewer software (v2.4; 3DHISTECH) to collect high-resolution images of regions of interest, which were then aligned with relevant sections of the mouse stereotaxic atlas[97] to localise anterograde labelled soma. As appropriate, mCherry-labelled fibre projections were then traced manually using a Cintiq16 graphics tablet (Wacom, Saitama, Japan). For analysis of mCherry and VGAT/VGLUT2 colocalization, confocal Z-stacks were imported to Matlab and analysed using custom routines. In brief, the Z-stack was divided into 70 × 70 μm (40 μm depth) subfields, with the 10 resulting subfields containing the highest proportion of mCherry labelled voxels selected for subsequent analysis. For each subfield we then quantified the proportion of mCherry labelled voxels that co-localised with VGLUT2/VGAT before and after local rotation of the VGAT/VGLUT2 channel of that subfield by 90°. For each image, we then tested for significant co-localisation by paired $t$ test between the % colocalization observed in the original vs. rotated subfields. This analysis was performed on a total of 23 matched regions of interest from VGLUT2 and VGAT co-stained sections (from $n = 5$ mice; $n = 2$–8 regions imaged/mouse).

For analysis of c-Fos expression in the brain, cell counting was performed manually by experimenters blind to the experimental group. For hypothalamic c-Fos expression, cells were labelled directly in CaseViewer software across hypothalamic sections corresponding to 0.22 to 1.46 mm posterior to bregma (typically 7 sections/animal). Images were then aligned with appropriate stereotaxic atlas sections, and the locations of manually labelled cells used to quantify expression density using Matlab routines, as described previously[41]. For analysis of c-Fos expression in the visual thalamus, superior colliculus, septal complex, habenula and hippocampus, images of regions of interest (typically $n = 12$, 8, 4, 8 and 8 hemisections/animal, respectively) were exported to ImageJ. Expression density was then determined (with experimenters blind to experimental group, as above) for manually counted cells and manually drawn boundaries (following the overt contours of those anatomical regions). In all cases, we used the mean expression density across all analysed sections/hemisections for each animal for subsequent analysis.

## Chemogenic manipulations

**Behavioural testing.** After at least 3 weeks recovery from viral injections (see above), mice (female, 3-4months old) were taken from the colony room during the mid-projected day (ZT3-8) and received an i.p injection of 0.1 mg/kg clozapine (CLZ; Tocris Bioscience). Throughout, mice were tube handled to reduce stress. After 45 mins mice were introduced to the experimental arena. Full details of the experimental arena and recoding set up have been reported previously[60]. In brief, mice were placed in a 30x30cm square open field arena under a constant background illumination (White LED, $1.94 \times 10^{13}$ melanopsin effective photons/cm²/s), with additional infrared light supplied to facilitate imaging via 4 programmable cameras (frame rate = 15 Hz; Chamaleon 3, Teledyne FLIR, OR, USA). White noise (64 dB(C)) was played continually to mask noises from the surrounding environment. Mice subsequently experienced 14 blocks of trails, each of which consisted of a sudden noise (1 s duration), a diffuse light flash (~150-fold increase in irradiance for all photoreceptors) and an overhead looming stimulus. Stimulus order was randomised across blocks, with a ~70 s interval between each stimulus (~1 h total experiment duration). Between blocks, auditory stimuli were randomised to entail either a pure tone C6 at 102 dB(C) or a white noise 89 dB(C), and the looming stimuli were randomised between a black looming disc (87% Michelson contrast) and looming grating disc (spatial frequency: 0.068 cyc/deg. Michelson contrast: 94% Black vs white). For the final 7 blocks in each experiment, mice were provided access to a shelter (a

red plastic tube), placed at the edge of the arena. Video images were captured for both the 10 s preceding and post-stimulus presentation. In initial analysis we did not find any qualitative differences in multidimensional behavioural responses (see below) to the two auditory or the two looming stimuli so data were pooled for analysis (2-way RM ANOVAs for multidimensional behavioural profile during 1 s post-stimulus window; **Loom**: Behavioural component, $F_{2.2,79.0} = 458$, $P < 0.0001$, Stimulus type, $F_{1,37} = 2.9$, $P = 0.10$, Behaviour X Stimulus, $F_{1.8,66} = 1.65$, $P = 0.20$; **Sound**: Behavioural component, $F_{2.6,97.0} = 493$, $P < 0.0001$, Stimulus type, $F_{1,37} = 0.1$, $P = 0.76$, Behaviour X Stimulus, $F_{2.0,76} = 1.79$, $P = 0.17$). Similarly, we did not see any overt qualitative changes in behaviour following inclusion of the shelter so we pooled data from across all experimental blocks for final analysis (2-way RM ANOVAs for multidimensional behavioural profile during 1 s post-stimulus window; **Flash**: Behavioural component, $F_{2.2,80.3} = 589$, $P < 0.0001$, Stimulus type, $F_{1,37} = 0.13$, $P = 0.13$, Behaviour X Stimulus, $F_{1.7,61} = 1.82$, $P = 0.17$; **Loom**: Behavioural component, $F_{2.5,93.3} = 609$, $P < 0.0001$, Stimulus type, $F_{1,37} = 0.01$, $P = 0.91$, Behaviour X Stimulus, $F_{1.9,73} = 1.06$, $P = 0.35$; **Sound**: Behavioural component, $F_{2.9,106} = 522$, $P < 0.0001$, Stimulus type, $F_{1,37} = 0.59$, $P = 0.44$, Behaviour X Stimulus, $F_{2.8,104} = 1.27$, $P = 0.28$).

Tracking of body landmarks was performed offline for each individual camera by using DeepLabCut[98]. The coordinates of landmarks across the four cameras were then triangulated to obtain an initial 3D reconstruction. Outliers and missing data were then corrected by fitting a Statistical Shape Model, also estimated from the data, as described previously[60,99]. In brief, for each frame the mouse pose (X) was described by the equation:

$$X(t) = \left( \bar{X} + \sum P_i b_i(t) \right) R(t) + T(t) \tag{1}$$

Here, $t$ represents the time of the current frame, $X$ the body landmark coordinates, $\bar{X}$ the mean body pose coordinates, $P_i$ the mouse eigenposes, $b_i$ the parameters that track changes in body shape and R and T the rigid (rotation and translation) transformations representing the animal's position in the arena. From this refined 3D reconstruction 9 behavioural features were extracted as in ref. 60. These comprised 3 postural measures: Body Rear (vertical distance between mouse neck and tail base), Body Elongation & Body Bend (respectively deformations in mouse body shape on the XZ and XY planes, relative to the mean body shape). The remaining features captured aspects of the animals' movement: ΔRear, ΔBody Elongation, and ΔBody Bend (respectively, the velocity with the relevant parameters change from the previous frame); Locomotion (velocity with which the XY position of the mouse body changes relative to the previous frame); Body Rotation (Frobenius distance between the rotation matrices describing mouse position on the current vs. preceding frame); 'Freeze' (negation of the XYZ distance between all mouse body markers, relative to preceding frame). The latter measure provided a continuous measure of mouse immobility, bounded at 0 (no detectable movement). Since freezing is generally considered as an all or nothing response, we converted this to a binary measure, with a summed change in the 3D position of mouse body parts less than 2 mm scored as 1 ('Freeze'), otherwise 0 ('No Freeze'). As such, when averaged across trials we obtained a measure of Freeze probability as a function of time relative to stimulus onset.

For analysis of spontaneous behaviour, we pooled the 10 s pre-stimulus epochs across all trails/stimuli for each animal, as well as calculating the mean distance from the centre of the area (in the X-Y plane), and compared each behavioural component between experimental groups by one-way ANOVA. For analysis of stimulus-evoked responses, for each animal, we calculated the average for each behavioural component following each stimulus (1 s window starting 2 frames after stimulus onset when behavioural responses first emerged), after or before subtracting the mean values in the 2 s epoch immediately preceding the stimulus. In both cases, the resulting datasets for each behavioural component were analysed across stimulus types by mixed-effects ANOVA (with stimulus-type as within-subjects and experimental group as between-subjects factors). In additional analysis, we also calculated the duration of stimulus-driven freezing (whereby an event was initiated by the first occurrence of freezing during the 1 s analysis window specified above and terminated by two consecutive frames without freezing). Analysis presented in the manuscript derives from $n = 38/39$ mice with appropriate viral construct expression tested in this paradigm (see 'Viral Manipulations' section). One (Gq-DREADD expressing) mouse was excluded from analysis due to a technical error with stimulus delivery, resulting in final group sizes of $n = 14$ Gq-DREADD, $n = 12$ Gi-DREADD and $n = 12$ control vector expressing mice.

**Post hoc validation.** Following behavioural testing, mice were individually housed in light-tight cabinets under a 12:12 h LD cycle. On the day before the experiment, commencing from the normal start of the dark phase, mice were then placed into constant darkness. During the early-mid projected day (~pZT 4.5), mice were injected with 0.1 mg/kg of clozapine i.p. (under v. dim red light) and rehoused in darkness. After 90 mins mice received an overdose with urethane i.p (dose) using a dim red head torch, and were transcardially perfused before preparation for retinal and brain immunohistochemistry as described above.

### Statistics
Throughout, data are presented as Mean ± SEM, unless otherwise stated. Details of initial data processing and derived measures are provided in the relevant methodological sections above ('Visual stimuli and snalysis', 'Imaging and analysis', 'Behavioural testing'). All higher-order analysis was performed in Graphpad Prism (v9.3.1, GraphPad Software, CA, USA), using appropriate (two-tailed) statistical tests for two or more groups of categorical or numerical data (including repeated measures where relevant), as reported throughout the manuscript.

### Reporting summary
Further information on research design is available in the Nature Portfolio Reporting Summary linked to this article.

## Data availability
The data underlying all quantitative analysis that supports the findings of this study are provided in the Source data accompanying this paper. Other raw data is available from the corresponding author on request. Source data are provided in this paper.

## Code availability
Code for multi-dimensional mouse behavioural analysis is available at Zenodo and is maintained on GitHub[99].

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

## Acknowledgements

This work was supported by grants from the Biotechnology and Biological Sciences Research Council to (BBSRC, UK) to TMB and RJL (BB/S015272/1) and the Leverhulme Trust to TMB (RPG-2022-168). The authors thank Nina Milosavljevic, Marina Gardesevic, Dean Stewart and Lauren Walmsley for technical assistance with viral injections and Peter March, Roger Meadows, Steve Marsden, and James Bagnall for technical assistance with microscopy.

## Author contributions

T.M.B., J.W.M., R.S., R.J.L. and E.T. designed the experiments. J.W.M., E.T., T.M.B., C.W., A.W., W.F. and M.P.H. performed the experiments. J.W.M., E.T., A.E., R.S., T.M.B., C.W., A.W., W.F. and M.P.H. analysed the data. T.M.B. wrote the paper with assistance from all the authors.

## Competing interests

The authors declare no competing interests.
