## [Transparent Peer Review file · Nature Communications]

A lateral hypothalamic region supporting diverse visual processing and modulation of visually-guided behaviour

Corresponding Author: Professor Timothy Brown

Version 0:

Reviewer comments:

Reviewer #1

(Remarks to the Author)

The manuscript by Moulard et al. tests the role of lateral hypothalamic area (LHA) on visual processing and visually-guided behavior. By large-scale electrophysiological recording, viral tracing and chemogenetic manipulation in mice, the authors firstly describe the visual processing in LHA neurons in many aspects, revealing that LHA neurons are highly selective for spatiotemporal contrast. It is interesting as hypothalamus is thought to mainly process visual information in a coarse-tuned manner.

The authors also attempt to reveal the role of LHA in neuronal circuits of visual threat detection. Overall, the data quality appears good, and results are mostly consistent, however some parts need additional experiments, presentations and/or classification to be fully convincing and to warrant publication.

Major:

1. The authors showed that most LHA neurons exhibit small receptive fields and sensitive to moving stimuli, and attributed this to retino-LHA projection. However, the recorded visually-evoked LHA neuronal activity could also originate from visual thalamus such as dLGN and SC, or others, and the large-scale electrophysiological recording could not exclude this indirect visual pathway from retina to LHA. In the result, the authors had employed retrograde tracing from LHA. It is also necessary to examine whether other visual thalamus and/or visual cortices are targeted, and also their potential role in the visual processing of LHA neurons.
2. In line 84, the authors declared that “the majority of neurons (n=25/44) showed only transient increases in firing”. However, the percentage of this group of neurons is around 57%. Maybe “over half” is more accurate to describe here than “the majority”.
3. The figure 7 targeted the LHA-projecting RGCs via a unilateral injection of AAV2retro-Cre into the LHA, but no data or evidence of injection site has been shown in the manuscript. Although AAV2retro-Cre does not express fluorescence, injection of AAV combined with fluorescent dye could be an appropriate way to label the injection site. Considering that the different parts of nucleus may receive projections from different retinal quadrants, the result that ‘LHA-projecting RGCs are highest dense in ventrotemporal and lowest in doronasal retinal quadrants’ may need to be corrected by the injected sites in LHA.
4. Figure 7f needed to be plotted by both single and merged channels for better visualization.
5. In the figure 7f, the dendritic field of most EYFP-positive RGCs look asymmetric. Are most LHA-projecting RGCs are asymmetric in their dendritic field? The classification of different types of RGCs need to be done.
6. Overall, some figures are lack of readability and are always presented too crowd to distinguish different parts. For example, in the figure 10a, diagram of 3D construction is too crowd and the captions are too close to other diagrams. A more concise and readable version of figures is needed.

Minor:

7. In the figure 5C , the dashed line had not been indicated in the legends
8. I don't quite understand what is the significance of comparing the soma size of RGC projected to LHA and SCN in Figure 7.GH. It seems irrelevant to the main story and unnecessary to appear.
9. Figure 7 H should to be h.

10. Figure 7b, the V is misplaced under the arrows. The EYP should be EYFP (also in Figure 5b).

11. In figure 10, the graphic diagram did not show how the AAV were delivered. Although the method is similar as figure 9, the authors should at least state in the figure that they manipulate the neuronal activity of LHA-projecting RGCs.

Before resubmission the authors should proofread it carefully, especially for the figures, and the editor should also do so before publication.

Reviewer #2

(Remarks to the Author)

This is a very comprehensive study on the visual properties of the LHA visually responsive cells. Not only are the findings of these cells unexpected, but the responses of these cells are very well documented, with described inputs and outputs and functional manipulation. The study is well executed, using good methodology. The amount of data is impressive. Some issues, however, need to be addressed before recommending for publication in Nature Communications.

One problem with the wording in general is describing LHA as a visual centre. Having, for example, movement activated cells in the visual cortex wouldn't make it a motor centre. The authors do not compare the number of total recorded cells in the visually responsive electrode channels to the number of visually responsive cells. In addition, these visually responsive cells were only found in one part of the LHA. Authors should consider using different language here. As a related point, the results should summarize in one place the n of mice in each of the experiments, the total n of recorded/isolated cells, and the total numbers of visually responsive cells in recordings. The proportion of these cells compared to other recorded cells is not entirely clear.

Line 43 - Sensory responses (including visual) have been demonstrated in orexin neuron subpopulation of the LHA: see Fig 5. <https://doi.org/10.1016/j.pneurobio.2020.101771>

Also for review see <https://www.sciencedirect.com/science/article/pii/S0031938420302663>

Line 55 - "employing mice where native M-cone opsin is replaced by the human L56 cone opsin (Opn1mwR)" I don't see a short description where the reasoning for this is given. A quick description in the methods or results, on how this exactly helps elucidate the different photoreceptor contributions to visual responses would be useful, especially for those not very familiar with the field.

It would be useful from the authors to report latencies to the described visual responses, such that they may be compared to latencies observed in other visual processing areas.

Line 161, figure 2,S2 - Authors need to either justify the usage of this period or employ a different method. Fig S2d seems to show some differences also in the acute period. Maybe choosing the peak values rather than the mean of the arbitrarily chosen 250ms period after stimulus onset would be less susceptible to subjectivity in analysis. In addition, the results of the test in figure S2d and others are not well reported. Precise p and t-values should be given.

Line 187 - The melanopsin responses are described as 'sluggish' in line 159 to describe the result of no difference in acute portions of the responses in S2d, but here fast melanopsin response dynamics are suggested. Authors should explain / or suggest how melanopsin response kinetics can display both sluggish and fast kinetics, with reference to the method used in each experiment.

Are the cells in the LHA only representing a small portion of the visual field as shown in 4C or have the authors not investigated the wider area. Authors should discuss the relevance of this small coverage of visual field for the presumed purpose of LHA-based vision. Authors look whether S- and L- cone distributions mirror those of the distribution on the retina (paragraph starting at line 293), but it isn't clear why more of the visual field wouldn't be represented here, perhaps a point for discussion.

In the experiment under 'Identity of retinorecipient lateral hypothalamic neurons' starting at line 454 the authors demonstrate, using tracing methods, that the retinorecipient neurons in the LHA are exclusively glutamatergic. However, in the following paragraph starting at line 470 they verify in-vivo that none of the visually responsive neurons are GABAergic. Having found co-localisation with glutamatergic markers, why didn't the authors instead/in-addition confirm with optogenetic stimulation/suppression that the visually responsive neurons are glutamatergic? This seems like a logical continuation and it is strange that it is omitted.

In some places in the text, the authors report p-values without reporting the test or the test statistics. For example, line 508, line 517, line 521, etc. The authors should stay consistent with statistical test reporting, either giving a full report in text, or omitting the information in text, just stating that there is/isn't a significant difference and reporting the full statistical report of the test in the associated figure legend. Authors should make this consistent across the entire manuscript and make sure there are full reports for each test (some are partial/missing).

Starting in line 523 the authors describe finding no c-Fos expression differences due to chemogenetic manipulation, so they suggest a modulatory rather than driving role of these projections to the hippocampus. This seems quite vague, as modulatory projections should also impact activity in the region during the 90 minutes before the animal was culled. The authors should elaborate or leave out this part.

For the data in Fig 10. the authors run a mixed-effects ANOVA on the behaviour and DREADD effects. Adding so many

behaviors in the test, and indeed, using it as a variable, can dilute the effect of p-testing via multiple comparisons. As such. For example, in Fig 10g, body bending seems impacted by DREADDs even though the tone is used instead of visual information. This might indicate that actually there is some kind of integration of visual and sound information in these cells, since blocking (or activating) their visual inputs only can still have an impact on auditory behavioural responses. Authors should consider maybe running a stimuli x DREADD mixed model for different behaviours (which also may have a weird interpretation) or running a series of one-way (repeated-measures) ANOVAs on individual behaviours to determine the effect of DREADDs. Otherwise, they should do more to justify including behaviour as a variable. There are other problems there (behavioural measures are not independent – an increase in locomotion would necessitate a decrease in freezing, for example), and there are probably other dependencies. In a mixed-effects model, there should be a mix of repeated measures and between subjects tests. Here I assume, although it is not explicitly stated, that behaviour was treated as a between subject variable, which it is not because it is in the same mice, and the measures are not independent of each other.

Figure 10 legend or the corresponding methods section also doesn't explain the difference between delta measures and regular measures and why some were included as deltas and some not. For example, what is delta(rearing)? Is it the period that the animal is changing from another behaviour to rearing or from rearing to another behaviour, or both, or something else entirely? If it is only behavioural transition to or from behaviour, a simple count would be better.

In line 1006 of methods, authors describe no qualitative difference in responses to two different types of auditory or looming stimuli. There is no description in what these qualitative measures could be and why quantitative measurements were not used, given they were measured regardless. Same applies for the inclusion of shelter in the experiment.

The way that behavioural variables in Fig. 10 were operationalized seems very strange. First of all, using quantile normalization would overrepresent behaviours that don't happen so often as the one that happens the most. Second of all, if each behavioural feature was normalized in the 0-1 range, as described in the methods, how is it that some values in Fig 10 are negative? Was there an additional subtraction of the baseline, and if so which period was used to assess this baseline? If yes, it would only make sense to baseline subtract per trial, but then if the animal was frozen before a trial, there would be a massive skew toward negative if the animal moves after onset (for example). These measures are not so clear even with reference to the cited paper developing this method (<https://www.sciencedirect.com/science/article/pii/S0960982220313385>). Is the output of the measure a classifier of behaviours or does it include the probability that a certain behavior is occurring at a certain moment in time? Which behaviours are then allowed to co-occur (eg. Movement and freezing are exclusive of each other, but movement and body bend are not)? Authors should explain much more carefully how these measures were derived, pre and post processed, if they are to use them. Without this, the validity of the figure is difficult to understand currently.

Minor:

Line 40 – no definition of peri-SON before using abbreviation

Line 284 – vary instead of varying

Line 508 – 'control of Gi-DREADD groups' should be or? Probably typo

Line 580 – visual input written twice

Line 685 - vis instead of via

Line 697 – DFERADD

No legend for panel c) in Figure 4.

Fig 10. - Behavioral instead of behavioural in many places.

Fig S1h – why is it the only figure where it says 405nm? Typo?

Reviewer #3

(Remarks to the Author)

In this study, Moulard et al. examine the role of the lateral hypothalamic area (LHA) in visual processing and behavioral control, which has traditionally been considered separate from other visual pathways. Using a variety of techniques, they provide the first description of the visual processing capabilities of the LHA and suggest that this activity is involved in the regulation of behavior. They found that while some LHA cells respond to overall light levels through melanopsin-driven responses, most are highly selective for spatiotemporal contrast, including motion detection. The authors found that the LHA also displays a retinotopic organization, and its neurons project to key behavioral control centers, influencing responses to environmental cues such as flashes of light and looming threats. The authors conclude that this establishes the LHA as a critical visual center for managing behavior and responding to environmental threats.

While the quality of the data presented is solid for the physiology, I have some concerns about the clarity of this part. In addition, I have several open questions that challenge the interpretation of the results, particularly for the second part of this paper regarding anatomy and behavior.

Major

Physiology of LHA

The paper is well written. However, the presentation of the figures can be optimized. The main messages are (i) diversity of responses in LHA, (ii) anatomical characterization of RGC & RGC receiving LHA projections, and (iii) behavioral analysis.

For the first part concerning the visual responses in LHA, the message can be streamlined. In my opinion, it would help to make the message more compact and move the less relevant information to the supplement and compress Figs.1-6. Since the authors have recorded a pallet of stimuli, each figure provides an independent message. However, I think that it would be more adequate to analyze them together, providing a functional characterization of visual responses, rather than making one figure for each of the stimuli used. This would provide an overall picture, which can't be done by the reader if it wouldn't reanalyze the data. A nice example could be the paper by Baden et al. 2016. I don't ask to make the same analysis but to compress the data and look at the interdependencies across stimuli to give a functional classification of the LHA.

The LHA is a long structure also in the anterior-posterior axis, but I couldn't find any direct assessment of where the probe was positioned. While the authors have provided characterization on the dorsoventral and mediolateral axis (suppl. Info), it would be relevant to add information on the anterior-posterior axis too. Would this reveal a retinotopic map? This would reveal if there is any bias in the recording position. This is relevant also for the anatomy. While some information on the anterior-posterior (rostral-caudal) axis is provided for the c-fos counting and data on the anterograde labeling experiments, nothing is presented for the recording site position nor injection sites apart from "desired stereotaxic coordinates".

Anatomy

The authors used serotypes that do not spread well in the retina, especially AAV1, which are not good serotypes to infect the retina (at least in our hands). They have strong biases in their infections, spreading very little beyond the infection side and having, at least in our hands, a bias towards the blood vessels. This can be changed with AAV2, specifically the AAV2.7M8 strain, the latter has been developed for that particular purpose. So the level of anterograde labeling will be very likely biased and low. It would be important to make a control to show the level of infectivity in the retina by infecting a reporter mouse line (e.g. Ai32) with AAV1 and AAVrb to test for infection biases in the retina. In addition, the retina processes across the LHA should be characterized using classical CTB injections (as a ground truth), which then can be compared with the projection patterns using AAV in the LHA

Another implicit assumption, at least as the data is presented, is that the projections from the retina to the LHA are specific. They may well be axonal projections from cells that project to multiple brain areas, including the LHA. This would strongly bias the interpretation of the c-fos, behavioral and anatomical results. For example, for the anatomical analysis of LHA projections: after AAV-Cre infection, all RGC recipient areas could be infected. Subsequent secondary infection by AAV-ChR2-cherry could therefore be non-specific (normally many areas that go through the injection site are likely to be labeled). This could occur if the injection site to the LHA crossed other retinal recipient layers. It would be advisable to confirm that this is not the case by whole-brain confirmation of off-targets. In addition, modulating LHA RGC projections, if they are due to collateral axons, would not allow determining the role of LHA in any behavior, since any perturbation of the LHA-projecting cells will ultimately influence many areas at once.

Most concerning, however, is the low resolution of the images shown in Figure 8. Anatomical images are pixelated. I can't judge the data well at this resolution, but it seems to me that fixation is poor, resulting in autofluorescence that looks like processes. The poor fixation seems to be evident in the HC image (see holes in the HC) and spots across the HC and strong autofluorescence in the lower parts of the image. The processes in the HC might be autofluorescence of blood vessels, which sometimes can be confused with processes. Not certain about this, but it would fit the poor fixation (holes nearby). Moreover, the example sections show single cells in the LHA, but the septal region appears to be filled with processes. This is inconsistent by orders of magnitude. I'm not an IHC expert, but in my layman's eye, this should be evaluated and the quality and resolution should be improved to be able to judge Fig. 8.

Behavior

Similarly, the DREADD experiments may also be biased by the infections. Therefore, it would be advisable to show (i) where in the retina the RGC are labeled (because of stereotactic retro-AAV infections in the LHA and biased AAV serotypes in the retina). and (ii) clear examples with labeled DREADD processes across the LHA. In addition, one needs to confirm that the RGC projections are specific to the LHA, otherwise the authors cannot conclude any behavioral effects. This is required for the anterograde labeling, as well as for the retrograde labeling experiments. At least the c-fos labeling suggests that RGCs that project to LHA also project to other areas. If you observe the SCN c-fos labeling it seems drastically different when excitatory or inhibitory DREADDs are used. The position of the cells seems to change depending on whether excitatory or inhibitory DREADDs or controls are used (Fig. 9d, see the distribution of c-fos expression in the SCN). While the quantification shows something else (Fig. 9e), the raw data presented suggests that this is not consistent with the quantification. I also find it intriguing that outside the LHA more c-fos labeled cells can be seen. How do the authors interpret these results? This is key, how would the authors determine the relative role of the behavioral effect between SCN and LHA (or other retinorecipient areas)? In my view, the most parsimonious explanation would be that the manipulations alter the SCN and thus one would observe changes.

Finally, to examine if the behavioral effects are specific, as stated in this manuscript, it is important to look at the absolute data, not the non-normalized data, and to look at the statistics again. The normalized mean can hide a lot of interesting dynamics and I would suggest presenting all data points. One possibility is to plot the absolute speeds as a heatmap, but I leave it to the authors to find a convincing way to describe their results. There are interesting complexities hidden by normalization and averaging. Also, I wouldn't call the behavioral responses "high-dimensional", particularly if the normalized mean of, e.g., locomotion, is reported. Also, it is strange to plot freezing as the way it is presented. Freezing is normally defined as a time of absolutely no behavior. Thus, if the normalized values are presented, it is very difficult to understand these plots. What is a "positive" value for freeze, for example?

Minor

I won't judge the details in the text and structure, as the text is well written, but I would encourage the authors to extract the interdependencies in physiology. This would compress the figures, and the text will probably need major changes.

- It is not clear why the authors start with their *Opn1mw* mice in the first figures. I would present WT animals and only use the *Opn1mw* data to specifically address the question of melanopsin cells in LHA. This is fine, but I think it would be easier to build a case.

- Fig. 1. Given the small number of cells, instead of presenting the average of 1 cell, I would like to see the raster plot of a cell with a different presentation and then summarize the average as a heat map sorted by "transientness", this will provide a better understanding of the variability of responses (for LHA and SCN). I also prefer to see firing rates, more than changes in rate.

- Fig. 4. The authors report a bias towards frontal RF. It is not clear to me whether this bias is due to the recording position.

- Fig. 6. The standard way to show changes in firing rate would be to show the absolute rates, not the relative change. Overall, the message would be the same, but it would allow the reader to better scrutinize the data.

- Fig. 7b, there V is misplaced in the dorso-ventral schematic, and it is not clear why the cell number count is needed.

- Fig. 7 GH. It seems that the distribution of dendritic size of mel + cells is different between SCN and LHA, can the authors comment?

Reviewer #4

(Remarks to the Author)

Version 1:

Reviewer comments:

Reviewer #1

(Remarks to the Author)

After reviewing the authors' revisions in response to my initial comments, I find their modifications satisfactory. The revised manuscript has adequately addressed my concerns, and I support its acceptance in its current form for publication in Nature Communications.

Reviewer #2

(Remarks to the Author)

The authors have sufficiently addressed all of the raised concerns.

Reviewer #3

(Remarks to the Author)

Mouland et al. did a good job revising some of my previous comments. The presentation and results of the visual response properties of LHA are convincing and, in my opinion, represent the main finding of this work. The authors did thorough work, and I applaud them for it. I don't have concerns about the data presenting the visual properties of LHA. However, I still have concerns about the interpretation and data in the second section regarding the indication that the response properties are specific retinal projections to the LHA and that this activity is relevant to complex behavioral control. Since these are core points the authors want to bring up, as seen in the title, I have listed a remaining weak points.

First, regarding direct projections from the retina, the paper implicitly claims that ~10% of the neurons are visual and that they inherited their properties from retinal cells and not from other pathways. The direct influence is inferred by specific projections to the LHA, mainly through retrograde tracing experiments that do not show collateral projections to other visual areas, as well as the response dynamics and measured receptive fields match the position of the retina. These are shown to correlate with c-Fos labeling.

However, as with their mCherry results, long-range projections may require more time to become apparent. I recently

learned that Chr2 is not actively transported and thus requires time to reach distant axon terminals because it passively diffuses through the Golgi apparatus. Therefore, the absence of these processes in other areas could also be due to this. Also, it is unclear whether the right sections are being tested, at least for the SC. It seems that these sections are posterior, which would represent a different retinotopic space. The resolution of the images is too low for a thorough analysis (they are too pixelated when zoomed in, Fig. 6I and supplement). I believe the authors, but I would carefully check all rostral and caudal slices in the SC. Second, the argument that the responses are fast is weak, particularly if the timing of responses to visual stimuli onset is not tested and compared to other visual response areas. I'm not asking to make any new experiments, but just want to mention that having recorded in the LHA with DREADDs in the retina, would have allowed to specifically show which functional aspects are inherited and which not from direct retinal projections and having done pseudorabies tracing from the LHA with reporters that are generally used to visualize projections, would have been the clearest way to show the specificity to the LHA. Thus, currently I am still not yet convinced that these projections are highly specific. However, I would be fine with de-emphasizing this point and rewording the paper a bit. The LHA clearly has visual response cells that have been thoroughly characterized, and I don't think it's relevant to state that visual activity is only acquired via retinal projections or if there are other pathways that might influence the activity too. So this, I think, can be reworded.

I'm more cautious about the behavioral experiments. They need more thorough analysis, particularly of effects that seem to have statistical significance. For example:

The statistics are all normalized to the change, but this could be due to minor changes in overall locomotion activity. The heat maps showing the change. In my opinion, it would be best to include most of Fig. 9 as a supplementary figure and focus on the relevant behavioral properties in the main figure. Showing the changes in much more detail would be helpful. It is also important to show the distribution of the data (e.g., violin plots) rather than bars with SEM because bars with SEM hide a lot of variability and do not allow us to understand the distribution. The heat maps are too small, and the trends among animals are pooled. As discussed in the paper, enhancing or suppressing the inputs leads to similar trends, which is unexpected and raises the possibility that the animal "sees" a blurry stimulus. In that case, looming stimuli might be perceived as having less contrast, which would change the behavior. Ideally, a shelter would be added to elicit escapes (I'm not asking to repeat these experiments). Without a shelter, behavior will adapt after repeated stimulation. In that case, it is important to observe the trend over time, as well as before and after CNO. Given that one of the main claims are based on the behavioural experiments, these should be rock solid. I have the impression that this is not yet the case.

Minor:

Title: I would remove "sophisticated" because I wouldn't define it as such based on the data presented. I also don't think "behavioral control" is not the right term. As mentioned above, having blurred vision or a blurred state wouldn't count as behavioral control, and one doesn't know if this is the case.

Abstract: I wouldn't describe adjusting physiology or behavior according to variations in ambient luminance as a "simple" task. In fact, I would argue that all of these are open questions when examined in detail.

Introduction: Instead of summarizing the experiments that have been done, I would suggest summarizing the results at the end of the introduction.

Results: There are better ways to define the significance of "visual responsiveness." For future analyses, check the zeta-test from the Heimel lab, it is very useful.

Discussion: It should refer to the figures of the results; there are no references to them.

Reviewer #4

(Remarks to the Author)

Version 2:

Reviewer comments:

Reviewer #3

(Remarks to the Author)

Like the other reviewers, I am happy to recommend this paper for publication. While the authors and I have slightly different views on the robustness of the visually induced behavioral effects mediated by the LHA, the data are presented well and readers can draw their own conclusions. These are very difficult experiments, and I acknowledge that. I expect that dedicated behavioral experiments in the future will shed more light on the exact roles, but this is clearly out of scope. Thus, I find the proposed title much better. I congratulate the authors on their thorough work.

We thank the Editor and Reviewers for taking the time to consider our manuscript and for the constructive comments and suggestions. We have now thoroughly revised the manuscript, including the addition of substantial new data and analysis, alongside relevant revisions and improvements throughout the manuscript text. Coupled with our point-by-point response below, we trust the new manuscript addresses all issues raised in the original review.

Reviewer #1 (Remarks to the Author):

The manuscript by Moulard et al. tests the role of lateral hypothalamic area (LHA) on visual processing and visually-guided behavior. By large-scale electrophysiological recording, viral tracing and chemogenetic manipulation in mice, the authors firstly describe the visual processing in LHA neurons in many aspects, revealing that LHA neurons are highly selective for spatiotemporal contrast. It is interesting as hypothalamus is thought to mainly process visual information in a coarse-tuned manner. The authors also attempt to reveal the role of LHA in neuronal circuits of visual threat detection. Overall, the data quality appears good, and results are mostly consistent, however some parts need additional experiments, presentations and/or classification to be fully convincing and to warrant publication.

Major:

1. The authors showed that most LHA neurons exhibit small receptive fields and sensitive to moving stimuli, and attributed this to retino-LHA projection. However, the recorded visually-evoked LHA neuronal activity could also originate from visual thalamus such as dLGN and SC, or others, and the large-scale electrophysiological recording could not exclude this indirect visual pathway from retina to LHA. In the result, the authors had employed retrograde tracing from LHA. It is also necessary to examine whether other visual thalamus and/or visual cortices are targeted, and also their potential role in the visual processing of LHA neurons.

We thank the reviewer for this comment. We are happy to now include brain image data from our retrograde tracing studies which did not reveal retro-labelled cells across LGN, superior colliculus or visual cortex (Fig S6). To our knowledge, no other study has previously reported such connectivity and our present findings align with our examination of the Allen Mouse brain Connectivity Atlas data (e.g. experiments 114754390, 112827164, 100141598, 479670988, 293914766, 147212977, 307297141, 180296424), where no detectable projections from the SC, LGN complex or visual cortex to the LHA region are evident. We did observe some retro-labelled cells in the SCN (Fig S6). While the nature of the labelling approach makes it uncertain whether these cells directly contact the visually responsive LHA neurons, such a projection would, in any case, not explain the observed responses for several reasons: 1) LHA visual responses are both more diverse and sensitive than those of SCN cells (i.e. they appear at intensities at which SCN cells do not respond), 2) We have added analysis of response latencies (Fig S1f and Fig S3e,f) and show that LHA responses are significantly faster than SCN (and LGN cells) recorded under identical conditions/with identical stimuli. We have added some additional discussion around these points (lines: 709-719).

2. In line 84, the authors declared that “the majority of neurons (n=25/44) showed only transient increases in firing”. However, the percentage of this group of neurons is around 57%. Maybe “over half” is more accurate to describe here than “the majority.”

We agree and have changed to read ‘a greater proportion’ (line: 93).

3. The figure 7 targeted the LHA-projecting RGCs via a unilateral injection of AAV2retro-Cre into the LHA, but no data or evidence of injection site has been shown in the manuscript. Although AAV2retro-Cre does not express fluorescence, injection of AAV combined with fluorescent dye could be an appropriate way to label the injection site. Considering that the different parts of nucleus may receive projections from different retinal quadrants, the result that 'LHA-projecting RGCs are highest dense in ventrotemporal and lowest in doronasal retinal quadrant' may need to be corrected by the injected sites in LHA.

While the AAV does not directly encode any reporter, we performed those studies in Ai32 mice (cre-dependent, ChR2-EYFP reporter) such that cells across the brain and retina (and their processes) are labelled. While this naturally means that labelling in the brain is not restricted to the injection site *per se*, we have found that the primary injection site is still readily identifiable based on a region of especially bright fluorescence. We have included example images and schematics of injection sites for all such experiments as well as newly added data using different targeting approaches (Fig S5a,e,l,n,q, Fig S6, Fig6l, Fig S7d). Since the injections on which the specific quantification the reviewer is referring to (now Fig 6a-c and Fig S5a-d) encompassed the full dorsal ventral extent of the LHA, we are confident that this is a reasonable reflection of the overall retinal projections. Accordingly, we consistently found equivalent asymmetry across experiments (which likewise routinely encompassed the full dorsal ventral extent of the retinorecipient part of the LHA; Fig S5).

4. Figure 7f needed to be plotted by both single and merged channels for better visualization.

We have modified as requested (now Fig 6g).

5. In the figure7f, the dendritic field of most EYFP-positive RGCs look asymmetric. Are most LHA-projecting RGCs are asymmetric in their dendritic field? The classification of different types of RGCs need to be done.

We thank the reviewer for this comment. Across the (melanopsin-ve) labelled RGCs we see a range from symmetric to very asymmetric with ~50% of cells of equivalent symmetry to the M1-like melanopsin labelled population. We have adjusted the range of examples in the figure (now Fig 6g) and have added some quantification of asymmetry (Fig 6i; methods text, lines: 1096-1098). Given the spectrum of neuroanatomical properties we identify, while we are confident that the LHA receives input from multiple RGC classes, we do not see a strong basis upon which to go any further in terms of definitive classification of melanopsin-ve RGCs labelled here.

6. Overall, some figures are lack of readability and are always presented too crowd to distinguish different parts. For example, in the figure 10a, diagram of 3D construction is too crowd and the captions are too close to other diagrams. A more concise and readable version of figures is needed.

We are acutely aware that this is large and multifaceted study, which given the range of approaches and associated control data and analysis required makes for rather dense figures. In revising the manuscript we have modified every figure and done our best to declutter and move less essential information to supplemental figures, whilst accommodating additional data and analysis as requested by Reviewers.

Minor:

7. In the figure 5C, the dashed line had not been indicated in the legends

This simply reflected 50% of maximal response, as a reference. We agree however that this is non-essential and have removed from the revised figure (now Fig 4c).

8. I don't quite understand what is the significance of comparing the soma size of RGC projected to LHA and SCN in Figure 7.GH. It seems irrelevant to the main story and unnecessary to appear.

We agree that this is not essential to the main story and have therefore move to supplemental (Fig S5m). We nonetheless think it is of potential interest to some (eg a comment from another reviewer below) to note that the melanopsin+ cells retro-labelled from the LHA have similar morphology to those projecting to the SCN, hence we did not wish to remove altogether.

9. Figure 7 H should to be h.

10. Figure 7b, the V is misplaced under the arrows. The EYP should be EYFP (also in Figure 5b).

We thank the Reviewer for pointing out these errors and have corrected this in the revised figure (now Fig 6).

11. In figure 10, the graphic diagram did not show how the AAV were delivered. Although the method is similar as figure 9, the authors should at least state in the figure that they manipulate the neuronal activity of LHA-projecting RGCs.

We have added this information to the figure (now Fig 9a).

Before resubmission the authors should proofread it carefully, especially for the figures, and the editor should also do so before publication.

We have endeavoured to remove any typographical errors!

Reviewer #2 (Remarks to the Author):

This is a very comprehensive study on the visual properties of the LHA visually responsive cells. Not only are the findings of these cells unexpected, but the responses of these cells are very well documented, with described inputs and outputs and functional manipulation. The study is well executed, using good methodology. The amount of data is impressive. Some issues, however, need to be addressed before recommending for publication in Nature Communications.

One problem with the wording in general is describing LHA as a visual centre. Having, for example, movement activated cells in the visual cortex wouldnt make it a motor centre. The authors do not compare the number of total recorded cells in the visually responsive electrode channels to the number of visually responsive cells.

We take the reviewers point (and it was certainly not our intention to give readers the impression that the LHA is *per se* a visual centre – rather that the retinorecipient part of the LHA processes sophisticated visual signals). Accordingly, we have adjusted the title (now: ‘A lateral

hypothalamic region supporting sophisticated visual processing and behavioural control') and text throughout the manuscript to avoid giving this impression. We think our current treatment is justified given that, in retinorecipient parts of the LHA, we reliably find cells that respond across a range of stimulus dimensions and now provide additional evidence that these are driven directly via the retina rather than via inputs from primary visual regions (eg Fig S1f, FigS3e,f, Fig S6). In this sense, we argue that what we find is different from the Reviewer's example of visual cortex where any movement related signals must, by necessity, arise by a much more circuitous route. As to the Reviewer's question around non visually responsive cells: 1) for our initial mapping we now show the anatomical locations of non-responsive cells in FigS1c, 2) For our subsequent recordings that specifically targeted visually responsive regions, almost every cell we recorded responded to some aspect of our stimulus set (summarised in Fig 5). We have also added a summary table (Table S1) as per the Reviewer's request, below, and enhanced discussion around the question of what fraction of LHA cells are visually responsive (lines: 692-708)

In addition, these visually responsive cells were only found in one part of the LHA. Authors should consider using different language here. As a related point, the results should summarize in one place the n of mice in each of the experiments, the total n of recorded/isolated cells, and the total numbers of visually responsive cells in recordings. The proportion of these cells compared to other recorded cells is not entirely clear.

As per our comments above, we have endeavoured throughout to use our wording carefully to avoid giving the (unintended) impression that our findings apply to the LHA as a whole. We have added Table S1 that provides the summary information, as requested.

Line 43 - Sensory responses (including visual) have been demonstrated in orexin neuron subpopulation of the LHA: see Fig 5. <https://doi.org/10.1016/j.pneurobio.2020.101771> [doi.org] Also for review see <https://www.sciencedirect.com/science/article/pii/S0031938420302663> [sciencedirect.com]

We thank the reviewer for flagging this point and have adjusted the text accordingly (lines: 43-47), which we hope the reviewer considers a fair summary; While sensory responses have been detected with calcium imaging approaches the temporal resolution of such approaches makes it hard to distinguish directly driven responses from those involving multisynaptic pathways and the specific roles/properties of retinal inputs to the LHA have not previously been investigated.

Line 55 - "employing mice where native M-cone opsin is replaced by the human L56 cone opsin (Opn1mwR)" I don't see a short description where the reasoning for this is given. A quick description in the methods or results, on how this exactly helps elucidate the different photoreceptor contributions to visual responses would be useful, especially for those not very familiar with the field.

We thank the reviewer for this suggestion. We have adjusted the last paragraph of the introduction to make it clearer why we use these mice prior to our initial description of their use in the results section.

It would be useful from the authors to report latencies to the described visual responses, such that they may be compared to latencies observed in other visual processing areas.

We again thank the reviewer for this suggestion. We have added this information and, since we have used identical stimulus approaches in some of our previous work recording from other visual targets – have been able to provide direct like for like comparisons (identical stimuli, mouse strains, analysis procedures etc). The relevant information is now provided in (Fig S1f and Fig S3e,f). Notably we find responses are, on average, modestly but significantly faster than those of neurons in SCN and also LGN (consistent with the longer tract length to reach LGN and generally low sensitivity of SCN cells which translates to somewhat slower kinetics).

Line 161, figure 2, S2 - Authors need to either justify the usage of this period or employ a different method. Fig S2d seems to show some differences also in the acute period. Maybe choosing the peak values rather than the mean of the arbitrarily chosen 250ms period after stimulus onset would be less susceptible to subjectivity in analysis. In addition, the results of the t test in figure S2d and others are not well reported. Precise p and t-values should be given.

We thank the reviewer for this suggestion. We have changed to using the peak firing in this and all other related analysis (which has indeed reduced the nominal divergence the reviewer refers to) and now provide precise P-values throughout.

Line 187 - The melanopsin responses are described as 'sluggish' in line 159 to describe the result of no difference in acute portions of the responses in S2d, but here fast melanopsin response dynamics are suggested. Authors should explain / or suggest how melanopsin response kinetics can display both sluggish and fast kinetics, with reference to the method used in each experiment.

We apologise for the lack of clarity in our writing here, we meant fast in the context of other melanopsin driven responses (which vary depending on ipRGC type and irradiance but are typically substantially slower than rod/cone responses). Nonetheless, having revised the analysis as per the Reviewer's helpful suggestion, peak responses in this cell population do not in fact differ significantly at the highest irradiance (as expected for strongly rod/cone dominated responses; Fig 2d). Hence, we have now removed the portion of text in question.

Are the cells in the LHA only representing a small portion of the visual field as shown in 4C or have the authors not investigated the wider area. Authors should discuss the relevance of this small coverage of visual field for the presumed purpose of LHA-based vision. Authors look whether S- and L- cone distributions mirror those of the distribution on the retina (paragraph starting at line 293), but it isn't clear why more of the visual field wouldn't be represented here, perhaps a point for discussion.

We thank the reviewer for this comment. The retro-labelling data we previously presented and now expand upon in the revised manuscript (Fig6, FigS5, Fig8, FigS11) illustrates that LHA-projecting RGCs can be found across the retina but are significantly more common in ventrotemporal portions. That finding aligns with our electrophysiological results (Fig 3c, FigS10d), where >60% cells displayed clearly mappable RFs and >88% responded to spatially structured stimuli, indicating RFs that intersected the region of stimulus delivery (n=193/320 and n=282/320 respectively, combining data from *Opn1mw^R* and GAD-ChR2 experiments). In sum, those findings are compatible with the presence of cells that sample from regions of visual space

outside those we tested but demonstrate that these are comparatively less common (as per our neuroanatomical data). As for the purpose of the enriched input from this part of visual space, the data imply regions in front of the animal and overhead are preferentially represented which seems to make sense in the context of roles in goal-directed behaviours (eg defensive, hunting). We have expanded our discussion of these data to capture these points (lines: 676-681 and 755-760).

In the experiment under 'Identity of retinorecipient lateral hypothalamic neurons' starting at line 454 the authors demonstrate, using tracing methods, that the retinorecipient neurons in the LHA are exclusively glutamatergic. However, in the following paragraph starting at line 470 they verify in-vivo that none of the visually responsive neurons are GABAergic. Having found co-localisation with glutamatergic markers, why didnt the authors instead/in-addition confirm with optogenetic stimulation/suppression that the visually responsive neurons are glutamatergic? This seems like a logical continuation and it is strange that it is omitted.

While we certainly take the reviewers point, unfortunately the proposed experiment is far less informative than it might at first seem. Hence, whereas in GAD-ChR2 mice we can readily distinguish direct ChR2-mediated responses (excitatory) from synaptically driven responses (inhibitory) this would not be the case if we directed ChR2 to glutamatergic neurons. Indeed, given that our optogenetic stimuli would activate the terminals of any glutamatergic cells projecting to the region stimulated, our expectations are that essentially every cell would show rapid excitatory responses to this stimulus (likely involving a combination of direct and synaptically mediated excitation). Certainly, since RGCs are glutamatergic, we would expect any visually responsive cell to respond, whether or not it was itself a glutamatergic neuron. For this reason, when planning the experiments (which we in fact undertook before results of the neuroanatomical tracing were known), targeting GABAergic neurons was a logical choice. As we note in the manuscript (lines: 522-525), we think the inclusion of this data is worthwhile as it provides a useful control against the theoretical possibility that our AAV-based tracing approach might be in some way biased (whilst also providing independent confirmation of key findings from electrophysiological recordings earlier in the manuscript).

In some places in the text, the authors report p-values without reporting the test or the test statistics. For example, line 508, line 517, line 521, etc. The authors should stay consistent with statistical test reporting, either giving a full report in text, or omitting the information in text, just stating that there is/isn't a significant difference and reporting the full statistical report of the test in the associated figure legend. Authors should make this consistent across the entire manuscript and make sure there are full reports for each test (some are partial/missing).

We have endeavoured to ensure that statistics are restricted to figure legends, wherever relevant, full details of tests and precise p values are reported throughout (where >0.001).

Starting in line 523 the authors describe finding no c-Fos expression differences due to chemogenetic manipulation, so they suggest a modulatory rather than driving role of these projections to the hippocampus. This seems quite vague, as modulatory projections should also impact activity in the region during the 90 minutes before the animal was culled. The authors should elaborate or leave out this part.

We agree that, without context this statement is unhelpful and the results section is not the right place to provide such. Accordingly, we removed that piece of text and include some consideration around the point instead in the discussion (lines: 739-743). In sum, it is well established that pyramidal neurons such as those in the hippocampus can receive thousands of excitatory inputs from different sources and often require simultaneous activation of multiple inputs to influence neural activity at the level of the soma (coincidence detection e.g. doi: 10.1038/nrn2286). We, therefore consider it reasonable to suggest that LHA-projections to the hippocampus are insufficient to drive c-Fos expression in their own right under our experimental conditions but could modulate responses under other conditions where additional inputs to the HC are more active (eg when exploring a novel environment).

For the data in Fig 10. the authors run a mixed-effects ANOVA on the behaviour and DREADD effects. Adding so many behaviors in the test, and indeed, using it as a variable, can dilute the effect of p-testing via multiple comparisons. As such. For example, in Fig 10g, body bending seems impacted by DREADDs even though the tone is used instead of visual information. This might indicate that actually there is some kind of integration of visual and sound information in these cells, since blocking (or activating) their visual inputs only can still have an impact on auditory behavioural responses. Authors should consider maybe running a stimuli x DREADD mixed model for different behaviours (which also may have a weird interpretation) or running a series of one-way (repeated-measures) ANOVAs on individual behaviours to determine the effect of DREADDs. Otherwise, they should do more to justify including behaviour as a variable.

There are other problems there (behavioural measures are not independent – an increase in locomotion would necessitate a decrease in freezing, for example), and there are probably other dependencies. In a mixed-effects model, there should be a mix of repeated measures and between subjects tests. Here I assume, although it is not explicitly stated, that behaviour was treated as a between subject variable, which it is not because it is in the same mice, and the measures are not independent of each other.

We thank the reviewer for these comments and apologise for any ambiguity in our reporting of the stats. We, of course, agree that there is partial interdependency across the behavioural components and, as such, felt the mixed effect model we employed was the best choice for analysis. For clarity, in that model the various behavioural components were indeed treated as within-subjects variables while manipulation (Gq/Gi/control) was the between-subjects variable. Hence, our view was that this provided the most robust way to test whether behaviour as a whole, evoked by the specific stimuli, varied in a coordinated manner as a function of experimental groups. Regardless, we have been happy to completely revise the analysis approach, as per the Reviewer's suggestion (Fig 9, Fig S12). We now use a Stimulus X experimental manipulation mixed effect model for each behavioural component separately (Stimulus treated as a within subject variable and experimental group as between subjects). Key effects remain, as per our previous analysis, while we do not in any case find a significant effect of DREADD manipulation on auditory evoked responses.

Figure 10 legend or the corresponding methods section also doesn't explain the difference between delta measures and regular measures and why some were included as deltas and some not. For example, what is delta(rearing)? Is it the period that the animal is changing from another

behaviour to rearing or from rearing to another behaviour, or both, or something else entirely? If it is only behavioural transition to or from behaviour, a simple count would be better.

Again, we apologise for any lack of clarity in our description of the results and methods. We have now enhanced our descriptions (Results, lines: 591-595; Methods, lines: 1172-1213), which we hope now makes everything clearer. In sum there are 3 postural related measures (rear, elongation and bend) and 6 motion related metrics that variously described the rate of change (frame to frame) in mouse posture (Δ rear, Δ elongation, Δ bend), position in the behavioural area (Locomotion, rotation) or immobility (Freezing)

In line 1006 of methods, authors describe no qualitative difference in responses to two different types of auditory or looming stimuli. There is no description in what these qualitative measures could be and why quantitative measurements were not used, given they were measured regardless. Same applies for the inclusion of shelter in the experiment.

We thank the reviewer for making this point. In the methods we have now included full details of statistical analysis upon which we based the decisions to pool data (lines: 1157-1171).

The way that behavioural variables in Fig. 10 were operationalized seems very strange. First of all, using quantile normalization would overrepresent behaviours that don't happen so often as the one that happens the most. Second of all, if each behavioural feature was normalized in the 0-1 range, as described in the methods, how is it that some values in Fig 10 are negative? Was there an additional subtraction of the baseline, and if so which period was used to assess this baseline? If yes, it would only make sense to baseline subtract per trial, but then if the animal was frozen before a trial, there would be a massive skew toward negative if the animal moves after onset (for example). These measures are not so clear even with reference to the cited paper developing this method (<https://www.sciencedirect.com/science/article/pii/S09609822220313385> [sciencedirect.com]).

We realise in hindsight that our efforts to keep the text as concise as possible came at the expense of a full clarification of how the data was treated, for which we apologise. In our revised manuscript we have removed the quartile normalisation and now operate on the raw data (with similar outcomes/conclusions). We are happy, however, to clarify what was shown in the original manuscript (and why), in case it helps. First, for the avoidance of any doubt, we should make it clear that the normalisation was applied across the pooled data (every stimulus, animal and trial) such that that data retained any animal- or stimulus-specific differences in that behavioural component. As such the normalisation simply put each variable onto a common scale between the most we ever observed it and the least. We felt that was important for the analysis approach we considered to be the most powerful (i.e. the Behaviour X Experimental group mixed effect models for each stimulus as discussed above); specifically because this avoided issues with the fact that the different behavioural components have different units and scales. Naturally, this is no longer relevant given our adoption of the alternative analysis approach suggested by the Reviewer.

As to the question of negative values: yes, in the original manuscript, we analysed data as change from baseline (by subtracting the mean across the 2s preceding the stimulus for each trial; reported in figure legends and methods section). In the revised manuscript/analysis we have retained that approach for the data presented in the main figure (Fig 9c,d). Hence, while we agree

that if an animal is strongly engaged in a specific behaviour prior to stimulus onset this would lead to a reduced 'response' in the analysis, the converse is true for the alternative approach. If we simply use the raw values, if an animal is strongly engaged in a specific behaviour prior to the onset of the stimulus it could appear in the analysis as a strong 'response' even if the animal was entirely indifferent to the stimulus. While both approaches have potential flaws we feel the change in behaviour following stimulus onset is, on balance, a more informative metric as to whether the animal is actually responding. Nonetheless, we have been happy to include heatmaps showing the raw behaviour of all animals in the main figure (Fig 9b) and to include equivalent analysis to that shown in Fig 9d using raw (non baseline subtracted) data (Fig S12c-l) so the reader can judge for themselves. As the Reviewer will see, while the latter is a slightly less sensitive analysis approach, key results are retained across both types of analysis.

Is the output of the measure a classifier of behaviours or does it include the probability that a certain behavior is occurring at a certain moment in time? Which behaviours are then allowed to co-occur (eg. Movement and freezing are exclusive of each other, but movement and body bend are not)? Authors should explain much more carefully how these measures were derived, pre and post processed, if they are to use them. Without this, the validity of the figure is difficult to understand currently.

As above, we hope our adjustments to results and methods text now make this much clearer. In summary, the approach is not a behaviour classifier per se – each component can co-occur (with the exception of freezing). We should, in particular, note here that in the original manuscript the parameter we described as 'Freeze' was a continuous measure of immobility (i.e. how little the 3D position of the animal had changed frame to frame). In hindsight, we realise this could cause confusion, insofar as Freeze is generally considered to be a binary event. Hence in the new manuscript we have binarized that variable. As such 'Freeze' necessitates essentially zero values for all other movement related variables, otherwise variables can vary independently.

Minor:

Line 40 – no definition of peri-SON before using abbreviation

Line 284 – vary instead of varying

Line 508 – 'control of Gi-DREADD groups' should be or? Probably typo

Line 580 – visual input written twice

Line 685 - vis instead of via

Line 697 – DFERADD

No legend for panel c) in Figure 4.

Fig 10. - Behavioral instead of behavioural in many places.

Fig S1h – why is it the only figure where it says 405nm? Typo?

We thank the Reviewer for pointing these out and have corrected all in the revised manuscript.

Reviewer #3 (Remarks to the Author):

In this study, Moulard et al. examine the role of the lateral hypothalamic area (LHA) in visual processing and behavioral control, which has traditionally been considered separate from other visual pathways. Using a variety of techniques, they provide the first description of the visual processing capabilities of the LHA and suggest that this activity is involved in the regulation of

behavior. They found that while some LHA cells respond to overall light levels through melanopsin-driven responses, most are highly selective for spatiotemporal contrast, including motion detection. The authors found that the LHA also displays a retinotopic organization, and its neurons project to key behavioral control centers, influencing responses to environmental cues such as flashes of light and looming threats. The authors conclude that this establishes the LHA as a critical visual center for managing behavior and responding to environmental threats.

While the quality of the data presented is solid for the physiology, I have some concerns about the clarity of this part. In addition, I have several open questions that challenge the interpretation of the results, particularly for the second part of this paper regarding anatomy and behavior.

Major

Physiology of LHA

The paper is well written. However, the presentation of the figures can be optimized. The main messages are (i) diversity of responses in LHA, (ii) anatomical characterization of RGC & RGC receiving LHA projections, and (iii) behavioral analysis. For the first part concerning the visual responses in LHA, the message can be streamlined. In my opinion, it would help to make the message more compact and move the less relevant information to the supplement and compress Figs. 1-6. Since the authors have recorded a pallet of stimuli, each figure provides an independent message. However, I think that it would be more adequate to analyze them together, providing a functional characterization of visual responses, rather than making one figure for each of the stimuli used. This would provide an overall picture, which cant be done by the reader if it wouldnt reanalyze the data. A nice example could be the paper by Baden et al. 2016. I dont ask to make the same analysis but to compress the data and look at the interdependencies across stimuli to give a functional classification of the LHA.

We thank the reviewer for this suggestion. Given the unexpected nature of our findings and the range of different data sets and stimulus types involved to support and control for our conclusions, it would not be feasible to condense all the data into a single figure. When putting together the original manuscript we included the range of analysis and control data that we have found reviewers typically want to see when assessing this type of work. Nonetheless, in revising the manuscript we have identified areas where content can be aggregated and/or less critical analyses move to supplemental. As such we have been able to combine what was Figures 2 and 3 (Now Fig 2) and Figures 5 and 6 (now Fig 4) such that each of Figs 1-4 now aggerate related content across datasets. As per the Reviewer's suggestion, we have also added a summary figure (Fig 5) which provides an overview of the different types of responses across the various stimulus types. We feel this is a useful addition, in the context of detail provided by previous figures, but certainly couldn't substitute for presentation of that previous information for a study like this.

The LHA is a long structure also in the anterior-posterior axis, but I couldnt find any direct assessment of where the probe was positioned. While the authors have provided characterization on the dorsoventral and mediolateral axis (suppl. Info), it would be relevant to add information on the anterior-posterior axis too. Would this reveal a retinotopic map? This would reveal if there is any bias in the recording position. This is relevant also for the anatomy. While some information on the anterior-posterior (rostro-caudal) axis is provided for the c-fos counting and data on the

anterograde labeling experiments, nothing is presented for the recording site position nor injection sites apart from “desired stereotaxic coordinates.

We thank the Reviewer for pointing this out. Since we always targeted the same nominal position on the rostral caudal plane (~central within the part of the LHA where retinal projections are found), there is minimal variation between experiments. We have added more specificity to our description in the results and methods (lines: 72-73 and 965-970) around this general point. We have also added analysis of whether RF position does vary systematically across the range tested (Fig S3g) but do not find any significant variation.

Anatomy

The authors used serotypes that do not spread well in the retina, especially AAV1, which are not good serotypes to infect the retina (at least in our hands). They have strong biases in their infections, spreading very little beyond the infection side and having, at least in our hands, a bias towards the blood vessels. This can be changed with AAV2, specifically the AAV2.7M8 strain, the latter has been developed for that particular purpose. So the level of anterograde labeling will be very likely biased and low. It would be important to make a control to show the level of infectivity in the retina by infecting a reporter mouse line (e.g. Ai32) with AAV1 and AAVrb to test for infection biases in the retina. In addition, the retina processes across the LHA should be characterized using classical CTB injections (as a ground truth), which then can be compared with the projection patterns using AAV in the LHA

We agree with the general point that, with any experimental technique, it is important to consider the potential for bias. As such, in the original manuscript we already included appropriate controls and independent confirmation of findings (eg electrophysiological and neuroanatomical) wherever possible and have been happy to enhance this in the revised manuscript.

Regarding our use of AAVs specifically; throughout, we used the commercially available AAV serotypes/constructs that best suited our specific experimental objectives and where there was substantial published data showing their effectiveness. For example, in the case of AAV1 there are many studies that have used this successfully for anterograde transsynaptic labelling of retinorecipient neurons across other visual targets (dLGN, vLGN, IGL, PON, SCN, superior colliculus; PMIDs: 279894590, 32497303, 35190665, 37715263). Whilst we don't disagree with the Reviewer that other serotypes may be better for retinal transfection overall, we are not aware of data showing these other serotypes are suitable for anterograde **transsynaptic** labelling, which was the specific objective of these experiments. In terms of the Reviewer's suggestion that labelling could be biased and low – certainly the efficiency of anterograde transsynaptic labelling is likely to be low. However, the particular objective of those experiments (to label a subset of retinorecipient LHA neurons so we could assess their projection targets and neurochemical identity) does not require on high efficiency *per se*. The question of bias is, of course, more important. The fact that the approach has been used to successfully identify retinorecipient neurons across many other visual targets (see above) already gives confidence that this is unlikely to be a major issue. More importantly though, our complimentary experiments using optogenetic cell identification (Fig S10) as well as effects of chemogenetic manipulation on brain c-Fos (Fig 8)

align with key conclusions from our AAV1-labelling (in terms of projection target and neurochemical phenotype). Nonetheless, we are certainly happy to note the formal possibility that our labelling approach might give an incomplete picture of the projections of retinorecipient LHA neurons in our revised discussion (lines:734-735).

Regarding the use of AAVretro (again very widely used across the literature; eg see PMIDs: 38714284, 30076594, 38246331, 36932080 for examples of broad spectrum RGC labelling), we already included a key control for bias; by demonstrating that the relatively high proportion of melanopsin-ve RGCs labelled was not some peculiarity of the virus (since injections into the SCN label almost exclusively M1-like ipRGCs as expected). We are happy to now include additional data demonstrating the assymetric distribution of retro-labelled RGCs is not some peculiarity of this AAV serotype since we consistently see it for LHA injections but not injections into LGN or SCN (Fig 6, Fig S5). We further show that CTB injections into the LHA of wildtype mice do recapitulate the retro-labelling pattern seen with AAVretro (Fig S5q-t).

As to the Reviewer's suggestion we should perform anterograde CTB tracing of LHA retinal projections; it is not entirely clear to us how this would validate (or not) what we find with AAV1 (insofar as it does not allow for transynaptic labelling). In terms of general labelling of retinal projections to the LHA – many previous studies have already demonstrated these using CTB or other approaches (PMIDs: 16736474, 24889098, 26895233, 33104235, 36932080). As such, we did not see value in replicating what has already been demonstrated several times. We have, however, been happy to include intersectional tracing of LHA projections (Fig 6j-l, FigS7) which both demonstrates a pattern of LHA retinal projections consistent with previous data and results from our AAV1 tracing and electrophysiology, whilst also addressing another comment from the reviewer around collaterals.

Another implicit assumption, at least as the data is presented, is that the projections from the retina to the LHA are specific. They may well be axonal projections from cells that project to multiple brain areas, including the LHA. This would strongly bias the interpretation of the c-fos, behavioral and anatomical results. For example, for the anatomical analysis of LHA projections: after AAV-Cre infection, all RGC recipient areas could be infected. Subsequent secondary infection by AAV-ChR2-cherry could therefore be non-specific (normally many areas that go through the injection site are likely to be labeled. This could occur if the injection site to the LHA crossed other retinal recipient layers. It would be advisable to confirm that this is not the case by whole-brain confirmation of off-targets. In addition, modulating LHA RGC projections, if they are due to collateral axons, would not allow determining the role of LHA in any behavior, since any perturbation of the LHA-projecting cells will ultimately influence many areas at once.

We thank the reviewer for raising these important points. Regarding AAV1 experiments, we did not see anterograde labelled cells outside of the LHA region, we have added representative images to demonstrate this (Fig S9a-c). More generally, in all cases, we target injections specifically so they don't pass through or near to other visual areas. We have made this clearer in the methods (lines: 981-985) and show data for injection sites and confirmation these don't impinge on the nearest visual regions for the other neuroanatomical tracing experiments (Fig S5 a,e,i,n,q, Fig S6). Regarding collaterals – we now include intersectional tracing of LHA retinal projections (Fig 6j-l, FigS7) using intra-LHA injection of AAVretro-DIO-ChR2-EYFP and intravitreal injection of AAV2-Cre. We did not observe any labelling in other visual targets. Likewise, for DREADD studies we now include analysis of c-Fos expression across the LGN complex and superior colliculus and

find no significant effect of DREADD manipulation on these areas. In sum, we are confident that off-target effects have not compromised the interpretation of any of the data presented.

Most concerning, however, is the low resolution of the images shown in Figure 8. Anatomical images are pixelated. I can't judge the data well at this resolution, but it seems to me that fixation is poor, resulting in autofluorescence that looks like processes. The poor fixation seems to be evident in the HC image (see holes in the HC) and spots across the HC and strong autofluorescence in the lower parts of the image. The processes in the HC might be autofluorescence of blood vessels, which sometimes can be confused with processes. Not certain about this, but it would fit the poor fixation (holes nearby). Moreover, the example sections show single cells in the LHA, but the septal region appears to be filled with processes. This is inconsistent by orders of magnitude. I'm not an IHC expert, but in my layman's eye, this should be evaluated and the quality and resolution should be improved to be able to judge Fig. 8.

We apologise for the resolution of the images in the original manuscript which has made it hard for the Reviewer to assess the degree of fibre labelling we observed. We have re-imaged the sections and added representative images from an additional animal in each case from which we hope the degree of specific fibre labelling we see in each region is now much clearer (Fig 7f-h). As to the suggestion that labelling in the HC might be autofluorescence from blood vessels, our finding these labelled processes significantly colocalise with VGLUT2 (but not VGAT; Fig 7i,j; Fig S9d-g) is incompatible with such a notion, nor would this obviously explain why labelling specifically localised to the SLM layer of the hippocampus (rather than throughout the brain). Consistent with our findings, untargeted anterograde tracing from the LHA region shows an equivalent projection specifically to the SLM layer of the hippocampus (as well as the septal and habenula regions where we here report labelled fibres; PMID: 35579973). Similarly, retrograde tracing from the hippocampus reveals labelled cells in precisely the regions of the LHA where we find retinorecipient neurons (PMID: 33935658); we have updated our consideration around these points in the discussion (lines: 728-742). By contrast, regarding the Reviewers comment about 'holes in the HC' – these are blood vessels that run transverse through the plane of section. These are commonly visible in coronal sections of the hippocampus across the literature rather an indication of poor fixation (eg see example images below from the Mouse Brain Atlas of Paxinos and Franklin, 2001; RFig1).

Figure Redacted

RFig 1. Example plates from the Mouse Stereotaxic atlas (Paxinos & Franklin, 2001) illustrating transverse running blood vessels in coronal sections of mouse hippocampus.

Behavior

Similarly, the DREADD experiments may also be biased by the infections. Therefore, it would be advisable to show (i) where in the retina the RGC are labeled (because of stereotactic retro-AAV infections in the LHA and biased AAV serotypes in the retina). and (ii) clear examples with labeled DREADD processes across the LHA. In addition, one needs to confirm that the RGC projections are specific to the LHA, otherwise the authors cannot conclude any behavioral effects. This is required for the anterograde labeling, as well as for the retrograde labeling experiments. At least the c-fos labeling suggests that RGCs that project to LHA also project to other areas. If you observe the SCN c-fos labeling it seems drastically different when excitatory or inhibitory

DREADDs are used. The position of the cells seems to change depending on whether excitatory or inhibitory DREADDs or controls are used (Fig. 9d, see the distribution of c-fos expression in the SCN). While the quantification shows something else (Fig. 9e), the raw data presented suggests that this is not consistent with the quantification. I also find it intriguing that outside the LHA more c-fos labeled cells can be seen. How do the authors interpret these results? This is key, how would the authors determine the relative role of the behavioral effect between SCN and LHA (or other retinorecipient areas)? In my view, the most parsimonious explanation would be that the manipulations alter the SCN and thus one would observe changes.

We are happy to now include several examples of the distribution of labelled RGCs in the retina of these animals (Fig 8b, Fig S11a), where we consistently observed a pattern of RGC labelling equivalent to our previous tracing studies.

Regarding DREADD labelling in the brain, when planning these studies, we expected to be able to use this to provide precisely the information the Reviewer requests. In fact, however, while we reliably found a subset of mCherry labelled cell bodies in retina, we could not detect fibre labelling in the brain (in the LHA or elsewhere). In hindsight we see we are not alone in this finding - other relevant studies that have done something similar with chemogenetic manipulation of RGCs also do not show (or find) fibre labelling in the central targets (eg PMID: 35263618, 33171117; 30795900; 31333190; 34433830). It may be that the timecourse required for robust terminal expression is much longer than the 4-weeks used here (and typically) and/or that the DREADD ligand manipulations used to validate DREADD function drive some transient receptor internalisation (see PMIDs: 26889809 and 34060867 for reviews). In either case, while this precludes direct visualisation of the targeted region, multiple lines of evidence provide confidence in the selectivity of our manipulation: i) the number and distribution of DREADD expressing RGCs aligns our previous tracing studies, ii) we find no evidence that LHA-projecting RGCs send significant collaterals to other visual regions (Fig 6l, FigS7), iii) in none of our tracing experiments did we see any evidence our injections impinged on other visual targets (Fig 6, Fig S5-7) and, most critically, iv) we find significant modulation of c-Fos in the LHA and identified LHA target regions (Fig 8d-i) but not other visual regions (SCN, LGN, Superior colliculus; Fig S11). We have added a section to the discussion summarising these points (lines: 743-754).

Regarding the Reviewer's final points about c-Fos and effects on the SCN being the most likely origin, we do not see a valid justification for this opinion. While we are uncertain exactly what the Reviewer was seeing in the images, it will always be dangerous to draw conclusions based on a subjective comparison of two images. We suspect a particular issue that might have misled the Reviewer here is the fact the different brain regions naturally have different levels of spontaneous c-Fos. Whereas basal c-Fos expression in the LHA is low under our conditions, the SCN has both a very high cell density and high levels of spontaneous c-Fos expression (as expected, e.g. PMID: 15748791). As such random differences in the SCN between a pair of images might subjectively appear more salient. Our quantification is based on counting multiple sections/hemisections per animal (Methods lines: 1135-1147). For our SCN and LHA analysis this typically comprised 7 (bilateral) sections/animal spanning the full rostral caudal extent of the SCN and the anterior portion of the LHA that receives retinal input (counted by an experimenters blind to group assignment). That quantification clearly shows there is nothing approaching a significant difference across groups in the SCN (now Fig S11c). To aid the reader in getting a better feel for the data, in the revised manuscript we increased the number of representative images for LHA (now Fig 8d) and especially SCN, to cover the full rostral caudal extent (now Fig S11b). As we hope is now evident, there is no systematic between group differences in the SCN Fos expression (nor

in other visual targets, Fig S11d-g). As an aside, we also note that SCN neurons are all inhibitory GABAergic cells. Accordingly, while our data rule out the mechanism hypothesised by the reviewer, that theoretical mechanism would also be wholly incompatible with the fact our Gq-DREADD manipulation increase c-Fos expression in the LHA and its identified target regions.

In sum, we do not see any meaningful basis on which one might conclude anything other than retinal projections to the LHA as the most parsimonious origin for the observed effects of our chemogenetic manipulations.

Finally, to examine if the behavioral effects are specific, as stated in this manuscript, it is important to look at the absolute data, not the non-normalized data, and to look at the statistics again. The normalized mean can hide a lot of interesting dynamics and I would suggest presenting all data points. One possibility is to plot the absolute speeds as a heatmap, but I leave it to the authors to find a convincing way to describe their results. There are interesting complexities hidden by normalization and averaging. Also, I wouldnt call the behavioral responses “high-dimensional”, particularly if the normalized mean of, e.g., locomotion, is reported. Also, it is strange to plot freezing as the way it is presented. Freezing is normally defined as a time of absolutely no behavior. Thus, if the normalized values are presented, it is very difficult to understand these plots. What is a “positive” value for freeze, for example?

We thank the reviewer for these helpful comments and suggestions. We are happy to now show heatmaps with raw data for all animals (Fig 9b) and have changed analysis to remove the normalisation, including analysis on baseline subtracted (Fig 9c,d) and entirely raw data (Fig S12). We have changed our description of responses from ‘high-dimensional’ to multidimensional. Regarding ‘freezing’, as we note above in a response to another reviewer: the parameter we originally referred to as ‘freeze’ was really a continuous measure of immobility. We have changed to now use a binary definition (methods: 1192-1196). For the data presented in the main figure (where for each animal we show the average across trials) we are therefore plotting the probability of freezing within a specific-time bin relative to stimulus onset. We have additionally added some analysis of freeze duration (Fig S12l; methods text: 1206-1209), which likewise reveals a specific effect of chemogenetic manipulation for looming stimuli.

Minor

I won't judge the details in the text and structure, as the text is well written, but I would encourage the authors to extract the interdependencies in physiology. This would compress the figures, and the text will probably need major changes.

We have made appropriate adjustments to the text to accommodate changes made throughout the manuscript.

- It is not clear why the authors start with their Opn1mw mice in the first figures. I would present WT animals and only use the Opn1mw data to specifically address the question of melanopsin cells in LHA. This is fine, but I think it would be easier to build a case.

We take the reviewers point and thought carefully about how best to integrate the range of different datasets collected as part of this study. As it stands, our initial large scale MUA screening

was performed in *Opn1mw^R* mice (as were most of the key experiments). As such we think it makes most sense to start with that data.

- Fig. 1. Given the small number of cells, instead of presenting the average of 1 cell, I would like to see the raster plot of a cell with a different presentation and then summarize the average as a heat map sorted by "transientness", this will provide a better understanding of the variability of responses (for LHA and SCN). I also prefer to see firing rates, more than changes in rate.

We thank the reviewer for this suggestion. For Fig 1d,e (and related Fig S1d) we show population averages and heatmaps for all cells to illustrate variability within/between categories (as determined by the objective classification criteria employed, methods; lines 890-893). Where we are plotting single cell data, throughout, we show absolute firing rates. For population averages (or firing heatmaps requested by reviewer), we consider changes in firing rate to be far preferable because variations in baseline between cells add variability that obscures to salient information.

- Fig. 4. The authors report a bias towards frontal RF. It is not clear to me whether this bias is due to the recording position.

We have provided fairly extensive coverage of the relationship between RF-field position/sensory properties and recording position in the manuscript and do not see any clear evidence that RF azimuth position varied according to recording location (Fig S3g,h,j).

- Fig. 6. The standard way to show changes in firing rate would be to show the absolute rates, not the relative change. Overall, the message would be the same, but it would allow the reader to better scrutinize the data.

We have changed this (now Fig 4i) to show absolute rates.

- Fig. 7b, there V is misplaced in the dorso-ventral schematic, and it is not clear why the cell number count is needed.

We thank the reviewer for pointing this out. We have corrected and removed the cell counts.

- Fig. 7 GH. It seems that the distribution of dendritic size of mel + cells is different between SCN and LHA, can the authors comment?

Following a comment from another reviewer we moved the SCN data to supplemental (now Fig S5m). As in the original manuscript, we did perform a statistical comparison of the distribution of soma sizes and dendritic field diameter for Mel+ cells projecting to LHA and SCN and do not find any statistically significant difference (Results, lines: 431-432).

We thank the Editor and Reviewers for taking the time to assess our revised manuscript and are please to see that two of the Reviewers found this to address all the points raised. We are happy to now provide a further revised manuscript which, alongside our point-by point response below, we trust addresses all the additional points noted by Reviewer 3.

Reviewer #1 (Remarks to the Author):

After reviewing the authors' revisions in response to my initial comments, I find their modifications satisfactory. The revised manuscript has adequately addressed my concerns, and I support its acceptance in its current form for publication in Nature Communications.

Reviewer #2 (Remarks to the Author):

The authors have sufficiently addressed all of the raised concerns.

We thank both these Reviewers for their constructive comments and are pleased that our revised manuscript satisfied all the issues raised.

Reviewer #3 (Remarks to the Author):

Mouland et al. did a good job revising some of my previous comments. The presentation and results of the visual response properties of LHA are convincing and, in my opinion, represent the main finding of this work. The authors did thorough work, and I applaud them for it. I don't have concerns about the data presenting the visual properties of LHA. However, I still have concerns about the interpretation and data in the second section regarding the indication that the response properties are specific retinal projections to the LHA and that this activity is relevant to complex behavioral control. Since these are core points the authors want to bring up, as seen in the title, I have listed a remaining weak points.

We are pleased that our revisions largely addressed the Reviewer's previous comments and thank the Reviewer for the constructive points raised.

First, regarding direct projections from the retina, the paper implicitly claims that ~10% of the neurons are visual and that they inherited their properties from retinal cells and not from other pathways. The direct influence is inferred by specific projections to the LHA, mainly through retrograde tracing experiments that do not show collateral projections to other visual areas, as well as the response dynamics and measured receptive fields match the position of the retina. These are shown to correlate with c-Fos labeling. However, as with their mCherry results, long-range projections may require more time to become apparent. I recently learned that ChR2 is not actively transported and thus requires time to reach distant axon terminals because it passively diffuses through the Golgi apparatus. Therefore, the absence of these processes in other areas could also be due to this. Also, it is unclear whether the right sections are being tested, at least for the SC. It seems that these sections are posterior, which would represent a different retinotopic space. The resolution of the images is too low for a thorough analysis (they are too pixelated when zoomed in, Fig. 6l and supplement). I believe the authors, but I would carefully check all rostral and caudal slices in the SC. Second, the argument that the responses are fast is weak, particularly if the timing of responses to visual stimuli onset is not tested and compared to other visual response areas. I'm not asking to make any new experiments, but just want to

mention that having recorded in the LHA with DREADDs in the retina, would have allowed to specifically show which functional aspects are inherited and which not from direct retinal projections and having done pseudorabies tracing from the LHA with reporters that are generally used to visualize projections, would have been the clearest way to show the specificity to the LHA. Thus, currently I am still not yet convinced that these projections are highly specific. However, I would be fine with de-emphasizing this point and rewording the paper a bit. The LHA clearly has visual response cells that have been thoroughly characterized, and I don't think it's relevant to state that visual activity is only acquired via retinal projections or if there are other pathways that might influence the activity too. So this, I think, can be reworded.

We thank the reviewer for raising these points. In our previous revision we included fairly substantial discussion to the effect that the weight of evidence points to direct retinal projections as the primary origin of the responses we observed but that we couldn't exclude a contribution of multisynaptic pathways. We have been happy, in this second revision to slightly temper that statement, expand on the relevant evidence and include the Reviewer's suggestions above about possible future experiments to confirm origins (lines 715-740). In sum we trust these changes address the Reviewers concerns. For clarity, we also include below some additional discussion around specific points mentioned by the reviewer:

'The direct influence is inferred by specific projections to the LHA, mainly through retrograde tracing experiments that do not show collateral projections to other visual areas'

In addition to our own retrograde tracing data, there is a wealth of previous studies indicating RGCs project to the LHA region; cited in the manuscript and highlighted in our previous response (e.g. PMIDS: 16736474, 24889098, 26895233, 33104235, 36932080). As to the specific question of whether these could fully account for the LHA visual responses, the presence or absence of collaterals to other visual regions would have no bearing. Rather, the question is whether visually responsive neurons in the LHA receive input from other visual regions. In our previous revision we presented data from our intra-LHA retrolabelling studies confirming that we do not detect retrolabelled cells in the visual thalamus, superior colliculus or visual cortex (Fig S6). In our response to the Reviewer that requested this analysis, we also noted this aligns with anterograde tracing data from the Allen Mouse Brain Connectivity atlas (e.g. experiments 114754390, 112827164, 100141598, 479670988, 293914766, 147212977, 307297141, 180296424), where no detectable projections from the SC, LGN complex or visual cortex to the LHA region are evident. Insofar as we are also not aware of any published study has demonstrated any direct input to the LHA from visual regions, if LHA visual responses were indirectly inherited it would have to be via a quite circuitous route. We now made these points clearer as part of the revised discussion (lines 715-740).

'I recently learned that ChR2 is not actively transported and thus requires time to reach distant axon terminals because it passively diffuses through the Golgi apparatus. Therefore, the absence of these processes in other areas could also be due to this'

As detailed in the relevant Methods sections of the manuscript (lines 1038-1050), for all of the neuroanatomical studies presented animals were left a minimum of 4 weeks post-injection before data collection. This is at the upper limit of what has commonly been used in previous studies using fluorescently tagged ChR2 or other similar reporter constructs for tracing long-range projections (Most commonly 3 weeks; e.g. PMIDs: 28466070, 16600853, 21483674, 28228269, 36932080 have all observed robust superior colliculus labelling following intravitreal

injections with recovery times <4 weeks). We have been happy to include some discussion around this point in our newly revised manuscript (lines 764-768).

‘Also, it is unclear whether the right sections are being tested, at least for the SC. It seems that these sections are posterior, which would represent a different retinotopic space.’

We thank the reviewer for this comment. The sections actually reflect rather anterior regions of the SC (aligning with regions preferentially receiving input from temporal retina). We did not see labelling in any of the SC sections we collected and have been happy to add additional representative sections to reinforce that point, including even more anterior sections (Fig S7).

‘Second, the argument that the responses are fast is weak, particularly if the timing of responses to visual stimuli onset is not tested and compared to other visual response areas’

We are surprised by this comment given that our revised manuscript included direct like-for like analysis showing that LHA visual responses were on average faster than SCN and visual thalamic cells (Fig S1f and S3e,f). Those findings also align with published data from other groups for LGN and SC visual responses (eg PMIDS: 30281795, 12944530, 23486939). We have made these points clearer as part of our enhanced discussion relevant to this point (lines 721-724).

I’m more cautious about the behavioral experiments. They need more thorough analysis, particularly of effects that seem to have statistical significance. For example: The statistics are all normalized to the change, but this could be due to minor changes in overall locomotion activity. The heat maps showing the change. In my opinion, it would be best to include most of Fig. 9 as a supplementary figure and focus on the relevant behavioral properties in the main figure. Showing the changes in much more detail would be helpful. It is also important to show the distribution of the data (e.g., violin plots) rather than bars with SEM because bars with SEM hide a lot of variability and do not allow us to understand the distribution. The heat maps are too small, and the trends among animals are pooled. As discussed in the paper, enhancing or suppressing the inputs leads to similar trends, which is unexpected and raises the possibility that the animal “sees” a blurry stimulus. In that case, looming stimuli might be perceived as having less contrast, which would change the behavior. Ideally, a shelter would be added to elicit escapes (I’m not asking to repeat these experiments). Without a shelter, behavior will adapt after repeated stimulation. In that case, it is important to observe the trend over time, as well as before and after CNO. Given that one of the main claims are based on the behavioural experiments, these should be rock solid. I have the impression that this is not yet the case.

We appreciate the Reviewer’s caution here although it seems from these comments that our previous response to this Reviewer was, perhaps, not clear enough about the revisions we have already made to the manuscript. Most notably, regarding the Reviewers initial comment, our revised manuscript already included parallel analyses of both the change in behaviour as well as absolute post-stimulus behaviour (Main and supplemental figure respectively). As we discussed in detail in our previous response to Reviewer 2, while both approaches have their pros and cons, we consider the former on balance more informative (since it helps to minimise variability unrelated to the stimulus-evoked response). Regardless, key results are retained across both analyses (as well as relative to the analysis presented in the original manuscript). Our previous revision also included another analysis of stimulus-evoked freezing duration (Fig S12g in this second revision), which likewise used absolute (not baseline corrected data) and again

confirmed conclusions of other analysis. In sum then, we have already confirmed our conclusions hold across multiple analyses with and without adjusting according to pre-stimulus behaviour. In addition, for the avoidance of doubt, the heatmaps presented in the main figure already showed absolute behaviour (not baseline normalised, mean and variance of baseline pooled across animals/stimulus types was simply used to define a consistent colour scaling to facilitate comparisons across stimuli/groups). Alongside the other plots presented in the main and supplemental figures, we felt this already provided the reader with an appropriate range of visualization and analyses of the data. We are happy, however, to include further modifications and additions to address the Reviewers new comments above.

Firstly, we have replaced bar graphs throughout the relevant main and supplemental figures with box and whisker plots so the reader can better gauge the data distribution. We have also reorganised presentation of data across the main and supplemental figures. We have restricted data presentation in the main figure (Fig 9) to the behavioural components that are most robustly modulated by our DREADD manipulation, with other behavioural components now moved to supplemental (Fig S12). The increased space this provides has allowed us to increase the size of the heatmaps and to relevant present analysis of both baseline normalised and absolute data side by side the main figure (Fig 9d,e) and supplement (Fig S12c,d).

With respect to the Reviewers further comments we have also been happy to add some additional analysis and discussion (Fig S13, lines 823-824), which alongside the additional contextual information below we trust address all the reviewers remaining points.

We, of course, agree with the reviewer that animals are likely to show some behavioural adaption upon repeated stimulus presentation. Our past experience has been that this is relatively minimal in our paradigm (PMID: 31316114, 33007242). Nonetheless, with the explicit purpose of limiting any adaptation/habituation, we did not collect data pre-CLZ injection in the present study. Rather, animals received injections 45min before being introduced to the arena for the first time (such that testing occurred during the time window when DREADD-mediated effects would be maximal; eg PMIDS: 32193397, 27426512, 27893096). As we noted in the original manuscript, for these studies animals did receive a shelter during the second half of the experiment but we did not detect significant any differences in behavioural responses before vs. after availability of the shelter under our conditions (lines: 1192-1201). Nonetheless, as part of our expanded discussion around these data, we are happy to note that assessing the impact of adjusting parameters of the stimulus or context in which the animal experiences it will be interesting questions for future work (in line with the Reviewers comments above, lines: 831-834).

For the present study, we choose to average data for each animal across trials for each stimulus type across their exposure to the test arena. Given inherent within-individual trial-trial variability in behavioural assays of this type, we consider this to be the most robust analysis approach. Whilst the conclusions of our analyses (that RGC inputs to the LHA act to modulate visually-guided behavioural responses), stand whether or not any adaptation occurs during that time, we have happily added some additional analysis to explicitly address the extent to which averaging data across trials could have influenced our conclusions (Fig S13). Specifically, we find no evidence that the observed effects of DREADD manipulations on looming-induced freezing and flash-induced locomotion change significantly as we include progressively more trails, beyond the very first exposure, in our averages (effects of experimental group but no interaction with number of trails averaged). We confirm this both for the baseline corrected data (Fig S13a,c) and absolute post-stimulus behaviour (Fig S13b,d). Within group variance, of course, tends to reduce as more trials are averaged but the fundamental nature of the effects we report is evident even

from the first trial (i.e. greater loom-associated freezing and reduced post-flash flash locomotion). Alongside the results presented in Fig 9 and Fig S12, these findings confirm our conclusions are robust and not dependent on specific choices made at the analysis stage regarding adjusting for pre-stimulus behaviour or pooling data across trials.

Minor:

Title: I would remove "sophisticated" because I wouldn't define it as such based on the data presented. I also don't think "behavioral control" is not the right term. As mentioned above, having blurred vision or a blurred state wouldn't count as behavioral control, and one doesn't know if this is the case.

We think the visual response properties presented are 'sophisticated' in the context of what is expected for the hypothalamus but are happy to revise that part of the title. Likewise, we think it is clear from our data that LHA visual inputs modulate visually-guided behaviours but see where the reviewer is coming from with their comment. We have therefore adjusted the title to read: 'A lateral hypothalamic region supporting diverse visual processing and modulation of visually-guided behaviour'

Abstract: I wouldn't describe adjusting physiology or behavior according to variations in ambient luminance as a "simple" task. In fact, I would argue that all of these are open questions when examined in detail.

We take the Reviewer's point. In this context by 'simply' we meant serving just that one function (as opposed to other roles). We have been happy to revise the text to read 'serving to solely adjust'

Introduction: Instead of summarizing the experiments that have been done, I would suggest summarizing the results at the end of the introduction.

We thank the Reviewer for this suggestion and realise there are different schools of thought on this point. Given that the introduction immediately follows the abstract, which does summarise the key findings, we think a brief summary of how we set out to tackle the question is a more appropriate prelude to the Results.

Results: There are better ways to define the significance of "visual responsiveness." For future analyses, check the zeta-test from the Heimel lab, it is very useful.

We thank the Reviewer for this suggestion which we will consider employing in the future.

Discussion: It should refer to the figures of the results; there are no references to them.

We have been happy to add some references to figures supporting key aspects of the results under discussion.

We thank the Editor and Reviewer for taking the time to review our work and are delighted that the study has now been accepted for publication.

We have revised the manuscript in line with the editorial requests as detailed in the attached checklist.

Regarding comments from the remaining reviewer:

Reviewer #3 (Remarks to the Author):

Like the other reviewers, I am happy to recommend this paper for publication. While the authors and I have slightly different views on the robustness of the visually induced behavioral effects mediated by the LHA, the data are presented well and readers can draw their own conclusions. These are very difficult experiments, and I acknowledge that. I expect that dedicated behavioral experiments in the future will shed more light on the exact roles, but this is clearly out of scope. Thus, I find the proposed title much better. I congratulate the authors on their thorough work.

We thank the Reviewer for their comments and are please to see that no additional revisiosn are requested.